# Great Models Think Alike and this Undermines AI Oversight

**Shashwat Goel**[∘ 1 2]   **Joschka Strüber**[∘ 3 4]   **Ilze Amanda Auzina**[∘ 3 4]   **Karuna K Chandra**[∘ 5]
**Ponnurangam Kumaraguru**[5]   **Douwe Kiela**[6 7]   **Ameya Prabhu**[∘ 3 4]   **Matthias Bethge**[3 4]   **Jonas Geiping**[1 2 3]

🗄 Sample-wise Predictions    🌐 model-similarity.github.io    🐙 lm-similarity

## Abstract

As Language Model (LM) capabilities advance, evaluating and supervising them at scale is getting harder for humans. There is hope that other language models can automate both these tasks, which we refer to as "AI Oversight". We study how model similarity affects both aspects of AI oversight by proposing *Chance Adjusted Probabilistic Agreement* (CAPA): a metric for LM similarity based on overlap in model mistakes. Using CAPA, we first show that LLM-as-a-judge scores favor models similar to the judge, generalizing recent self-preference results. Then, we study training on LM annotations, and find complementary knowledge between the weak supervisor and strong student model plays a crucial role in gains from "weak-to-strong generalization". As model capabilities increase, it becomes harder to find their mistakes, and we might defer more to AI oversight. However, we observe a concerning trend – model mistakes are becoming more similar with increasing capabilities, pointing to risks from correlated failures. Our work underscores the importance of reporting and correcting for model similarity, especially in the emerging paradigm of AI oversight.

## 1. Introduction

Machine Learning model capabilities have improved immensely over the last few years. Scaling up the amount of data used for training has played a crucial role in these improvements (Kaplan et al., 2020). Initially, most of the gains in Language Model (LM) capabilities came from scaling pretraining data (Llama Team, 2024a). Recently, there

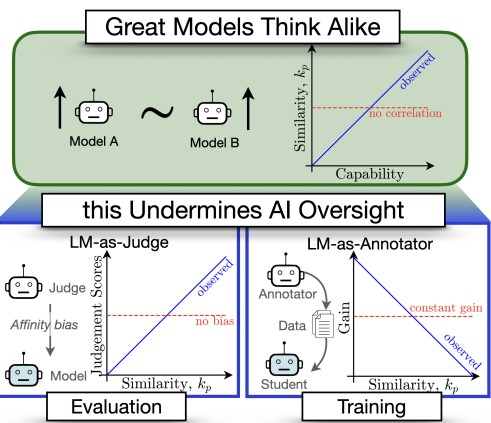

*Figure 1.* **Our Main Contributions**. We develop a novel probabilistic metric for model similarity, CAPA ($\kappa_p$), which adjusts for chance agreement due to accuracy. Using this, we find (1) LLM-as-a-judge scores are biased towards more similar models controlling for the model's capability (2) Gain from training strong models on annotations of weak supervisors (weak-to-strong generalization) is higher when the two models are more different, (3) Concerningly, model errors are getting more correlated as capabilities increase.

is increasing interest in post-training, either with human preferences (Ouyang et al., 2022), or task-specific expert annotations (Lightman et al., 2024). Collecting human preferences or annotations is slow and expensive. Therefore, with increasing model capabilities, an attractive alternative is to use LMs to annotate training data (Gilardi et al., 2023) and score model outputs (Zheng et al., 2023), to boost both training (Stiennon et al., 2020) and evaluation (Li et al., 2024b). In this paper, we refer to both these techniques together as *AI oversight*[1].

Can we rely on AI oversight going forward? This remains a topic of much debate. In this work, we study oversight from the perspective of model similarity. When assessing or teaching humans, it is well recognized that individuals have different strengths and weaknesses. Similarly, two models with 50% accuracy may misclassify completely different samples and thus be highly dissimilar (having differ-

[1]ELLIS Institute Tübingen [2]Max Planck Institute for Intelligent Systems [3]Tübingen AI Center [4]University of Tübingen [5]IIIT Hyderabad [6]Contextual AI [7]Stanford University [∘]Core contributors. Correspondence to: Shashwat Goel <shashwatnow@gmail.com>.

*Proceedings of the 42nd International Conference on Machine Learning*, Vancouver, Canada. PMLR 267, 2025. Copyright 2025 by the author(s).

---

[1]The term is inspired by "scalable oversight" (Bowman et al., 2022), which studies human-in-the-loop mechanisms for AI Safety.

ent 'strengths'). To measure model similarity, we build on *error consistency* (Geirhos et al., 2020), which measures overlap in the samples where two models err beyond what is expected by chance due to the two models' accuracies. In Section 2, we extend the error consistency metric in two crucial ways – 1) by counting differences in predictions rather than correctness for each sample, and 2) incorporating output probabilities. In this way, our novel similarity metric, *Chance Adjusted Probabilistic Alignment* (CAPA), allows us to quantify functional similarity between models. We use this to analyze both evaluation and training using AI oversight as depicted in Figure 1:

**1. LLM-as-a-Judge.** Prior work has shown that LM judges are biased towards their own generations (Liu et al., 2024; Panickssery et al., 2024). It might seem possible to avoid this concern by simply using a different model as the judge. However, just like human evaluators prefer candidates with similar traits (Bagues & Perez-Villadoniga, 2012), could LM judges also exhibit this *affinity bias*? In Section 3, we study this using CAPA, finding LM judges indeed assign higher scores to models that are more similar to themselves.

**2. Training LMs on annotations of other LMs.** Next, we study the effect of similarity on inter-LM training setups, where one model annotates data used to train another model. We hypothesize that performance gained through such training leverages complementary knowledge among models, and is thus inversely proportional to CAPA. We investigate this hypothesis in Section 4, following the weak-to-strong generalization setup (Burns et al., 2024), where a strong (larger) student model is shown to outperform the weaker (smaller) supervisor whose annotations it is fine-tuned on. Indeed, we find performance gains are higher when the weak supervisor and the strong student model are more different. Moreover, our findings indicate a higher performance ceiling for weak-to-strong generalization than previously estimated, if the weak supervisor's complementary knowledge is leveraged effectively.

**3. With increasing LM capability errors are becoming more correlated.** AI oversight is gaining popularity as capabilities increase. The above results show the benefits of diverse models for AI oversight – less similarity between models reduces bias in LLM-as-a-judge, and also leads to greater gains when training on LM annotations. Unfortunately, in Section 5 we find that as popular frontier LMs become more capable, their mistakes become more similar as captured by CAPA. This trend indicates a risk of common blind-spots and failure modes when using AI oversight, which is concerning for safety (Kenton et al., 2024).

Overall, our work proposes a novel probabilistic metric for model similarity, and demonstrates the risks of correlated mistakes in the emerging paradigm of AI oversight. We hope the community shifts towards releasing sample-

wise model predictions alongside benchmark scores (Burnell et al., 2023; Ghosh et al., 2024), as they enable richer analysis like measuring similarity.

## 2. Methodology: Measuring LM Similarity

We begin by describing how we quantify model similarity.

### 2.1. Background

**Functional similarity**: Prior work on model similarity has focused on two classes of similarity measures: representational and functional similarity (see Klabunde et al. (2025) for a recent survey). *Representation similarity* metrics focus on the weights and activations of the networks (Kornblith et al., 2019), comparing how features are represented internally. In contrast, *functional similarity* metrics focus on the input–output behavior of the model. Functional similarity metrics are more broadly applicable than representation metrics as (1) they allow comparisons across model families and architectures and (2) are applicable for models behind an API (where weights are not released). Functional similarity metrics are more interpretable because they operate on data samples rather than noisy, complex internal representations (Golechha & Dao, 2024). Despite large architectural differences across models and model families, their outputs can still be fairly similar. Moreover, Geirhos et al. (2020) argue models with similar internal mechanisms make similar mistakes, and thus mistakes can proxy whether models use similar internal mechanisms. Therefore, in the present work we focus on functional similarity metrics.

**Error Consistency**: A popular similarity metric designed in the context of comparing mistakes of image-classifiers to humans is error consistency (Geirhos et al., 2020). It quantifies the overlap on samples where two models make mistakes while normalizing for chance overlap due to accuracy. First, they define $c_{\text{obs}}$ as the "observed error overlap" i.e., the fraction of samples on which both models are correct or both models are wrong. This itself is used a metric in recent work on LM similarity, Dutta et al. (2024). However, as Geirhos et al. (2020) point out, $c_{\text{obs}}$ has a crucial shortcoming: two independent models with high accuracy will have a higher $c_{\text{obs}}$ by chance than two models with low accuracy (❶). An independent model here is one that is correct on a uniform random subset (size corresponding to accuracy) of samples, and wrong on the others. For instance, two independent models with 90% accuracy will agree on at least 81% of the samples by chance, whereas for two models with 50% accuracy, the lower-bound on chance agreement drops to 25%. Consequently, to account for this, Geirhos et al. (2020) calculate the "expected error overlap" ($c_{\text{exp}}$) as $c_{\text{exp}} = \text{acc}_1 \cdot \text{acc}_2 + (1 - \text{acc}_1)(1 - \text{acc}_2)$ where $\text{acc}_i$ is the accuracy of model $i$. Similar to Cohen's $\kappa$ (Cohen, 1960), error consistency (Eq. 1) is then defined as the fraction of ex-

*Table 1.* **Comparison of Functional Model Similarity Metrics**. Only our metric, CAPA, satisfies all three desiderata:
**❶** *Adjusts for accuracy* – The metric should not inflate scores for high accuracy model pairs due to lesser scope to disagree.
**❷** *Distinguishes different mistakes* – The metric should consider different wrong predictions as a disagreement.
**❸** *Incorporates probabilities* – The metric should use the probability distribution over predictions provided by the models.

| Metric | Adjusts for Accuracy | Distinguishes different mistakes | Incorporates Probabilities |
|---|:---:|:---:|:---:|
| %Flips $= 1 - c_{\text{obs}}$ (Dutta et al., 2024) | ✗ | ✗ | ✗ |
| Cohen's $\kappa$, Scott's $\pi$, Fleiss $\kappa$ | ✗ | ✓ | ✗ |
| %Agreement (Zheng et al., 2023) | ✗ | ✓ | ✗ |
| Error Consistency (Geirhos et al., 2020) | ✓ | ✗ | ✗ |
| Pearson / Matthew's Correlation of Errors | ✓ | ✗ | ✗ |
| Divergence metrics like KL, JSD | ✗ | ✓ | ✓ |
| CAPA (Ours) | ✓ | ✓ | ✓ |

cess agreement observed ($c_{\text{obs}} - c_{\text{exp}}$) from what is possible beyond chance ($1 - c_{\text{exp}}$):

$$k = \frac{c_{\text{obs}} - c_{\text{exp}}}{1 - c_{\text{exp}}}, \tag{1}$$

### 2.2. Our Contribution

We identify two key limitations of error consistency ($k$):

**Does not distinguish differing mistakes (❷):** If two models make wrong but different predictions, error consistency still counts that as an agreement. For example, two models that are always wrong, even in different ways, have perfect error consistency ($k = 1$). It thus overestimates similarity.

**Does not capture probability information (❸):** For comparison to humans, error consistency assumes a single top prediction, whereas models inherently output a probability distribution. Ignoring these probabilities can lead to incorrect conclusions about model similarity. Consider two models whose outputs are $[0.49, 0.51]$ and $[0.51, 0.49]$. Despite their small differences, binary labels would classify them as making entirely different predictions (0 and 1). Conversely, models with predictions $[0.99, 0.01]$ and $[0.51, 0.49]$ may share the same binary output (0 and 0) but differ significantly in confidence distribution.

**Novel Metric.** We redefine $c_{\text{obs}}$ and $c_{\text{exp}}$ to address the above limitations. For clarity we adjust the notation of our agreement metrics to $c_{\text{obs}}^p$ and $c_{\text{exp}}^p$. To compute $c_{\text{obs}}^p$ we directly use the model output probabilities (Eq.2), thus accounting for disagreement on incorrect options and better capturing model similarity. This approach lets us calculate $c_{\text{obs}}^p$ without sample-wise ground-truth annotations. For $c_{\text{exp}}^p$, we take into account that the model output predictions can span over multiple options rather than only looking at sample-wise accuracy.

**Definition.** We define $\kappa_p$ in the context of Multiple Choice Questions (MCQs), which is the format of many popular

benchmarks for LMs. We provide a detailed derivation in Appendix A.1, with extensions to classification and exact match evaluations in Appendix A.3.

**Observed Agreement** ($c_{\text{obs}}^p$): It represents the probability of agreement if the model's predictions were sampled based on the *observed* likelihoods assigned over options. Formally,

$$c_{\text{obs}}^p = \frac{1}{|D|} \sum_{x \in D} \sum_{o_i \in O(x)} p_1(o_i) \cdot p_2(o_i), \tag{2}$$

where $p_1(o_i)$ and $p_2(o_i)$ are the output probabilities for model 1 and 2, respectively, on a data sample $x$ for option $o_i$. $O(x)$ are the possible options: $O(x) = [o_i, \ldots, o_n]$, and $|D|$ is the total number of data points.

**Chance Agreement** ($c_{\text{exp}}^p$): To account for higher accuracies inflating $c_{\text{obs}}^p$, we normalize by the agreement expected from two *independent* models. First, we define $\overline{p_j}$ as the average probability model $j$ assigns to the correct option across all samples. For a perfectly calibrated model $\overline{p_j}$ approaches accuracy, thus aligning with the motivations in error consistency. Then, we define independent models as assigning $\overline{p_j}$ probability to the correct option, and uniformly distributing the remaining $1 - \overline{p_j}$ probability over the incorrect options. The latter is necessary, as there is no coherent concept of "classes" for MCQ data, i.e. the options can be permuted. This prevents us from computing class marginals for the remaining options, such as in inter-annotator agreement metrics like Cohen's Kappa, Scott's Pi (Scott, 1955), Fleiss' Kappa (Fleiss et al., 1981). Formally,

$$c_{exp}^p = \underbrace{\overline{p_1} \cdot \overline{p_2}}_{\text{chance agreement on correct option}} + \\ \underbrace{(1 - \overline{p_1}) \cdot (1 - \overline{p_2}) \cdot \frac{1}{|D|} \sum_{x \in D} \frac{1}{|O(x)| - 1}}_{\text{chance agreement on incorrect option}} \tag{3}$$

where $|O(x)|$ is the number of options in question $x$.

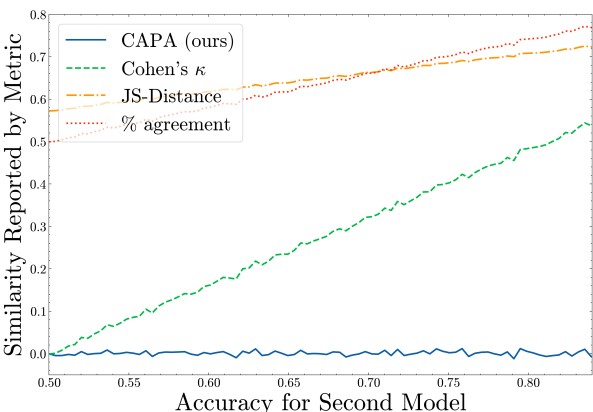

*Figure 2.* **Metric comparison for independent models with uncorrelated predictions.** In this simulation, for each model we select an independent random subset of samples as correct, with the first having a fixed 90% accuracy, while for the second accuracy is varied from 50% to 90%. CAPA correctly reports 0 similarity when models have uncorrelated errors.

Finally, the equation for CAPA is:

$$\kappa_p = \frac{c_{\text{obs}}^p - c_{\text{exp}}^p}{1 - c_{\text{exp}}^p} \qquad (4)$$

**Interpretation.** We prove $\kappa_p$ is bounded between $-1$ and $1$ in Appendix A.6. A value of 0 means the models have the same agreement as independent models given their accuracies. A negative value means the models disagree, and a positive value indicates they agree beyond independent models with their accuracy. *As $\kappa_p$ increases, it means models make more similar mistakes, their errors become more correlated, and they are functionally less different*. We use these interpretations interchangably.

**Alternatives and Justification.** We summarize comparisons to existing functional similarity measures based on key desiderata (❶-❸) in Table 1. To demonstrate the empirical effects, we perform a simulation, where we take two binary classification models, that are correct on an independent random subset of samples (more details in Appendix A.4). Since these models are independent, any agreement is by chance, and their predictions are entirely uncorrelated. In such a case, we want a metric that reports 0 similarity. In Figure 2 we report metric behavior when one model has a fixed 90% accuracy, while varying the accuracy of the other model from 50% to 90%. Our proposed metric, CAPA, indeed reports 0 similarity, indicating that the models are independent and their predictions uncorrelated. This is in contrast to popular metrics like Cohen's $\kappa$ (Cohen, 1960), used to measure agreement between two annotators, Jensen Shannon (JS) Distance, used to measure a normalized divergence between two probability distributions, and %Agreement, which is the number of same predictions made by the two models. All three alternative

metrics do not adjust for chance agreement due to higher accuracy, and thus lead to higher similarity scores being reported as the second model's accuracy increases, even though the two models continue to make independent and uncorrelated predictions. In Appendix A.1 we prove that CAPA is a generalization of error consistency and reduces to it for binary classification tasks if we discretize predictions to 1 on the correct option and 0 on others. As such, in the simulation presented here, error consistency would be quite similar to CAPA, as both metrics adjust for accuracy. In Appendix A.7 we show that CAPA demonstrates the expected trend most clearly also in settings when models have correlated predictions (Appendix Figure 8-9). Lastly, CAPA can be extended beyond pairwise comparisons to multiple models (Appendix A.2), and for completeness, we present probabilistic extensions for Cohen's $\kappa$, Scott's $\pi$, Fleiss' $\kappa_F$ in Appendix A.5.

## 3. Affinity Bias in LLM-as-a-Judge

Evaluating free-response model generations automatically at scale is tricky (Biderman et al., 2024). This has led to popular leaderboards like Arena-hard-auto (Li et al., 2024b), AlpacaEval 2.0 (Dubois et al., 2024), and Aidan-Bench (McLaughlin et al., 2024) adopting LLM-as-a-Judge for scoring. Recent work cautions that language models are biased towards their own outputs (Liu et al., 2024; Panickssery et al., 2024), and these leaderboards assume that excluding the judge model from the rankings circumvents this problem. However, human interviewers have been shown to also be biased towards candidates with similar knowledge and traits, a phenomenon called *affinity bias* (Bagues & Perez-Villadoniga, 2012). We study whether LMs also exhibit a bias toward similar models. This would indicate that it is not sufficient to just use a held-out LM as the judge; one must account for the confounder of similarity.

### 3.1. Experimental Setup

To study whether LM judges prefer more similar models, we evaluate a large set of judges and models on MMLU-Pro (Wang et al., 2024), a benchmark for hard problem solving questions across 14 domains. We filter 8,707 questions that can also be answered in a free-text response style, without options, following Myrzakhan et al. (2024). Each question is posed to every evaluated model as an MCQ and as an open-style question. The per-sample results of the former are used to compute the similarities of judge-model pairs, whereas responses to the latter are given to an LLM-as-a-judge. The judge has to make a binary decision on whether a free-text response is correct or wrong. This is done based on its own internal knowledge without access to a ground-truth solution. We call the average of binary judgments across the benchmark the model's *Judgment Score* for a given judge. Using a parallel MCQ evaluation with

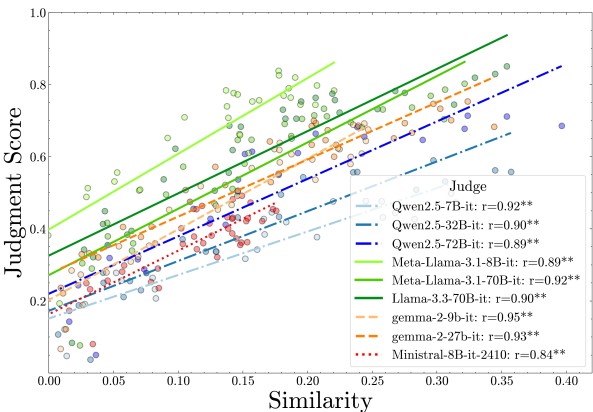

*Figure 3.* **Judgment Score Relation with Model Similarity on only across family pairs.** Each line is a regression model fit between judgment and similarity scores. The circle shape indicates that only across-family judge-model pairs are plotted. We report for each fit the corresponding Pearson correlation values, $r$. We found significant positive correlation between judgment scores and CAPA across all judges, $**$ indicates $p < 0.01$.

ground-truth answers allows us to compare the judgment scores with verifiable accuracy measurements (details and comparisons with alternatives are in Appendix B.3), consistent with prior scalable oversight research (Bowman et al., 2022). We compute pairwise similarity with CAPA, $\kappa_p$, across 9 judge and 39 model pairs. For a complete overview of models investigated, question filtering, inference setup, and prompts used for judges, see Appendix B.

### 3.2. Results & Discussion

**Q1: Do LM Judges favor more similar models?** As a motivating example, `Qwen2.5-72B-Instruct` as a judge scores `Qwen2.5-7B-Instruct` as being 71% correct, while the more capable (41% vs 51% MCQ accuracy) `Llama-3.1-70B-Instruct` is deemed less accurate at 67%. In Figure 3 we show that model favoritism extends beyond *self-* or *family-* preference to *all* models that are functionally similar. We find a significant ($p < 0.01$) positive correlation (average Pearson r=0.84) between LLM-as-a-judge scores and model similarity ($\kappa_p$) for all judges. To ensure this trend is not only an artifact caused by aggregating across all MMLU-Pro categories, per-category results for the same experiment are shown in the Appendix B.5. These exhibit the same affinity bias observed here for all different topics.

**Q2: Is this merely due to better accuracy?** Note that while $\kappa_p$ adjusts for inflation in agreement of highly accurate models, we do expect models with lower accuracy to be less similar with highly capable judge models, and thus have lower $\kappa_p$. To control for the accuracy of the evaluated model we perform multiple regression and partial correlation analysis (Table 2). The multiple regression analysis

*Table 2.* **Partial Correlation and Multiple Regression Results.** The table reports partial correlation results between judgment scores and model similarity when controlling for accuracy (r - Pearson correlation), as well as multiple regression results between judgment scores (DV) $\sim$ similarity (IV) and accuracy (IV). $*$ and $**$ indicate significance level p<0.05 and p<0.01 respectively. Across all judges we find a significant partial correlation, which implies that after controlling for accuracy there remains a relationship between judge score and model similarity. With respect to Multiple regression, across all judges we find a significant effect of both IV on judgment scores while holding the other constant, suggesting a strong positive relationship (for details, see Appendix B.6).

| Judge | Partial Cor. | Multiple Reg. | |
|---|---|---|---|
| | $r$ | sim | acc |
| Qwen2.5-7B-It | 0.60** | 0.59** | 0.51** |
| Qwen2.5-32B-It | 0.43** | 0.41** | 0.86** |
| Qwen2.5-72B-It | 0.42** | 0.47** | 1.04** |
| Meta-Llama-3.1-8B-It | 0.65** | 1.15** | 0.53** |
| Meta-Llama-3.1-70B-It | 0.45** | 0.61** | 0.92** |
| Llama-3.3-70B-It | 0.35* | 0.50* | 1.02** |
| gemma-2-9b-It | 0.65** | 0.76** | 0.69** |
| gemma-2-27b-It | 0.65** | 0.71** | 0.68** |
| Ministral-8B-It-2410 | 0.60** | 0.82** | 0.43** |

shows that both, accuracy and similarity, have a significant positive effect on the judge score. The coefficient of accuracy increases for more capable judge models, consistent with prior work showing improved alignment with human judges (Thakur et al., 2024). We find that especially for small models (<32B) the effect of similarity is greater. The partial correlation results control for accuracy and confirm that there is still a significant effect of similarity on judgment scores even for the best judge models. Altogether, the statistical analysis confirms that judgment scores are confounded by affinity bias. For extended results also including model size as a confounder please see Appendix B.1.

**Q3: Does this transfer to LLM-as-a-judge for binary preference in free-form generation?** In most cases, LLM-as-a-judge is not used to check the technical correctness of responses, but rather to match human preference by selecting the better one from a pair of responses. To extend our findings to the additional domain of chat responses, we provide an additional experiment on the AlpacaEval benchmark in Appendix B.4 (Dubois et al., 2023). We show that the Elo obtained by using the same judges as before to obtain binary preferences still strongly correlates with model similarity on MMLU-Pro. This indicates the existence of an affinity bias in settings other than QA.

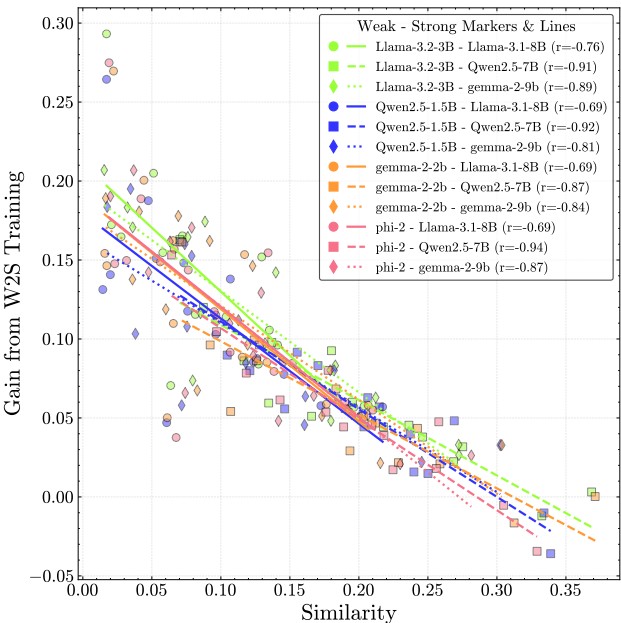

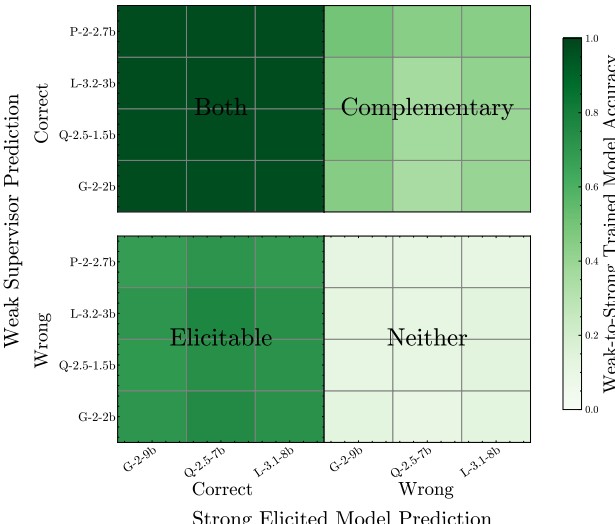

*Figure 5.* **Role of Complementary Knowledge and Elicitation in Weak-to-Strong Generalization**. We decompose the accuracy of the weak-to-strong trained model on four parts of the test data distribution, based on the correctness of the weak supervisor and an oracle strong elicited model which uses ground-truth annotations. Sub-rectangles represent weak, strong model pairs. Results are averaged across 15 tasks. Complementary knowledge transfer explains weak-to-strong model accuracy beyond elicitation.

*Figure 4.* **Similarity vs Gain from Weak-to-Strong Training.** Across 12 model pairs, the strong student gains more from weak-to-strong training on tasks where it is more different from the weak supervisor ($p < 0.01$).

## 4. Learning from Complementary Knowledge of LM Annotators

We now study the role of similarity in AI supervising training. This can allow scaling up training data by reducing reliance on costly and time-intensive expert human inputs. There is hope that models can learn from other models to improve further even on tasks where they surpass human capabilities (Hughes et al., 2024). Where could this improvement come from? We hypothesize that the existence of complementary knowledge or complementary capabilities between two LMs can be one mechanism for driving improvements from LM annotations, if effectively leveraged. This complement can exhibit itself in the form of differing predictions on training data, and can thus be quantified using functional similarity between the supervisor and student model. Lower $\kappa_p$ is indicative of more complementary knowledge, and as an implication of our hypothesis, should inversely correlate with the performance gain of a student model when trained on the supervisor's annotations.

### 4.1. Experimental Setup

Burns et al. (2024) study training a larger student model on annotations from a small task-finetuned "expert" teacher. They find the student can outperform the supervisor, a phenomenon they call "weak to strong generalization". We study this setup as it can seem counter-intuitive when viewed from the lens of accuracies. How can training a 60% ac-

curacy student on a 70% accuracy task-finetuned teacher lead to 75% accuracy? We adopt a lens of complementary knowledge to understand weak-to-strong generalization.

Following Burns et al. (2024), we define "weak" and "strong" based on model size. We study 4 weak models in the $1 - 3B$ parameter range, and 3 strong models in the $7 - 9B$ parameter range, for a total of 12 model pairs, and 15 of the binary classification NLP tasks studied in Burns et al. (2024), specified in Table 9. The smaller models do have lower accuracy (before finetuning) than the bigger ones across tasks, as shown in Appendix Figure 17, justifying calling them "weak" and "strong". We measure similarity between the weak supervisor and base student model on the validation set. We then perform weak-to-strong training on the student model, using the confidence-weighted finetuning objective proposed in Burns et al. (2024). We investigate if similarity is an apriori predictor of performance gained on the test set. The full setup is consistent with the open-weight model reproduction by (Scherlis et al., 2024), and is described in Appendix C.1.

### 4.2. Results & Discussion

**Q1: Does Complementary Knowledge Influence Performance Gain?** Figure 4 shows that for all model combinations, similarity between the weak supervisor and initial strong student inversely correlates with the improvement obtained from weak-to-strong training ($r = -0.85$). Even

*Table 3.* **Accuracy gains possible from weak-to-strong training.**
We average accuracies across 15 datasets and 12 model pairs (180
training runs) and report gaps to the student model's initial accuracy. Complementary knowledge transfer can enable higher gains
than the previously considered ceiling estimate from elicitation.

| Model | Accuracy Gap |
|---|---|
| Initial Strong Student | 75.1% |
| Weak Supervisor | +4.1 |
| Weak to Strong Trained Student | +7.4 |
| **Ceiling Estimate** | |
| Ground-truth Elicitation (previous) | +11.2 |
| Elicitation $\cup$ Complementary (ours) | +14.1 |

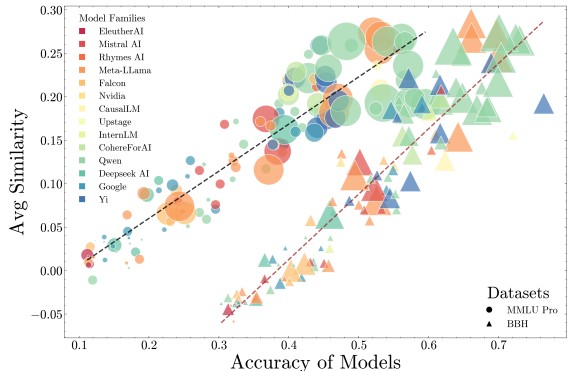

*Figure 6.* **Average Similarity ($\kappa_p$) vs Model Capability**. We split
130 LMs into 5 buckets based on their accuracy percentile. For
each LM we compute its mean similarity within the bucket (across
models from different developers), and plot it against model accuracy. The size of the scatter points indicates model size. As $\kappa_p$
measures overlap in mistakes, the positive correlation indicates LM
mistakes are getting more correlated with increasing capabilities.

after using partial correlation analysis to control for the accuracy gap between the weak supervisor and strong student,
similarity is inversely correlated with weak-to-strong gain
($r = -0.35, p < 0.01$). Thus, tasks where the supervisor
and student make less correlated errors tend to yield greater
improvements. This contributes towards understanding why
gains from weak to strong training vary across tasks, an
open question posed by Burns et al. (2024). The observation
is not specific to our similarity metric, Appendix Figure 16
shows the same trend holds with alternate metrics, albeit
with lesser variance explained.

**Q2. Does Complementary Knowledge Add Beyond Elicitation?** The original explanation for performance gains
from weak-to-strong generalization is that the weak supervisor "elicits" the latent knowledge in the superior representations of the stronger student (Burns et al., 2024). To
investigate whether complementary knowledge adds to this
explanation or is subsumed within it, we first obtain the
strong model with "upper-bound" elicitation, by finetuning it on ground-truth annotations. We refer to this as the
*strong elicited* model. We can then separate the test data
into four parts based on whether the strong elicited and
weak supervisor model were correct or wrong, measuring
average accuracy of the weak-to-strong model on each part
to disentangle gains from different factors. The experiment
setup is discussed further in Appendix C.2.

Figure 5 reports aggregate values across 15 tasks for
12 model pairs. Accuracy on the bottom-left quadrant
(avg. 71.9%) can only be due to successful elicitation,
as here the weak supervisor was wrong. Accuracy on
the top-right quadrant (avg. 42.2%) can only be due to
complementary knowledge transfer as here the upper-bound
elicitation model was wrong. This confirms that elicitation
plays an important role in weak-to-strong generalization,
with complementary knowledge transfer from the weak
supervisor also contributing to significant gains.

**Q3. Where can weak-to-strong training improve?** The
strong elicited model is considered to represent upper-bound
performance. However, recent work has shown that the performance of the strong elicited model can be surpassed (Shi
et al., 2025), indicating it is not really an "upper-bound".
Indeed, the strong elicited model's performance does not
account for potential gains from leveraging the complementary knowledge of the weak supervisor. We compute a new
ceiling, taking the union of correct predictions between the
weak supervisor and strong elicited model, which is significantly higher as shown in Table 3. Interestingly, on the
training set, the weak-to-strong trained model shows similar accuracy on the top-left and bottom-right quadrants as
shown in Figure 15. Yet, when generalizing to unseen samples, it falls back more often to its initial priors. We hope this
analysis guides future work on improving weak-to-strong
training methodology, by highlighting leveraging complementary knowledge as a concrete avenue for improvement.

## 5. Models are making more similar mistakes as capabilities increase

The previous two sections highlighted two major advantages
of having access to more diverse LMs: a) it leads to less
biased judges, b) it can drive more performance gains from
training on LM annotations. This points to the importance
of diversity, or lower model similarity, for AI oversight. As
AI oversight becomes increasingly relevant with advancing
capabilities, we now study similarity trends in existing LMs
across different levels of capability. It has been shown
model representations across modalities are converging with
increasing capabilities (Huh et al., 2024). Does this also
lead to more similar mistakes?

## 5.1. Experimental Setup

We collect sample-wise evaluation files for 130 official models from the OpenLLM Leaderboard 2 released by Hugging-Face, listed in Appendix D.5. We use MMLU-Pro (Wang et al., 2024) and Big Bench Hard (BBH) (Suzgun et al., 2023) as they measure a broad range of capabilities using MCQ, and frontier models have reasonable accuracies while not saturating these datasets. We first bucket these models into five performance percentile ranges. Then, for each model, we compute its mean similarity ($\kappa_p$) with models in the same bucket from different developers, to prevent confounding from distillation or continual training. More setup details are provided in Appendix D.1. In Appendix D.3 we also report pairwise results, and using the extension of $\kappa_p$ for sets of $M > 2$ models.

## 5.2. Results & Discussion

**Q1. Are model errors becoming more correlated with improving capabilities?** Figure 6 shows a strong positive correlation between model capabilities and $\kappa_p$, which measures similarity beyond chance agreement due to accuracy. In Appendix D.4 we find this also holds across individual categories in both datasets, not just in aggregate.

**Potential Implications.** If this trend continues, it could mean greater affinity bias when using LM judges, and lower potential for gains from inter-LM training in the context of our earlier results. It could undermine benefits from using LM juries by compromising independence and amplifying collective biases. Most concerningly, our results indicate that as model blind-spots get harder to detect, making us defer more to AI oversight, models also make more similar mistakes, posing safety risks from correlated failures.

**Q2. Why are model errors becoming more correlated?** This is an interesting research direction in itself. We perform a preliminary analysis in Appendix D.2, summarizing key conclusions here. First, we observe only a slight increase in similarity for harder questions in our datasets, indicating difficulty is not a significant confounder for this trend. We find this trend is stronger in instruction-tuned models, and using alternative architectures like Mamba (Gu & Dao, 2024) may not be enough to increase diversity.

## 6. Related Work

There is increasing interest in finding differences between models for applications like visual tools for comparative analytics (Strobelt et al., 2021; Kahng et al., 2024), efficient human evaluation (Boubdir et al., 2023), comparing learning algorithms (Shah et al., 2023), identifying side-effects of API updates (Eyuboglu et al., 2024) or quantization (Dutta et al., 2024). Prior work has also looked at qualitatively describing differences between data distributions (Zhong

et al., 2022; 2023; Dunlap et al., 2024; 2025). Our work proposes metrics to quantify LM differences (or similarity). Huh et al. (2024) used representation similarity metrics (Kornblith et al., 2019; Bansal et al., 2021) to show convergence in visual representations and their alignment with language representations. In contrast, we show model mistakes are becoming more correlated as capabilities improve, using sample level evaluations (Burnell et al., 2023) such as those available on OpenLLMLeaderboard (Myrzakhan et al., 2024) and HELM (Bommasani et al., 2023). Geirhos et al. (2020) proposed measuring "error consistency" between image classifiers and humans, with Geirhos et al. (2021) showing an early trend of data-rich models making more similar mistakes to humans. We enrich this metric, distinguishing between different mistakes and incorporating probabilistic information. Bommasani et al. (2023) used a similar metric to demonstrate homogenization in outcomes when using LMs for decision making. Our work shows that such risks from algorithmic monoculture (Kleinberg & Raghavan, 2021) are increasingly relevant as LM capabilities improve.

Our results on AI judges fall in a broader line highlighting their pitfalls (Zheng et al., 2025). These include biases such as favoring verbose texts or options at certain positions (Koo et al., 2024; Ye et al., 2025). Interestingly, these biases are also sometimes found in human annotators (Chen et al., 2024). In fact, there is rich literature documenting biases in human judgements of other humans. One such bias is affinity bias, where recruiters prefer candidates with similar knowledge and skills as them (Bagues & Perez-Villadoniga, 2012). We show LM judges also systematically favor other models that make similar mistakes, generalizing previous results that showed LMs favor their own outputs (Liu et al., 2024; Panickssery et al., 2024). Overall, we believe AI evaluators should be accompanied with formal checks like consistency (Fluri et al., 2024).

A second aspect of AI oversight is using another model's supervision to train better models. This is similar to training on text generated by an LM (Chiang et al., 2023) with ongoing debates about its benefits (Kazdan et al., 2025), and an emerging paradigm of exploiting a gap in difficulty between solution generation and evaluation (Song et al., 2025). In this paper, we study the more established setup of training LMs on LM annotations, where Burns et al. (2024) demonstrated the phenomenon of weak to strong generalization, and it has been leveraged for other applications like image classification (Guo et al., 2024) and aligning models (Zhu et al., 2025). Prior work has attempted to understand weak to strong generalization, notably using "misfit error" (Charikar et al., 2024), which shows that the student's disagreement with the weak supervisor *after* weak to strong training correlates with its accuracy gap from the weak supervisor. Instead, we show similarity between the

weak supervisor and strong student can *apriori* predict gains from weak-to-strong training. The benefit of model diversity has previously been discussed in related settings like knowledge distillation for image classifiers (Roth et al., 2024) and training chess models that outperform the humans they are trained on (Zhang et al., 2024).

## 7. Conclusion, Limitations, Future Work

Our paper shows the importance of measuring functional similarity for language models. We derive a novel, probabilistic metric for model similarity, CAPA ($\kappa_p$). We then use it to study the implications of similarity for AI oversight – showing affinity bias in AI judges, and the role of complementary knowledge when training on LM annotations, such as in weak-to-strong generalization. AI oversight will become more relevant as capabilities improve, so our finding that increasing capabilities could lead to more correlated errors is particularly concerning. Thus, we believe measuring and accounting for model similarity is going to be increasingly important. We now list some limitations of our work and avenues for future work, that can help develop a better understanding of model similarity and its implications.

**Establishing Causation**: We established similarity correlates with both aspects of AI oversight – evaluation and training supervision. To establish causality, we need methods to make a model less similar without harming capabilities, which is itself a challenging open problem.

**Extending to similarity metrics for free-text outputs**: Everyday use of generative models is based on their free-text responses. Like much work on benchmarking, we had to limit to MCQ tasks as the science of precisely evaluating free-text is still evolving (Biderman et al., 2024). For example, both model-free (Papineni et al., 2002) and model-based metrics (Pillutla et al., 2021) suffer from a wide range of syntax and style sensitivity issues (Kocmi et al., 2021; He et al., 2023). We hope the community takes up the challenge of designing similarity metrics for free-response text and reasoning. This would allow studying the role of similarity using more promising oversight setups like debate (Kenton et al., 2024) and process supervision (Lightman et al., 2024).

**Generator-Verifier gap**: AI oversight has recently shown promise for tasks where it is easier to validate a solution than generate it (Song et al., 2025). Similarity may continue to play a role here. (1) In evaluations, similarity in stylistic preferences of the generated solution may influence judge scores. (2) In training, the generator-verifier gap may be larger if models are more different.

**Safety implications**: Researchers separately develop many post-training interventions to reduce harmfulness, dual-use knowledge, dishonesty etc. In real world model deployments, all these problems have to be tackled at once, which can benefit from composing interventions (Kolbeinsson et al., 2025). If the benefits are decorrelated, composing would lead to greater compound safety. If the side effects are correlated, composing would lead to lower accumulation. More broadly, measuring LM similarity post-intervention can help characterize decorrelation in research bets, for granters (Canton, 2025). For example, LM unlearning was recently found to be functionally similar to refusal training (Łucki et al., 2025), even though it was proposed as a complementary safeguard (Li et al., 2024a). Finally, as we transition towards language agents, similarity can help understand collective "blind spots" (He et al., 2023), and could lead to better cooperation (Lowe et al., 2017) but also scheming (Balesni et al., 2024) between multiple agents.

**Qualitative analysis of model differences**: We developed quantitative methods for measuring LM similarity on a given data distribution. One exciting direction is to use these metrics to provide qualitative difference descriptors (Dunlap et al., 2025) between models, by describing data distributions where models are least similar.

## Acknowledgments

The authors would like to thank (in alphabetical order) Arvindh Arun, Nikhil Chandak, Thomas Klein, Ankit Sonthalia, Guinan Su, Mark Tygert, Vishaal Udandarao for helpful feedback. We thank HuggingFace for the public sample-wise predictions provided in OpenLLMLeaderboard, which enabled our work. This work was supported by the Tübingen AI Center. JS thanks the International Max Planck Research School for Intelligent Systems (IMPRS-IS) for support. AP and MB acknowledge financial support by the Federal Ministry of Education and Research (BMBF), FKZ: 011524085B and Open Philanthropy Foundation funded by the Good Ventures Foundation.

## Author Contributions

Shashwat conceived the project, proposed the CAPA metric, and led the LM Annotators experiments (Section 4). Joschka led the LLM-as-a-Judge experiments (Section 3). Ilze led statistical analysis of all results, and characterized properties of CAPA (Section 2, Appendix A). Karuna analyzed trends for capabilities-similarity (Section 5). Ameya helped across sections. The manuscript was written by Shashwat, Ilze, Ameya and Joschka. Douwe, Matthias and PK provided feedback and advice throughout the project. Jonas advised the design of all experiments.

## Impact Statement

Our paper can be considered as foundational research analyzing mechanisms of AI oversight and identifying an important shortcoming, and not tied to particular applications.

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

# Appendix

## Contents

# A. Metrics

The following section covers design details for CAPA $\kappa_p$. Firstly, we address derivation of CAPA in Section A.1 and its theoretical bounds in Section A.6. Secondly, we explain how to extend CAPA to multi-model set-up (Section A.2) and how to adapt CAPA to classification and exact match settings (Section A.3). Lastly, we introduce probabilistic versions of popular agreement metrics (Section A.5) and provide a comparison between them and CAPA (Section A.4).

## A.1. Derivation of CAPA

CAPA is intended to be used as a similarity metric in the context of model accuracies. As such, it extends Error consistency (Geirhos et al., 2020), a metric that adjusts chance agreement by taking into account model accuracy. In particular, the same formula is used to define Cohen's $\kappa$, Scott's $\pi$, Fleiss' $\kappa$, Error Consistency and CAPA:

$$\frac{\text{observed agreement} - \text{chance agreement}}{\text{maximum possible agreement - chance agreement}}, \tag{5}$$

where the excess agreement is subtracted both from the numerator and denominator, essentially calculating the proportion of possible excess agreement that is observed between the two models. Across all metrics, the maximum possible agreement is 1. Where CAPA differs from the existing metrics is how we calculate the observed and change agreement.

We redefine *error consistency* (Geirhos et al., 2020) by incorporating probabilistic information. To achieve this we introduce a probabilistic computation of the observed agreement, $c_{\text{obs}}$ as $c_{\text{obs}}^p$, and the chance agreement, $c_{\text{exp}}$ as $c_{\text{exp}}^p$. The new equation becomes:

$$\kappa_p = \frac{c_{\text{obs}}^p - c_{\text{exp}}^p}{1 - c_{\text{exp}}^p}. \tag{6}$$

**Observed agreement $c_{\text{obs}}^p$** : Given that we have the predicted output probabilities, $p_1(o_*)_x$, by a LM for all possible options, $O_x = [o_1, \ldots, o_N]$, for a data sample $x$, e.g. $p_1(o_*)_x \forall o_* \in O_x$, we can compute the relative observed overlap as:

$$c_{\text{obs}}^p = \frac{1}{|D|} \sum_{x=1}^{D} \sum_{i=1}^{O} p_1(o_i)_x \cdot p_2(o_i)_x \tag{7}$$

where $p_1(\cdot)$ is the predicted probability by model 1 and $p_2(\cdot)$ is the predicted probability by model 2. We would like to highlight that the above calculation is performed on **sample level** to avoid confusion with the common chance agreement $p_e$ calculation in Cohen's kappa [2].

**Agreement by chance $c_{\text{exp}}^p$** To estimate the model chance agreement $c_{\text{exp}}^p$ we first start by computing the average probability that a given model is correct $\overline{p_*}$:

$$\overline{p_*} = \frac{1}{|D|} \sum_{x=1}^{D} \sum_{i=1}^{O} \mathbb{I}[o_i = \text{gt}] p_*(o_i)_x \text{ where gt} = \text{ground truth} \tag{8}$$

Performing the above calculation per model accounts for the possibility that each model may have different marginal distributions. An assumption that is fair to assume in the context of LMs. Subsequently, given the $\overline{p_*}$ per model we can compute the probability that two models are **correct** by chance as: $\overline{p_1} \cdot \overline{p_2}$. Conversely, to account for model chance disagreement we (1) group all the remaining options as **incorrect** and (2) adjust for the number of options: $\frac{1}{|D|} \sum_{x=1}^{D} \frac{1}{|O_x|-1}(1 - \overline{p_1})(1 - \overline{p_2})$. These steps are necessary because (1) MCQ options can be permuted, therefore, class marginal probabilities cannot be computed, and (2) the chance disagreement without adjusting for the number of options overestimates the agreement by chance:

$$0 < \frac{1}{|D|} \sum_{x=1}^{D} \frac{1}{|O_x|-1}(1 - \overline{p_1})(1 - \overline{p_2}) \leq (1 - \overline{p_1})(1 - \overline{p_2}) \tag{9}$$

---

[2]Cohen's kappa uses the marginal probabilities across categories to estimate $p_e$. However, in MCQ there are no 'class categories' as the options can be permuted across data samples. Therefore, marginal probabilities cannot be estimated.

In particular, if the number of options is ignored then the underlying assumption is that both models put their 'incorrect' probability mass on the same option, following a Dirac delta $\delta(o_*)$ distribution. This is a very strong assumption, that overestimates model error agreement. Therefore, we propose to adjust this by assuming that the distribution for the incorrect options follows a uniform distribution $\mathbf{U}\{o_1, o_{n-1}\}$ as adjusted by our normalizing factor $\frac{1}{|D|}\sum_{x=1}^{D}\frac{1}{|O_x|-1}$, where $|O_x|$ is the total number of options for a sample $x$. As such, the overall agreement by chance probability is:

$$c_{\exp}^{p} = \underbrace{\overline{p_1 p_2}}_{\text{chance agreement correct}} + \frac{1}{|D|}\underbrace{\sum_{x=1}^{D}}_{\text{mean}} \underbrace{\frac{1}{|O_x|-1}}_{\text{uniformity assumption}} \underbrace{(1-\overline{p_1})(1-\overline{p_2})}_{\text{chance agreement incorrect}} \tag{10}$$

Moreover, for perfectly calibrated models the mean correct probability $\overline{p_*}$ would approach model accuracy, $\overline{p_*} \to \hat{p_*}$ and is upper bounded by it $\overline{p_*} < \hat{p_*}$ as $\overline{p_*}$ is computed based on probabilities ($\hat{p_*} = \frac{TP+TN}{|D|}$).

**Reduction of CAPA to Error Consistency for binary classification** In binary classification setting when the underlying probabilities are unavailable CAPA reduces to error consistency, as (1) $c_{\text{obs}}^{p} = \frac{1}{|D|}\sum_{x=1}^{D}\mathbb{I}[\arg\max p_1 = \arg\max p_2] = c_{\text{obs}}$, and (2) $c_{\text{obs}}^{p} = c_{\text{obs}}$ as $\overline{p_*} = acc_*$, and the normalizing factor simplifies to 1.

## A.2. Extending CAPA to more than two models

In Section 2.2, we computed functional similarity between a pair of models. Here, we extend CAPA to multi-model comparisons. In the inter-annotator agreement literature, Fleiss' $\kappa$ (Fleiss et al., 1981) is commonly used for this. However, it is ill suited to our modeling paradigm as it defines $c_{\exp}^{p}$ using the assumptions of Scott's $\pi$ instead of Cohen's $\kappa$ (this is problematic when measuring model similarity as discussed in the previous section). We derive CAPA for more than two models using first principles logic, similar to how Fleiss' $\kappa$ was derived.

Suppose the number of models is $M > 2$. We still use the $\frac{c_{\text{obs}}^{p} - c_{\exp}^{p}}{1 - c_{\exp}^{p}}$ formula, but change the definition of $c_{\text{obs}}^{p}$ and $c_{\exp}^{p}$. For $c_{\text{obs}}^{p}$, Fleiss' $\kappa$ measures the proportion of observed pairwise agreements from the total possible for each question, averaging across questions. This is equivalent to averaging the observed agreements for each pair of models when all models annotate all questions, which is true in our case[3]. This gives us $c_{\text{obs}}^{p} = \frac{2}{M(M-1)}\sum_{1\le i<j\le M}\frac{1}{|D|}\sum_{x=1}^{D}\sum_{k=1}^{O}p_i(o_k)_x \cdot p_j(o_k)_x$.

Second, for $c_{\exp}^{p}$, Fleiss' $\kappa$ measures the expected pairwise agreements if all $M$ models were independent. This can be obtained by averaging the $c_{\exp}^{p}$ for two models across all possible pairs of $M$ models. This gives us

$$c_{\exp}^{p} = \frac{2}{M(M-1)}\sum_{1\le i<j\le M}\left(\overline{p_i}\cdot\overline{p_j} + (1-\overline{p_i})\cdot(1-\overline{p_j})\cdot\left(\frac{1}{|D|}\sum_{x=1}^{D}\frac{1}{|O_x|-1}\right)\right)$$

.

## A.3. How to use CAPA beyond multiple choice?

In Section 2.2 we defined CAPA for MCQs as this is used throughout the paper, and more commonly for language models. For completeness, we now define CAPA for classification settings and exact match settings, which are alternate strategies for evaluating models.

**Classification**: Unlike MCQs, in this setting are coherent classes (categories), representing nominal data. The model output now is a probability distribution over $C$ classes. Therefore, we compute $c_{\text{obs}}^{p}$ across categories as follows:

$$c_{\text{obs}}^{p} = \frac{1}{|D|}\sum_{x\in D}\sum_{c_i\in C(x)}p_1(c_i)\cdot p_2(c_i), \tag{11}$$

where $p_*(c_i)$ is the output probability for class $c_i$. For the computation of $c_{\exp}^{p}$ we follow the same definition as in the main paper, but now $\overline{p_j}$ is computed for the correct class and the chance agreement on the incorrect class is adjusted by the number

---

[3]Fleiss $\kappa$ is also defined when not all annotators respond to every question, as long as the number of respondents per question is fixed.

| Metric | Formula | Description |
|---|---|---|
| Cohen's Kappa | $\kappa = \frac{P_o - P_e}{1 - P_e}$ | Measures inter-rater reliability $P_0$ while accounting for chance agreement $P_e$. |
| Scott's Pi | $\pi = \frac{P_o - P_e}{1 - P_e}$ | Similar to Kappa, but uses marginal probabilities for $P_e$. |
| Error Consistency | $k = \frac{c_{obs} - c_{exp}}{1 - c_{exp}}$ | Adjusts for accuracy via $c_{exp} = acc_1 \cdot acc_2 + (1 - acc_1)(1 - acc_2)$ |
| CAPA | $\kappa_p = \frac{c_{obs}^p - c_{exp}^p}{1 - c_{exp}^p}$ | Accounts for sample level probabilities $c_{obs}^p = \frac{1}{|D|} \sum_{x=1}^{D} \sum_{i=1}^{O} p_1(o_i)_x \cdot p_2(o_i)_x$ and accounts for accuracy via $c_{exp}^p = \bar{p}_1 \cdot \bar{p}_2 + \frac{1}{|D|} \sum_{x=1}^{D} \frac{1}{|O_x| - 1}(1 - \bar{p}_1)(1 - \bar{p}_2)$ |

*Table 4.* Comparison of different inter-rater metrics

of classes instead of number of options:

$$c_{\exp}^p = \underbrace{\overline{p_1} \cdot \overline{p_2}}_{\text{chance agreement on correct class}} + \tag{12}$$

$$\underbrace{(1 - \overline{p_1}) \cdot (1 - \overline{p_2}) \cdot \frac{1}{|D|} \sum_{x \in D} \frac{1}{|C(x)| - 1}}_{\text{chance agreement on incorrect class}} \tag{13}$$

In principle, the above implementation could also be adjusted to further take into account the class categories by computing the marginal probabilities per class as:

$$\overline{p(c_i)_*} = \frac{1}{|D|} \sum_{i=1}^{D} p_*(c_i) \text{ where } c_i \neq \text{ground truth}, \tag{14}$$

and replacing the chance agreement on incorrect class with the product of per class 'incorrect' probabilities.

**Exact or Fuzzy Match**: Here, models are not provided categories or options to choose between, and instead provide an answer from an unconstrained set. The model's output string is matched with a reference answer. Here, the probability of independent models agreeing by chance approaches zero due to an unconstrained set of outputs. Further computing probabilistic agreement is difficult over conditional distributions across multiple tokens. We recommend calculating the discrete version of CAPA, where $c_{obs}^{EM} = \frac{1}{|D|} \sum_{x=1}^{D} \mathbb{I}[m_1(x) == m_2(x)]$, and $c_{exp}^{EM} = acc_1 \cdot acc_2$, finally computing $\frac{c_{obs}^{EM} - c_{exp}^{EM}}{1 - c_{exp}^{EM}}$.

**Regression**: For tasks with numeric predictions like regression, for the observed agreement $c_{obs}$, one could measure a distance metric over the two models' predictions, aggregating across samples. Once again, models with lower error would have lower distance in prediction, so the challenge lies in defining chance agreement $c_{exp}$, for a model with a given error. We would have to make appropriate assumptions about the distribution of errors, such as assuming they are gaussian, and compute $c_{exp}$ accordingly.

### A.4. Detailed Discussion on Design Choices

In this section, we discuss alternative design choices we could have taken. For an overview of the equations for each metric, see table 4.

**Why not use inter-annotator agreement metrics like Cohen's $\kappa$?** Cohen's $\kappa$, Scott's $\pi$, Krippendorf's $\alpha$ measure how people differ when answering survey questions, focusing on the reliability of those questions and the data (Krippendorff, 2004). They assume nominal data and computes marginal probability distributions per category. However, MCQs do not have an inherent category 'a' or 'b', i.e. options can be permuted, so we cannot compute such marginal probability distributions. Moreover, measuring LM similarity requires adjusting for chance agreement due to accuracy to avoid inflating similarity for high-accuracy models (Geirhos et al., 2020). For inter-annotator agreement metrics stemming from human survey studies — where there is no built-in concept of accuracy — and thus they are unsuitable for LM analysis without additional modification.

**Should $c_{\text{exp}}^p$ be defined similarly to Cohen's $\kappa$ or Scott's $\pi$?** When measuring similarity between LM Judges and human annotators, Thakur et al. (2024) recommend using Scott's $\pi$ over Cohen's $\kappa$, as it is a better metric for inter-annotator agreement studies (Krippendorff, 2004). The two differ in how they compute $c_{\text{exp}}$, Scott's $\pi$ assumes that the two human raters are sampled from a common distribution, estimating it by averaging the marginal probabilities of the two raters. This is in contrast to Cohen's $\kappa$, which assumes the given different marginal distributions for the two raters. In our case, we wish to account for chance agreement due to accuracies rather than the marginal distribution over classes. To see the relative comparison of how Cohen's $\kappa$ and Scott's $\pi$ behave in our setting, we consider an example.

Suppose we have a binary classification problem, where both models always agree when they are both wrong as there is only one incorrect option. We now consider two pairs of models. Pair 1 has accuracies $0.2, 0.8$, whereas in pair 2, both models have accuracies $0.5$. Intuitively, if both pairs were to have the same observed agreement, it would be more surprising if this happened for pair 1 than pair 2, given the vast difference in their accuracy. In other words, models in pair 2 are more similar than expected for independent models with the given accuracies than pair 1. We want this to be reflected in our similarity metric.

For pair 1, Scott's $\pi$, $c_{\text{exp}}$ would be computed assuming a joint accuracy of $\frac{0.2+0.8}{2} = 0.5$, and for pair 2 with the same joint accuracy $\frac{0.5+0.5}{2} = 0.5$, giving $c_{\text{exp}} = 0.5^2 + (1-0.5)^2 = 0.5$. Cohen's $\kappa$ of pair 2 would be computed as $0.5 \cdot 0.5 + (1-0.5) \cdot (1-0.5) = 0.5$ too. However, for Cohen's $\kappa$ of pair 1, $c_{\text{exp}} = 0.2 \cdot 0.8 + 0.8 \cdot 0.2 = 0.32$. This means for a fixed observed agreement $c_{\text{obs}}$, say $0.5$, $\pi = \frac{0.5-0.5}{1-0.5} = 0$ for both models, and similarly $\kappa = 0$ for pair 2. However, for pair 1, $\kappa = \frac{0.5-0.32}{1-0.32} = 0.264$. Indeed, Scott's $\pi$ would lead us to think both pairs are equally similar, whereas $\kappa$ indicates pair 2 is more similar, beyond chance agreement arising due to accuracy. Thus $\kappa$ has the more desirable behavior.

More broadly, we do not wish to assume both models are drawn from a joint distribution, assigning them a common mean accuracy. However, Scott's $\pi$ does this, which makes sense when calculating reliability of surveys or measuring alignment between human and LLM judges. However, this does not make sense in our setting where we wish to adjust for chance agreement expected due to the two model's given accuracies. Hence, we choose to define $c_{\text{exp}}$ similar to Cohen's $\kappa$, where we retain the difference in the two model's accuracies when computing chance agreement.

**Why not use Matthews Correlation Coefficient**: We could take a completely different approach by computing the Pearson or Matthews Correlation Coefficient of the binary vectors of sample-wise correctness for the two models (Chicco et al., 2021). However, it would be difficult to incorporate probabilistic information, and that models can be incorrect and still disagree by predicting different options. In other words, it suffers from the same issues as error consistency, and we found it more difficult to extend.

**Why not use regression analysis?** We could have performed a multinomial regression using the probabilities of the first model to predict probabilities of the second model, using this predictability as a measure of similarity. However, it is unclear whether a linear model would be enough. Ideally this prediction should also be contextualized on the input sample, but for this we would need a model-based metric to obtain a representation of the input sample. We chose to stick to a more interpretable, closed-form metric.

**Why not use divergence metrics like KL or JSD?** KL-divergence or Jensen–Shannon Distance (JSD) can measure the divergence between probability distributions assigned by models to the options with lucrative information-theoretic properties. Further, JSD is a valid distance metric, and normalized between $0$ and $1$. We could use the mean JSD over all questions as a model similarity metric. However, higher-accuracy models are expected to have lower JSD simply because they have more correct answers, i.e, end up assigning more probability mass to correct options across samples. Retaining the information-theoretic properties of JSD while adjusting for chance agreement due to accuracy remains an interesting open problem.

**Why not use JSD of the two distributions instead of overlap in computing CAPA?**: We could have plugged $1 - JSD$ into $c_{\text{obs}}^p$ in the $\kappa_p$ formula. It is also possible to define $c_{\text{exp}}^p$ by computing JSD between the two independent model distributions defined and subtracting from 1. However, JSD instead of probabilistic overlap is not intuitively interpretable, especially when divided by the possible excess agreement as in $\kappa_p$. $c_{\text{obs}}^p$ computes the expected agreement when sampling from both models based on the probability distribution they assign to the options. Intuitively, it gives us the fraction of times the two models would agree if we kept sampling predictions from these distributions infinitely.

**Why a uniform prior?** In the present work we make the assumption that the prior error distribution follows a uniform distribution. In the context of multiple choice tasks, where by design each answer should have an equal distribution in the dataset, this is a valid and data grounded assumption. However, our metric is not limited to a uniform prior assumption. In

principle, one can easily adjust the agreement by chance calculation (eq. 10) by a different prior distribution assumption given the data observed, for example, a categorical distribution if the class distribution in the data is non-uniform.

## A.5. Probabilistic versions of popular agreement metrics

We now provide probabilistic versions of Cohen's $\kappa$ , Scott's $\pi$, and Fleiss' $\kappa_F$, so that the interested reader can contrast them with CAPA.

**Probabilistic Cohen's $\kappa$** One can obtain a probabilistic Cohen's $\kappa$ by computing $P_0$ as $c_{\text{obs}}^p$, therefore accounting for the observed agreement based on model output probabilities. While $P_e = \sum_{i=1}^{C} \frac{1}{|D|} \sum_{x=1}^{D} p_1(c_i)_x \cdot \frac{1}{|D|} \sum_{x=1}^{D} p_2(c_i)_x$ where we compute the product of marginals for each class.

**Probabilistic Scott's $\pi$** Similarly to Cohen's $\kappa$ to compute the observed agreement probabilistically we compute the average product across probabilities for 2 models, meaning $P_0$ becomes $c_{\text{obs}}^p$. While we adjust $P_e$ computation as follows: $P_e = \sum_{i=1}^{C} (\frac{1}{2}(\frac{1}{|D|} \sum_{x=1}^{D} p_1(c_i)_x + \frac{1}{|D|} \sum_{x=1}^{D} p_2(c_i)_x))^2$, where we now compute the sum of the marginal probabilities per class as we assume that both models have a shared marginal distribution.

**Probablistic Fleiss' Kappa ($\kappa_F$):** It extends the $\frac{c_{\text{obs}} - c_{\text{exp}}}{1 - c_{\text{exp}}}$ formula to more than two models, where the observed and chance agreement is computed across pairs of two in the set of models. Like $\pi$ it assumes chance predictions are sampled from a common combined distribution. While generally Fleiss' Kappa allows a partial random subset of annotators for each question, in our work we assume all models annotate all questions. Let $M$ be the number of models, and $|C|$ be the number of classes. Let $m_{xi}$ be the number of models that put sample $x$ in class $i$. Let $P_x = \frac{1}{M(M-1)} \sum_{i \in C} m_{xi}(m_{xi} - 1)$ be the proportion of observed pairwise agreements for each question. $c_{\text{obs}} = \frac{1}{|D|} \sum_{x \in D} P_x$. For the chance agreement, $c_{\text{exp}} = \sum_{i \in C} (\frac{\sum_{1 \le j \le M} p_j(i)}{M})^2$.

Let $M$ be the number of models, and $|C|$ be the number of classes. Let $m_{xi}$ be the number of models that put sample $x$ in class $i$. Let $P_x = \frac{1}{M(M-1)} \sum_{i \in C} m_{xi}(m_{xi} - 1)$ be the proportion of observed pairwise agreements for each question. $c_{\text{obs}} = \frac{1}{|D|} \sum_{x \in D} P_x$. For the chance agreement, $c_{\text{exp}} = \sum_{i \in C} (\frac{\sum_{1 \le j \le M} p_j(i)}{M})^2$.

## A.6. Theoretical bounds for CAPA

**Bounds for $c_{\text{obs}}^p$.** Compared to Geirhos et al. (2020) the resulting observed agreement is strictly greater than 0, as all probabilities are positive values, and strictly smaller than 1, as the sum of probability products is strictly smaller than the sum of probabilities:

$$0 < c_{\text{obs}}^p < 1 \tag{15}$$

Theorem: If $0 < a < 1$ and $0 < b < 1$, and $a + b = 1$, then $a^2 + b^2 < a + b$

Proof:

$$a^2 + b^2 < a + b$$
$$a^2 - a + b^2 - b < 0$$
$$a(a - 1) + b(b - 1) < 0$$

For $0 < a < 1$, $a > 0$ and $a - 1 < 0$, therefore, $a(a - 1) < 0$. For $0 < b < 1$, $b > 0$ and $b - 1 < 0$, therefore $b(b - 1) < 0$. Since $a(a - 1) < 0$ and $b(b - 1) < 0$, their sum will also be negative $a(a - 1) + b(b - 1) < 0$, this implies that indeed $a^2 + b^2 < a + b$.

**Bounds for $c_{\text{exp}}^p$.** The lower bound for $c_{\text{exp}}^p$ is when the first term approaches zero and the scaling fraction approach 0, thus resulting in $c_{\text{exp}}^p = 0$. The upper bound is maximized when both terms are maximized, but as the second term is the inverse of the first times a scaling factor, the maximum upper bound is 1 (as $\overline{p_1} \cdot \overline{p_2} \to 1$, $(1 - \overline{p_1}) \cdot (1 - \overline{p_2}) \to 0$), resulting in:

$$0 < c_{\text{exp}}^p < 1 \tag{16}$$

**Bounds for $\kappa_p$.** The upper bound for $\kappa_p$ is 1. In particular, $\kappa_p$ will always be strictly smaller than 1, but approaching it in the limit.

Theorem: Given $\kappa_p = 1$. Then by definition:

Proof:

$$1 = \frac{c_{\text{obs}}^p - c_{\text{exp}}^p}{1 - c_{\text{exp}}^p}$$
$$1 - c_{\text{exp}}^p = c_{\text{obs}}^p - c_{\text{exp}}^p$$
$$1 = c_{\text{obs}}^p$$

However, as $c_{\text{obs}}^p < 1$, $\kappa_p < 1$.

Although the above implies that CAPA does not obtain 'perfect agreement' as originally defined by Cohen's $k$, we show that this is not a concern for our metric as (1) when model probability for the correct class approach 1, $\kappa_p \to 1$ and (2) using probabilities allows us to capture observed agreement at a more precise level:

1. Theorem: Given probabilities [a,b] and [c,d], where $a, c \to 1$ and conversely $b, d \to 0$, $\kappa_p \to 1$:

   Proof:

   $$c_{\text{obs}}^p = a \cdot c + b \cdot d$$
   $$\text{as } a \cdot c \to 1 \text{ and } b \cdot d \to 0$$
   $$c_{\text{obs}}^p \to 1$$

   which confirms $\kappa_p \to 1$.

2. Geirhos et al. (2020) computes $c_{\text{obs}}$ as $c_{\text{obs}_{i,j}} = \frac{e_{i,j}}{n}$ where $e_{i,j}$ is the number of equal responses. As such, $c_{\text{obs}_{i,j}}$ is independent of the observed output probabilities. However, for a model pair with output probabilities [0.999.. , 0.000..1] versus [0.8. 0.2] (assume the same for both models), we would like the first case to have a higher observed agreement than the second, but Geirhos et al. (2020) fails to capture this, while $c_{\text{obs}}^p$ does:

   Theorem: Given two probabilities [a,b] and [c,d] where $0 < a, b, c, d < 1$, $a + b = 1$, $c + d = 1$, and $a > c, a > d$, $c > d, b < d$, indicates that $a \cdot a + b \cdot b > c \cdot c + d \cdot d$

   Proof:

   $$a \cdot a + b \cdot b > c \cdot c + d \cdot d$$
   $$a^2 + (1-a)^2 > c^2 + (1-c)^2$$
   $$a^2 + (1-a)^2 - (c^2 + (1-c)^2) > 0$$
   $$2a^2 - 2c^2 - 2a + 2c > 0$$
   $$2(a-c)(a+c-1) > 0$$
   $$(a-c)(a+c-1) > 0$$
   $$\text{as } a > c \Rightarrow (a-c) > 0$$
   $$\text{as } a > d \text{ and } c + d = 1 \Rightarrow d = 1 - c, a > 1 - c \Rightarrow a + c > 1, \text{ thus,} \Rightarrow (a+c-1) > 0,$$

   therefore, $a \cdot a + b \cdot b > c \cdot c + d \cdot d$.

The lower bound for $\kappa_p$ is -1. In particular, $\kappa_p$ will always be strictly greater than -1.

Theorem: Given $\kappa_p \geq -1$, and $0 < c_{\text{exp}}^p < 1$, and $0 < c_{\text{obs}}^p < 1$.

Proof:

$$\frac{c_{\text{obs}}^p - c_{\text{exp}}^p}{1 - c_{\text{exp}}^p} \geq -1$$

$$c_{\text{obs}}^p - c_{\text{exp}}^p \geq -(1 - c_{\text{exp}}^p)$$

$$c_{\text{obs}}^p + 1 - 2c_{\text{exp}}^p \geq 0$$

$$c_{\text{obs}}^p \geq 2c_{\text{exp}}^p - 1$$

minimal possible $c_{\text{obs}}^p \to 0$ (complete disagreement)

$$0 \geq 2c_{\text{exp}}^p - 1$$

$$1 \geq 2c_{\text{exp}}^p$$

$$0.5 \geq c_{\text{exp}}^p$$

therefore, $\kappa_p \geq -1$. Even though, the theoretical lower bound for $\kappa_p = -1$, to achieve $\kappa_p = -1$ in practice $c_{\text{obs}}^p$ must be 0 (both models perfectly oppose each other), leading that $c_{\text{obs}}^p = 2c_{\text{exp}}^p - 1, \to c_{\text{exp}}^p = 0.5$. As $c_{\text{obs}}^p$ is computed based on probabilities its value is $c_{\text{obs}} < 1$, therefore, the actual lower bound for $\kappa_p > -1$.

Altogether, the bounds for CAPA are as follows:

$$-1 < \kappa_p < 1 \tag{17}$$

### A.7. CAPA comparison with other inter-rater metrics

**Numerical Example**   For a simple mathematical example consider two models with 2 data samples with the following probability distributions, model 1 = [[0.9,0.1],[0.8, 0.2]] and model 2 = [[0.7,0.3],[0.6, 0.4]]. The underlying ground truth index is [0,1]. For Cohen's $k$ and Scott's $\pi$ we treat this is example as a binary classification with option A and B, converting the probabilities to model 1= [A,A], model 2 = [A, A] (these metrics do not take accuracy into account). The accuracy for both models is 50%. In table 5 we report the computed similarity for each metric as well specify the exact computation values. As it can be noted, all other metrics suffer from the following limitations: (1) Cohen's $\kappa$ and Scott's $\pi$ treat the problem as a classification, as such both metrics report that the similarity between models is 0.00, indicating no relationship as $P_o = P_e$, (2) Probabilistic versions of the metrics slightly deviate from 0.00 however still undermine model similarity, (3) Error consistency over estimates model similarity by ignoring model output probabilities in its $c_{\text{obs}}$ calculation. As such, only CAPA is able to accurately account for the observed sample level similarity across the two models.

*Table 5.* Numerical Example

| Metric | Similarity | Computation |
|---|---|---|
| $\kappa$ | 0.00 | $P_o = \frac{2}{2} = 1.0,\ P_e = \frac{2}{2} \cdot \frac{2}{2} + \frac{0}{2} \cdot \frac{0}{2} = 1.0$ |
| Probabilistic $\kappa$ | 0.01 | $P_o = \frac{1}{2}(0.9 \cdot 0.7 + 0.1 \cdot 0.3 + 0.8 \cdot 0.6 + 0.2 \cdot 0.4) = 0.61$ |
| | | $P_e = \frac{0.9+0.8}{2} \cdot \frac{0.7+0.6}{2} + \frac{0.1+0.2}{2} \cdot \frac{0.3+0.4}{2} = 0.605$ |
| $\pi$ | 0.00 | $P_o = 1.00,\ P_e = \left(\frac{2+2}{2 \cdot 2}\right)^2 + \left(\frac{0+0}{2 \cdot 2}\right)^2 = 1.0$ |
| Probabilistic $\pi$ | $-0.04$ | $P_o = \frac{1}{2}(0.9 \cdot 0.7 + 0.1 \cdot 0.3 + 0.8 \cdot 0.6 + 0.2 \cdot 0.4) = 0.61$ |
| | | $P_e = \left(\left(\frac{0.9+0.8}{2} + \frac{0.7+0.6}{2}\right)\frac{1}{2}\right)^2 + \left(\left(\frac{0.1+0.2}{2} + \frac{0.3+0.4}{2}\right)\frac{1}{2}\right)^2 = 0.625$ |
| error consistency | 1.00 | $c_{\text{obs}} = 1.00,\ c_{\text{exp}} = 0.5 \cdot 0.5 + (1-0.5)(1-0.5) = 0.5$ |
| CAPA | 0.21 | $c_{\text{obs}}^p = \frac{1}{2}(0.9 \cdot 0.7 + 0.1 \cdot 0.3 + 0.8 \cdot 0.6 + 0.2 \cdot 0.4) = 0.61$ |
| | | $\bar{p}_1 = \frac{1}{2}(0.9 + 0.2) = 0.55,\ \bar{p}_2 = \frac{1}{2}(0.7 + 0.4) = 0.55$ |
| | | $c_{\text{exp}} = 0.55 \cdot 0.55 + \frac{1}{2}\frac{2}{2-1}(1-0.55)(1-0.55) = 0.51$ |

**Simulation Experiments** We design a simulation set-up to compare the 'behavior' of the above listed inter-rater metrics with our novel contribution CAPA. In particular, we limit the simulation to a binary classification problem as standard metrics like Cohen's $k$ and Scott's $\pi$ are ill-suited for multiple choice question settings. In total we investigate the performance of 5 metrics: Cohen's $k$ Probabilistic, Scott's $\pi$ Probabilistic, Error consistency, JS-Distance and CAPA. We simulate N=10000 observations for 2 models.

First, we investigate the metric behavior when both models tend towards agreement and their errors are correlated. In particular, we set the accuracy of model A to 90% and it always favors the 1st option (has a high calibration, 0.99), meaning the model is highly confident in it's predictions (e.g. single data point is [0.99, 0.01]). For the other model, model B, we iteratively increase it's accuracy by adjusting it's calibration from 0.01 to 0.99 for the first option, as such, making the models more similar artificially.

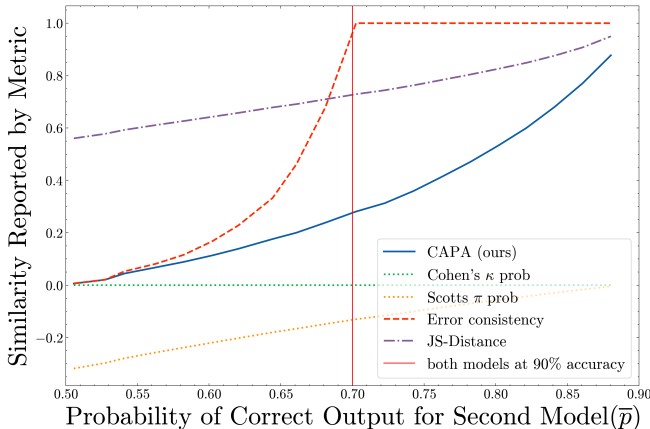

*Figure 7.* **Metric comparison when two models tend towards agreement.** For model A we set accuracy to 90% and calibration to 0.99 of the correct answer. For model B we increase its calibration from 0.01 to 0.99 of the correct answer, so its answer distribution becomes increasingly similar to model A. On y-axis we are plotting metric value on x-axis we are reporting $\overline{p}$ for model B which, as the model becomes more calibrated, approaches accuracy of model A.

As can be noted in Fig. 7 all previous similarity metrics fail to capture the full range of increasing model similarity: (1) EC only captures model similarity up to when their accuracies are matched (both models reach 90% accuracy), yet fails to further distinguish between models that have high agreement [0.99, 0.01] and [0.99, 0.01] versus medium [0.99, 0.01] and [0.75, 0.25]; (2) while Jensen–Shannon distance (computed as 1 - JS-distance) correctly captures the gradual increase in model similarity it overestimates it initially, this is due to the fact that the distance is computed from the average joint distribution; (3) the probabilistic counterparts of standard inter-rater metrics like Cohen's $\kappa$ and Scott's $\pi$ (derivation details in App. A.5) are also ill suited. In particular, in the case of Cohen's $\kappa$ prob the class marginals (for computing the expected agreement) match the observed agreement, therefore leading to inter-rater agreement of 0 even with increasing model similarity, while in the case of Scott's $\pi$ prob we do observe a change in similarity score due to it's assumption of joint marginal distribution, but also because of this assumption the metric overestimates the expected agreement leading to negative (opposing) inter-rater agreement, which is an incorrect conclusion given the simulation data.

Second, we explore the metric behavior when both model accuracies are increasing yet errors remain random (uncorrelated). We again simulate 10'000 binary data points with uniform class distribution. For model A we set the accuracy to 90%, while for model B we incrementally (100 steps) increase it's accuracy from 50% to 90% by randomly selecting data samples which are correct at each step. As such, the errors are uncorrelated between the two models. Results are reported in main paper in Sec. A.7.

**Limitations of CAPA.** In addition, we perform a simulation experiment when the two models become increasingly dissimilar. In this set up we change that the model B always prefers the second option. Thus, by iteratively increasing its calibration we obtain models that maximally differ in their probability distribution, e.g. [0.99, 0.01] and [0.01, 0.99] respectively. As such, also the accuracy of model B decreases overtime from random chance (50%) to 0.10 %, and we would like to obtain metric of -1. As it can be seen in Fig. 8, CAPA never reaches -1. Importantly, the same issue also can be observed for error consistency. This observation comes from the fact that both metrics use the original Cohen's $\kappa$ equation. As explained in Section A.6, $\kappa_p = -1$ iff $c_{\exp}^p = 0.5$. For probabilistic Cohen's $\kappa$ we see the same observation as in Fig. 7, the marginal probability computation is not suited for the given problem. Interestingly, probabilistic Scott's $\pi$ is the only metric that approaches -1. Whilst a desired final outcome, Scott's $\pi$ overestimates model disagreement when model probabilities are independent, [0.99, 0.01] and [0.5, 0.5].

**Possible solution for lower bound.** In the context of the current work, the above limitation is not an issue, as models are trained to maximize accuracy, hence, there will always be some level of agreement. However, if CAPA would be used in

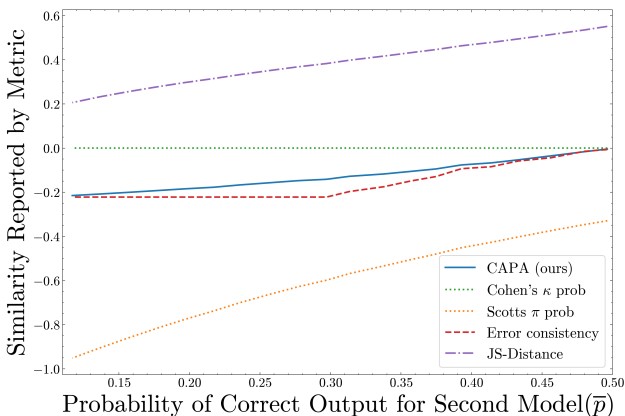

*Figure 8.* Metric comparison when models tend towards disagreement (Read plot from right to left). We compare different metric values for two models in a binary setting. For model A, we set accuracy to 90% and calibration to 0.99 (the model is highly confident in its answers). For model B, we incrementally increase its disagreement with model A by pushing its probability mass to the second option and increasing its calibration to 0.99.

*Figure 9.* Metric comparison when models tend towards disagreement with adjusted CAPA. Replication of fig. 8 but with adjusted CAPA as $\hat{\kappa_p}$, computation following eq. 18.

settings like preference judgments, we would advise to adjust the computation of $c_{\exp}^p$ as described by Safak (2020):

$$\kappa_p = \begin{cases} \frac{c_{\text{obs}}^p - c_{\exp}^p}{1 - c_{\exp}^p} & c_{\text{obs}}^p \geq c_{\exp}^p \\ \frac{c_{\text{obs}}^p - c_{\exp}^p}{c_{\exp}^p - c_{\text{obs-min}}^p} & c_{\text{obs}}^p < c_{\exp}^p \end{cases} \tag{18}$$

where $c_{\text{obs-min}}^p$ is computed as $c_{\text{obs-min}}^p = \max(0, \overline{p_1} + \overline{p_2} - 1)$. This resolves the observed limitation of CAPA over the negative domain, see Fig. 9. Now, as the models become increasingly dissimilar $\hat{\kappa}_p$ approaches -1.

# B. LLM-as-a-Judge

In this section, we extend the LLM-as-a-judge experiments introduced in Section 3. First, we compare CAPA with the related concept of error consistency (Geirhos et al., 2020), demonstrating its advantages in this context. We then present additional experiments to analyze the quality and behavior of the judges, as well as the performance of the evaluated models on the open-style MMLU-Pro benchmark.

To validate our findings, we provide detailed results from the statistical tests summarized in Table 7. Specifically, we conduct Shapiro-Wilk and Breusch-Pagan tests to confirm that the assumptions of normality and homoscedasticity required for partial correlation and multiple regression analyses are satisfied.

Additionally, we outline the experimental setup, including: (1) the filtering process for MMLU-Pro to obtain open-style questions only, (2) the methodology for free-form chain-of-thought inference on this benchmark, and (3) the design of the LLM-as-a-judge evaluation framework. To ensure full reproducibility, we include all prompts and specify the language models used as judges and evaluated models at the end of this section.

## B.1. Extended Multiple Regression Analysis with Model Size as a Confounder

We extend the main paper regression analysis (Sec. 3) by also including model size as a possible confounder of judgment score. We standardize model size by removing the mean and scaling to unit variance, so the model size as a independent variable is on a comparable scale as model similarity and accuracy. We remove 6 outliers based on the fact that the studentized residuals $> 3$. All outliers where in combination with model 'HuggingFaceTB-SmolLM2-135M-Instruct'. The extended regression results are reported in table 6. We report that even when controlling for model accuracy and model size, our computed model similarity score remains a significant predictor of judgment score. Furthermore, whilst for some models model size is a significant predictor of judgment score, when controlling for model similarity and accuracy, the effect size is very small, around 0. All in all, the additional results confirm that our metric accounts for model size as a possible confounder.

*Table 6.* Extended Multiple Regression Results

| Judge | sim | acc | size |
|---|---|---|---|
| Qwen2.5-7B-It | 0.49** | 0.68** | -0.02 |
| Qwen2.5-32B-It | 0.40** | 1.02** | -0.03** |
| Qwen2.5-72B-It | 0.42** | 1.23** | -0.03** |
| Meta-Llama-3.1-8B-It | 0.93** | 0.61** | -0.02 |
| Meta-Llama-3.1-70B-It | 0.74** | 0.92** | -0.03** |
| Llama-3.3-70B-It | 0.61** | 1.03** | -0.04** |
| gemma-2-9b-It | 0.67** | 0.78** | -0.02 |
| gemma-2-27b-It | 0.62** | 0.77** | -0.02 |
| Ministral-8B-It-2410 | 0.65** | 0.50** | -0.02 |

## B.2. Comparison of Judge Scores for Our Similarity vs Error Consistency

In Figure 10 we compare the relationship of judgment scores on the filtered MMLU-Pro dataset using different similarity metrics. On the left, we use CAPA and on the right, we compare against the original error consistency of Geirhos et al. (2020). In both cases, we can see a correlation between the judge scores and the similarity of the LLM-as-a-judge and the model being evaluated. However, the relationship for CAPA is stronger, as shown by a mean Pearson $r$ of 0.9 which is greater than the one of 0.85 if error consistency is used.

## B.3. Evaluating Judge Scores Against Ground-Truth

### B.3.1. MCQ GROUND-TRUTH

Since we source MCQ evaluations from Huggingface OpenLLM Leaderboard 2 throughout the paper, we use their default method to obtain probabilities across MCQ options. For MMLU-Pro and BBH they report the log-likelihood of each option. We apply a softmax to normalize these to 1. We checked this leads to calibrated predictions for base models and overconfident

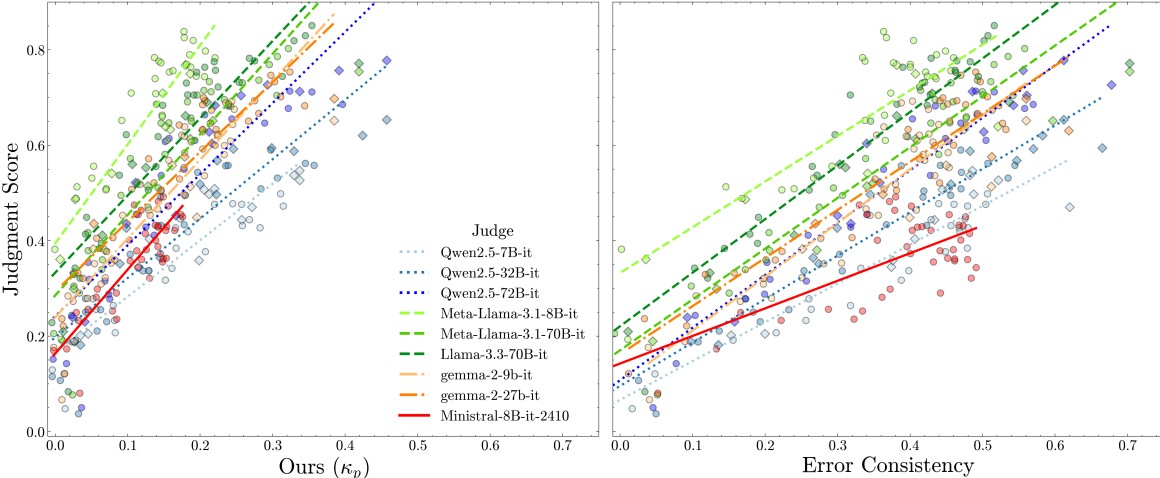

*Figure 10.* **Judgment Scores vs CAPA and vs Error Consistency.** We compare the relationship of judge scores on the filtered MMLU-Pro to our improved error consistency and to the original version of Geirhos et al. (2020).

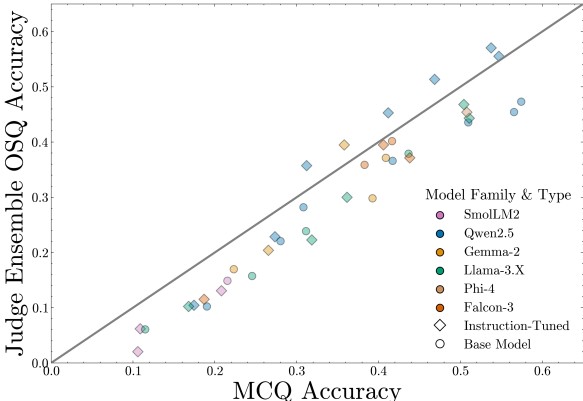

*Figure 11.* **Accuracy of free-form responses compared with multiple-choice accuracy on MMLU-Pro.** The free-form responses were rated using an ensemble of five capable LM judges. Each judge was given access to the original MMLU-Pro reference answers and their decisions whether a given response is correct or not were aggregated using majority voting.

predictions for instruct models, consistent with prior observations about uncertainty of language models (OpenAI, 2024).

### B.3.2. JUDGE ENSEMBLE WITH ACCESS TO REFERENCE ANSWERS

Since we are evaluating model responses given in open style on filtered MMLU-Pro questions using LM-as-a-judge, it is important to investigate whether the responses of the evaluated models are reasonable. To ensure that qualitative differences between models of different sizes and families remain, we compare their performance using free-form responses to the multiple-choice accuracy on the same set of questions. This is shown in Figure 11. Using the same question base for free-form and MCQ evaluation draws a direct connection between functional similarity and the behavior of LLM-as-judges. Focusing on a setting where we have access to ground-truth responses is important to accurately analyze the affinity biases of different LMs when used as evaluators.

**Experimental Setup** Every response is evaluated using an ensemble of five capable LMs used as LLM-as-a-judge from a range of different model families. The judge is given access to the question, the model's free-form response and all MMLU-Pro reference options. For each option we indicate if it is the correct or a wrong option. Using this information, the judge has to decide whether the model's response is correct or wrong. The prompt can be seen in Prompt B.11.2. For every per-sample response, we aggregate the five binary decisions using majority voting. Since there are five judges and it is a binary decision task, there are no ties. A qualitative analysis has shown the high quality of this process

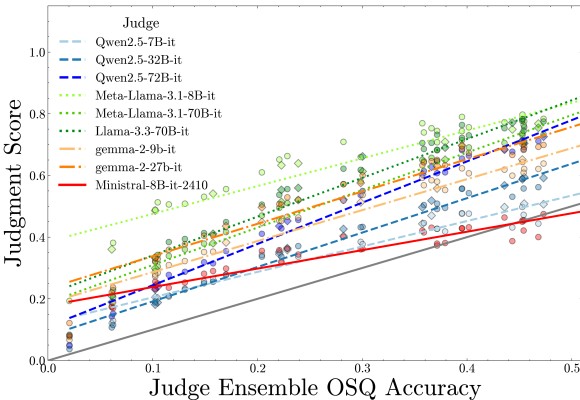

*Figure 12.* **Judgment Scores compared with the ensemble judgment accuracy given access to reference answers.** We compare the judgment scores of each judge using only their own knowledge and capabilities to the rating of a judge ensemble that has access to the ground-truth options. The latter is a good proxy of the real correctness of responses.

in determining the correctness of responses. The judges used are `gemma-2-27b-it`, `Qwen2.5-32B-Instruct`, `Qwen2.5-72B-Instruct`, `Llama-3.1-70-Instruct` and `Llama-3.3-70B-Instruct` (Gemma Team, 2024; Qwen Team, 2025; Llama Team, 2024a;c).

**Open-style and Multiple Choice Correlate** As we can see in Figure 11, there is a high alignment between the performance in MCQ style compared to free-form. For the majority of models, the ordering with MCQ accuracy and open-style accuracy is very similar. There is a consistent trend that performance on the more challenging open evaluation is approximately 5-10% lower. The exception is the instruction-tuned models from the Qwen2.5 and Gemma-2 model families that performed particularly well when giving free-form responses. For all other model families, the instruction tuned and base models show similar performance.

### B.3.3. JUDGE SCORE VALIDITY AGAINST REFERENCE-BASED ENSEMBLE

To evaluate the quality of different judges and to analyze their similarities and differences, we compare judge scores to the correctness assessments of the previously introduced ensemble of judges. The results are shown in Figure 12. As we can see, most models used as LLM-as-a-judge are able to correctly rank capable and less capable models.

**Capability-Dependent Affinity Effects** Even if the ordinal ranking of evaluated models is mostly accurate, there is a consistent trend that too many wrong responses are judged as being right. The exact behavior varies from judge to judge. Consider the small `Llama-3.1-8B-Instruct` for instance: it has a consistent positivity bias and rates too many wrong responses as correct, even for models of low capability. `Qwen2.5-72B-Instruct` on the other hand appears to be much more capable in identifying the wrong responses of low-capability models. However, as the evaluated LMs become stronger, it exhibits the same bias as the smaller Llama judge. This aligns with the findings of Section 3 that LLM-as-judges show an affinity bias, because more capable models are also more similar to `Qwen2.5-72B-Instruct`.

### B.4. Judge Preference in Chat Generation Tasks Against Similarity

**Experimental Setup** To evaluate judge bias in a different context, we evaluated each of the 23 instruction-tuned models of the original experiment on the benchmark AlpacaEval for free-form generation (Dubois et al., 2023). After generating responses for all models on the full set of AlpacaFarm prompts, we used the same nine LMs as judges as before to determine their preferences on all response pairs between the 253 model combinations. A re-implementation of AlpacaEval 2.0 was used to determine preferences (Dubois et al., 2024). The resulting judge preferences were shuffled and used to compute Judgment Elo for each judge. In total, there were 203,665 combinations of prompts and pairs of models to be judged.

**Free-form Judge Preference** Figure 13 shows the relation between a model's Judgment Elo on Alpaca Eval and judge-model similarities using our proposed metric CAPA on MMLU-Pro. Since Alpaca Eval cannot be easily turned into an MCQ benchmark, as required by our similarity metric, using the same similarities on MMLU-Pro as before was inevitable. Despite the domain gap between the data used for the LLM-as-a-judge experiments and CAPA, there is a clear positive correlation between the two, indicating the existence of an affinity bias. This also provides initial evidence that similarity on a diverse

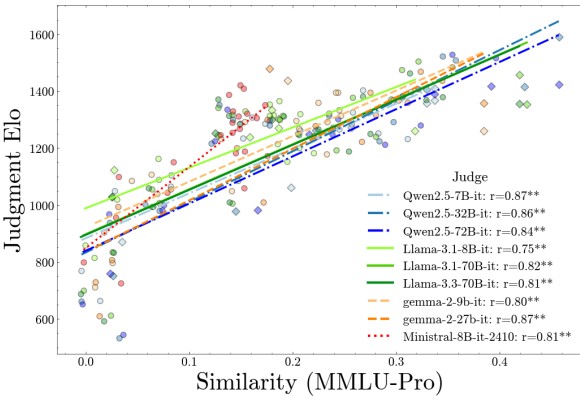

*Figure 13.* **Elo of LMs on AlpacaEval 2.0 as judged by different LLM-as-judges in relation with our similarity metric on MMLU-Pro.** We compare the Elo obtained by using different judges for binary preference pairs on Alpaca Eval to their similarity on the filtered MMLU-Pro dataset.

MCQ benchmark might transfer as a predictor across tasks.

### B.5. Judge Score Against Similarity per Category

MMLU-Pro is a very comprehensive MCQ benchmark to evaluate a model's common knowledge. By filtering some of the questions to ensure that only those that can be answered in free-form remain, we may have unintentionally introduced biases such as Simpson's paradox that affect the preference bias we observed. To ensure that this is not the case, we group the questions for free-form Judgment Scores and MCQ similarity by their original category in Figure 14.

For each category, we only used the judgments and MCQ responses of the questions belonging to that category to compute the Judgment Scores and CAPA as metric for model similarity. As in Figure 3, each line corresponds to one judge model. The same positive correlation between CAPA and Judgment Score can be seen for every category as on the original, aggregated Filtered MMLU-Pro benchmark. This indicates the existence of an affinity bias of LLM-as-a-judge irrespective of the category.

For better readability, we do not report the Pearson r in individual legends for each subplot. Apart from one exception, all Pearson correlation coefficients of the LLM-as-judges in each category were above 0.75 and statistically significant (p=0.01). The only exception was `google/gemma-2-9b-it` on the category "engineering" with $r = 0.28$ and $p = 0.211$.

### B.6. Statistical Testing

This section provides detailed statistical validation of the affinity bias observed in Section 3. We confirm that judge-model similarity correlates with judgment scores even after controlling for MCQ ground-truth accuracy, using partial correlation and multiple regression. Additionally, we verify the statistical assumptions (normality, homoscedasticity) required for these tests across all nine judges.

#### B.6.1. QUANTIFYING CORRELATION STRENGTH USING PARTIAL CORRELATION

We compute partial correlations between judge scores and judge-model similarity while controlling for ground-truth accuracy. All judges show statistically significant positive correlations (Table B.6.1), with coefficients ranging from $r = 0.35$ (`Llama-3.3-70B-Instruct`) to $r = 0.65$ (`Llama-3.1-8B-Instruct`). The strongest correlations occur for smaller judges from the same gemma-2 family (`gemma-2-27b-it` and `gemma-2-9b-it`), while larger Qwen2.5 judges exhibit moderate correlations ($r = 0.42$ and $r = 0.43$). All p-values remain significant ($p < 0.05$), with the most robust results for the larger gemma judge ($p = 0.00001$).

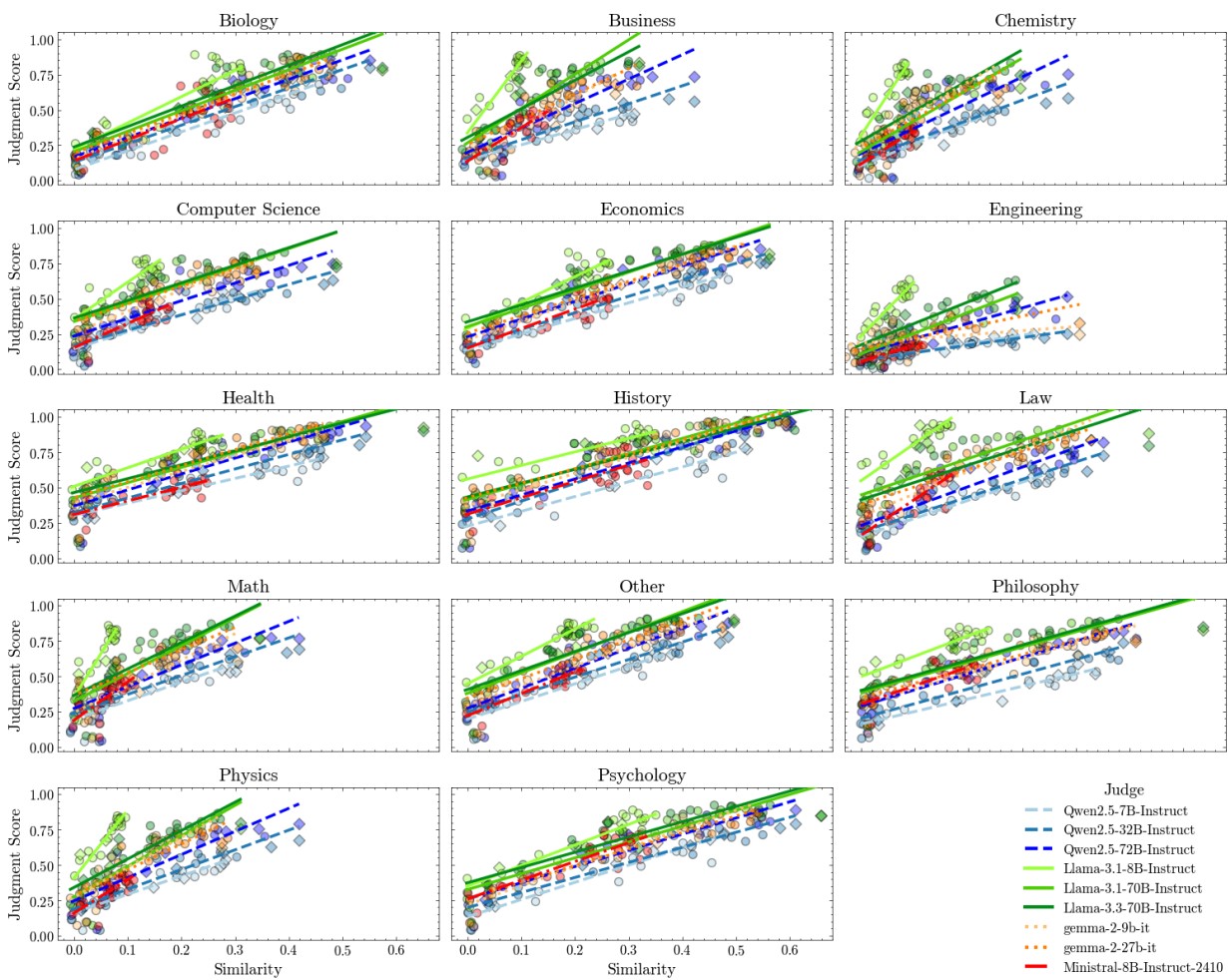

*Figure 14.* **Judgment Score Relation with Model Similarity per Category.** This plot shows the same experiment as Figure 3 except that all questions were grouped by MMLU-Pro category. Each line is a regression model fit between judgment and similarity scores as computed between model and judge pairs. Each point represents a single pair, and ⋄ indicates that both, the model and the judge, come from the same model family.

**Detailed Partial Correlation Results**

| Judge | n | r | CI 95% | p-val |
|---|---|---|---|---|
| Qwen2.5-7B-Instruct | 38 | 0.60043 | [0.34 0.77] | 0.00009 |
| Qwen2.5-32B-Instruct | 38 | 0.43376 | [0.13 0.66] | 0.00732 |
| Qwen2.5-72B-Instruct | 38 | 0.42353 | [0.12 0.66] | 0.00900 |
| Meta-Llama-3.1-8B-Instruct | 38 | 0.65172 | [0.42 0.81] | 0.00001 |
| Meta-Llama-3.1-70B-Instruct | 38 | 0.44770 | [0.14 0.67] | 0.00546 |
| Llama-3.3-70B-Instruct | 38 | 0.34882 | [0.03 0.6 ] | 0.03435 |
| gemma-2-9b-it | 38 | 0.64639 | [0.41 0.8 ] | 0.00002 |
| gemma-2-27b-it | 38 | 0.64808 | [0.41 0.8 ] | 0.00001 |
| Ministral-8B-Instruct-2410 | 39 | 0.59745 | [0.34 0.77] | 0.00007 |

### B.6.2. MULTIPLE REGRESSION

We perform multiple regression analysis with judgment scores as the dependent variable, using model similarity and ground-truth accuracy from the filtered set of MCQ questions as independent variables. Key results across all judges include:

- **Coefficient Significance:** Both similarity and accuracy show statistically significant effects ($p < 0.05$) for all judges. The similarity coefficients range from $\beta = 0.35$ for the large `Llama-3.3-70B-Instruct` to $\beta = 1.15$ for the smaller `Meta-Llama-3.1-8B-Instruct`, while accuracy coefficients span $\beta = 0.43$ for the small `Ministral-8B-Instruct-2410` to $\beta = 1.04$ for the large `Qwen2.5-72B-Instruct`.

- **Model Fit:** All regressions achieve high explanatory power with adjusted $R^2$ values between $0.87$ (`Ministral-8B`) and $0.92$ (`gemma-2-9b-it`).

- **Assumption Verification:**
  - *Normality:* Residuals are normally distributed (Shapiro-Wilk $p > 0.05$) for 7 of 9 judges. Exceptions: `Meta-Llama-3.1-8B-Instruct` ($p = 0.002$) and `Ministral-8B` ($p = 0.012$).
  - *Homoscedasticity:* All models satisfy constant variance assumptions (Breusch-Pagan $p > 0.05$).

For example, the `Qwen2.5-7B-Instruct` judge model shows:

- Significant positive effects for both similarity ($\beta = 0.59$, $p < 0.001$) and accuracy ($\beta = 0.51$, $p < 0.001$)

- Strong model fit ($R^2 = 0.91$, $F(2, 35) = 182.9$, $p = 2.95 \times 10^{-19}$)

- Normally distributed residuals (Shapiro-Wilk $p = 0.690$)

Full regression outputs for all judges are provided in the Tables below. We present detailed regression results for each judge model. Each judge's statistical analysis includes three components: (1) Test summary, (2) Coefficient estimates, and (3) Diagnostic statistics. The consistent significance of similarity coefficients confirms that affinity bias persists even when controlling for actual model capability.

| **Judge:** `Qwen2.5-7B-Instruct` (Qwen Team, 2025) | | | |
|---|---|---|---|
| Model: | OLS | Adj. R-squared: | 0.908 |
| Dependent Variable: | scores | AIC: | -134.3091 |
| Date: | 2025-01-30 11:42 | BIC: | -129.3963 |
| No. Observations: | 38 | Log-Likelihood: | 70.155 |
| Df Model: | 2 | F-statistic: | 182.9 |
| Df Residuals: | 35 | Prob (F-statistic): | 2.95e-19 |
| R-squared: | 0.913 | Scale: | 0.0015837 |

| | Coef. | Std.Err | t | P¿—t— | 95% CI | |
|---|---|---|---|---|---|---|
| Intercept | 0.092 | 0.019 | 4.885 | 0.000 | 0.054 | 0.131 |
| similarity | 0.586 | 0.132 | 4.442 | 0.000 | 0.318 | 0.853 |
| accuracy | 0.506 | 0.098 | 5.185 | 0.000 | 0.308 | 0.704 |

| | | | |
|---|---|---|---|
| Omnibus: | 2.363 | Durbin-Watson: | 2.097 |
| Prob(Omnibus): | 0.307 | Jarque-Bera (JB): | 1.400 |
| Skew: | -0.437 | Prob(JB): | 0.496 |
| Kurtosis: | 3.348 | Condition No.: | 27 |

**Normality & Homoscedasticity:** Shapiro-Wilk Test for Normality: Statistic=0.979, (p-value=0.690). Residuals are likely normally distributed. Breusch-Pagan test for homoscedasticity: Lagrange Multiplier statistic: 0.456 (p-value: 0.796), F-value: 0.213 (p-value: 0.809). No evidence of heteroscedasticity (the residuals have a constant variance, homoscedasticity met).

| **Judge:** Qwen2.5-32B-Instruct (Qwen Team, 2025) | | | |
|---|---|---|---|
| Model: | OLS | Adj. R-squared: | 0.907 |
| Dependent Variable: | scores | AIC: | -114.7801 |
| Date: | 2025-01-30 11:42 | BIC: | -109.8674 |
| No. Observations: | 38 | Log-Likelihood: | 60.390 |
| Df Model: | 2 | F-statistic: | 182.2 |
| Df Residuals: | 35 | Prob (F-statistic): | 3.14e-19 |
| R-squared: | 0.912 | Scale: | 0.0026477 |

| | Coef. | Std.Err | t | P>\|t\| | 95% CI | |
|---|---|---|---|---|---|---|
| Intercept | 0.045 | 0.028 | 1.620 | 0.114 | -0.011 | 0.101 |
| similarity | 0.414 | 0.145 | 2.848 | 0.007 | 0.119 | 0.709 |
| accuracy | 0.861 | 0.132 | 6.513 | 0.000 | 0.593 | 1.129 |

| | | | |
|---|---|---|---|
| Omnibus: | 0.227 | Durbin-Watson: | 2.052 |
| Prob(Omnibus): | 0.893 | Jarque-Bera (JB): | 0.411 |
| Skew: | 0.127 | Prob(JB): | 0.814 |
| Kurtosis: | 2.558 | Condition No.: | 25 |

**Normality & Homoscedasticity:** Shapiro-Wilk Test for Normality: Statistic=0.989, (p-value=0.965). Residuals are likely normally distributed. Breusch-Pagan test for homoscedasticity: Lagrange Multiplier statistic: 3.097 (p-value: 0.213), F-value: 1.553 (p-value: 0.226). No evidence of heteroscedasticity (the residuals have a constant variance, homoscedasticity met).

| **Judge:** Qwen2.5-72B-Instruct (Qwen Team, 2025) | | | |
|---|---|---|---|
| Model: | OLS | Adj. R-squared: | 0.913 |
| Dependent Variable: | scores | AIC: | -103.8097 |
| Date: | 2025-01-30 11:42 | BIC: | -98.8969 |
| No. Observations: | 38 | Log-Likelihood: | 54.905 |
| Df Model: | 2 | F-statistic: | 195.4 |
| Df Residuals: | 35 | Prob (F-statistic): | 1.03e-19 |
| R-squared: | 0.918 | Scale: | 0.0035339 |

| | Coef. | Std.Err | t | P>\|t\| | 95% CI | |
|---|---|---|---|---|---|---|
| Intercept | 0.064 | 0.032 | 2.038 | 0.049 | 0.000 | 0.128 |
| similarity | 0.474 | 0.171 | 2.766 | 0.009 | 0.126 | 0.822 |
| accuracy | 1.043 | 0.156 | 6.702 | 0.000 | 0.727 | 1.359 |

| | | | |
|---|---|---|---|
| Omnibus: | 0.139 | Durbin-Watson: | 1.776 |
| Prob(Omnibus): | 0.933 | Jarque-Bera (JB): | 0.286 |
| Skew: | -0.124 | Prob(JB): | 0.867 |
| Kurtosis: | 2.655 | Condition No.: | 25 |

**Normality & Homoscedasticity:** Shapiro-Wilk Test for Normality: Statistic=0.989, (p-value=0.968). Residuals are likely normally distributed. Breusch-Pagan test for homoscedasticity: Lagrange Multiplier statistic: 3.562 (p-value: 0.168), F-value: 1.810 (p-value: 0.179). No evidence of heteroscedasticity (the residuals have a constant variance, homoscedasticity

met).

| **Judge:** `Meta-Llama-3.1-8B-Instruct` (Llama Team, 2024a) | | | |
|---|---|---|---|
| Model: | OLS | Adj. R-squared: | 0.881 |
| Dependent Variable: | scores | AIC: | -114.9610 |
| Date: | 2025-01-30 11:42 | BIC: | -110.0482 |
| No. Observations: | 38 | Log-Likelihood: | 60.481 |
| Df Model: | 2 | F-statistic: | 138.6 |
| Df Residuals: | 35 | Prob (F-statistic): | 2.34e-17 |
| R-squared: | 0.888 | Scale: | 0.0026351 |

| | Coef. | Std.Err | t | P$>$—t— | 95% CI | |
|---|---|---|---|---|---|---|
| Intercept | 0.327 | 0.023 | 14.309 | 0.000 | 0.281 | 0.374 |
| similarity | 1.149 | 0.226 | 5.083 | 0.000 | 0.690 | 1.608 |
| accuracy | 0.532 | 0.106 | 5.035 | 0.000 | 0.317 | 0.746 |

| | | | |
|---|---|---|---|
| Omnibus: | 23.344 | Durbin-Watson: | 2.611 |
| Prob(Omnibus): | 0.000 | Jarque-Bera (JB): | 52.336 |
| Skew: | -1.424 | Prob(JB): | 0.000 |
| Kurtosis: | 7.994 | Condition No.: | 31 |

**Normality & Homoscedasticity:** Shapiro-Wilk Test for Normality: Statistic=0.892, (p-value=0.002). Residuals are likely not normally distributed. Breusch-Pagan test for homoscedasticity: Lagrange Multiplier statistic: 4.186 (p-value: 0.123), F-value: 2.166 (p-value: 0.130). No evidence of heteroscedasticity (the residuals have a constant variance, homoscedasticity met).

| **Judge:** `Meta-Llama-3.1-70B-Instruct` (Llama Team, 2024a) | | | |
|---|---|---|---|
| Model: | OLS | Adj. R-squared: | 0.903 |
| Dependent Variable: | scores | AIC: | -103.3457 |
| Date: | 2025-01-30 11:42 | BIC: | -98.4329 |
| No. Observations: | 38 | Log-Likelihood: | 54.673 |
| Df Model: | 2 | F-statistic: | 172.4 |
| Df Residuals: | 35 | Prob (F-statistic): | 7.60e-19 |
| R-squared: | 0.908 | Scale: | 0.0035773 |

| | Coef. | Std.Err | t | P$>$—t— | 95% CI | |
|---|---|---|---|---|---|---|
| Intercept | 0.140 | 0.031 | 4.533 | 0.000 | 0.077 | 0.202 |
| similarity | 0.615 | 0.208 | 2.962 | 0.005 | 0.193 | 1.036 |
| accuracy | 0.917 | 0.157 | 5.827 | 0.000 | 0.598 | 1.237 |

| | | | |
|---|---|---|---|
| Omnibus: | 4.624 | Durbin-Watson: | 1.984 |
| Prob(Omnibus): | 0.099 | Jarque-Bera (JB): | 3.237 |
| Skew: | -0.616 | Prob(JB): | 0.198 |
| Kurtosis: | 3.724 | Condition No.: | 28 |

**Normality & Homoscedasticity:** Shapiro-Wilk Test for Normality: Statistic=0.974, (p-value=0.502). Residuals are likely normally distributed. Breusch-Pagan test for homoscedasticity: Lagrange Multiplier statistic: 2.975 (p-value: 0.226), F-value: 1.487 (p-value: 0.240). No evidence of heteroscedasticity (the residuals have a constant variance, homoscedasticity met).

| **Judge:** `Llama-3.3-70B-Instruct` (Llama Team, 2024c) | | | |
|---|---|---|---|
| Model: | OLS | Adj. R-squared: | 0.884 |
| Dependent Variable: | scores | AIC: | -94.3204 |
| Date: | 2025-01-30 11:42 | BIC: | -89.4077 |
| No. Observations: | 38 | Log-Likelihood: | 50.160 |
| Df Model: | 2 | F-statistic: | 142.6 |
| Df Residuals: | 35 | Prob (F-statistic): | 1.49e-17 |
| R-squared: | 0.891 | Scale: | 0.0045363 |

| | Coef. | Std.Err | t | P¿—t— | 95% CI | |
|---|---|---|---|---|---|---|
| Intercept | 0.162 | 0.036 | 4.544 | 0.000 | 0.089 | 0.234 |
| similarity | 0.487 | 0.221 | 2.202 | 0.034 | 0.038 | 0.935 |
| accuracy | 1.022 | 0.177 | 5.770 | 0.000 | 0.662 | 1.381 |

| | | | |
|---|---|---|---|
| Omnibus: | 4.168 | Durbin-Watson: | 1.898 |
| Prob(Omnibus): | 0.124 | Jarque-Bera (JB): | 2.830 |
| Skew: | -0.584 | Prob(JB): | 0.243 |
| Kurtosis: | 3.652 | Condition No.: | 27 |

**Normality & Homoscedasticity:** Shapiro-Wilk Test for Normality: Statistic=0.978, (p-value=0.642). Residuals are likely normally distributed. Breusch-Pagan test for homoscedasticity: Lagrange Multiplier statistic: 2.500 (p-value: 0.287), F-value: 1.232 (p-value: 0.304). No evidence of heteroscedasticity (the residuals have a constant variance, homoscedasticity met).

| **Judge:** `gemma-2-9b-it` (Gemma Team, 2024) | | | |
|---|---|---|---|
| Model: | OLS | Adj. R-squared: | 0.917 |
| Dependent Variable: | scores | AIC: | -122.8074 |
| Date: | 2025-01-30 11:42 | BIC: | -117.8946 |
| No. Observations: | 38 | Log-Likelihood: | 64.404 |
| Df Model: | 2 | F-statistic: | 206.0 |
| Df Residuals: | 35 | Prob (F-statistic): | 4.36e-20 |
| R-squared: | 0.922 | Scale: | 0.0021435 |

| | Coef. | Std.Err | t | P¿—t— | 95% CI | |
|---|---|---|---|---|---|---|
| Intercept | 0.134 | 0.021 | 6.356 | 0.000 | 0.091 | 0.177 |
| similarity | 0.763 | 0.152 | 5.012 | 0.000 | 0.454 | 1.072 |
| accuracy | 0.688 | 0.097 | 7.129 | 0.000 | 0.492 | 0.884 |

| | | | |
|---|---|---|---|
| Omnibus: | 6.751 | Durbin-Watson: | 1.901 |
| Prob(Omnibus): | 0.034 | Jarque-Bera (JB): | 5.969 |
| Skew: | -0.621 | Prob(JB): | 0.051 |
| Kurtosis: | 4.492 | Condition No.: | 25 |

**Normality & Homoscedasticity:** Shapiro-Wilk Test for Normality: Statistic=0.959, (p-value=0.179). Residuals are likely normally distributed. Breusch-Pagan test for homoscedasticity: Lagrange Multiplier statistic: 2.550 (p-value: 0.279), F-value: 1.259 (p-value: 0.297). No evidence of heteroscedasticity (the residuals have a constant variance, homoscedasticity met).

| **Judge:** `gemma-2-27b-it` (Gemma Team, 2024) | | | |
|---|---|---|---|
| Model: | OLS | Adj. R-squared: | 0.919 |
| Dependent Variable: | scores | AIC: | -121.3075 |
| Date: | 2025-01-30 11:42 | BIC: | -116.3947 |
| No. Observations: | 38 | Log-Likelihood: | 63.654 |
| Df Model: | 2 | F-statistic: | 212.0 |
| Df Residuals: | 35 | Prob (F-statistic): | 2.76e-20 |
| R-squared: | 0.924 | Scale: | 0.0022298 |

|  | Coef. | Std.Err | t | P¿—t— | 95% CI | |
|---|---|---|---|---|---|---|
| Intercept | 0.191 | 0.022 | 8.655 | 0.000 | 0.146 | 0.235 |
| similarity | 0.705 | 0.140 | 5.034 | 0.000 | 0.421 | 0.989 |
| accuracy | 0.677 | 0.106 | 6.407 | 0.000 | 0.462 | 0.892 |

| | | | | |
|---|---|---|---|---|
| Omnibus: | 8.920 | | Durbin-Watson: | 1.882 |
| Prob(Omnibus): | 0.012 | | Jarque-Bera (JB): | 8.659 |
| Skew: | -0.791 | | Prob(JB): | 0.013 |
| Kurtosis: | 4.722 | | Condition No.: | 24 |

**Normality & Homoscedasticity:** Shapiro-Wilk Test for Normality: Statistic=0.945, (p-value=0.062). Residuals are likely normally distributed. Breusch-Pagan test for homoscedasticity: Lagrange Multiplier statistic: 1.645 (p-value: 0.439), F-value: 0.792 (p-value: 0.461). No evidence of heteroscedasticity (the residuals have a constant variance, homoscedasticity met).

| **Judge:** `Ministral-8B-Instruct-2410` (Mistral AI, 2024) | | | |
|---|---|---|---|
| Model: | OLS | Adj. R-squared: | 0.868 |
| Dependent Variable: | scores | AIC: | -146.8086 |
| Date: | 2025-01-30 11:42 | BIC: | -141.8179 |
| No. Observations: | 39 | Log-Likelihood: | 76.404 |
| Df Model: | 2 | F-statistic: | 125.6 |
| Df Residuals: | 36 | Prob (F-statistic): | 5.80e-17 |
| R-squared: | 0.875 | Scale: | 0.0012608 |

|  | Coef. | Std.Err | t | P¿—t— | 95% CI | |
|---|---|---|---|---|---|---|
| Intercept | 0.117 | 0.016 | 7.187 | 0.000 | 0.084 | 0.150 |
| similarity | 0.825 | 0.185 | 4.470 | 0.000 | 0.451 | 1.199 |
| accuracy | 0.432 | 0.063 | 6.803 | 0.000 | 0.303 | 0.560 |

| | | | | |
|---|---|---|---|---|
| Omnibus: | 9.707 | | Durbin-Watson: | 1.766 |
| Prob(Omnibus): | 0.008 | | Jarque-Bera (JB): | 8.755 |
| Skew: | -0.999 | | Prob(JB): | 0.013 |
| Kurtosis: | 4.180 | | Condition No.: | 36 |

**Normality & Homoscedasticity:** Shapiro-Wilk Test for Normality: Statistic=0.925, (p-value=0.012). Residuals are likely not normally distributed. Breusch-Pagan test for homoscedasticity: Lagrange Multiplier statistic: 1.270 (p-value: 0.530), F-value: 0.606 (p-value: 0.551). No evidence of heteroscedasticity (the residuals have a constant variance, homoscedasticity met).

### B.7. Experimental Setup for Filtering MMLU-Pro

We evaluate or models and judges on a set of questions that can be answered as MCQ as well as in open-style without access to reference options. This benchmark is obtained by using the filtering process proposed by Myrzakhan et al. (2024) on MMLU-Pro, whereas it was originally used to filter MMLU (Hendrycks et al., 2021b; Wang et al., 2024). Every question is evaluated twice using a `Qwen-2.5-32B-Instruct` LM: first, it is judged in a binary way whether it is possible to answer the question without access to the MCQ options. In the second iteration, the judge gives a fine-grained confidence score. If either the binary decision is positive or the confidence is above a threshold, the question becomes part of our filtered benchmark. After this filtering process, 8707 of the original 12032 questions remain. The detailed prompts are described in Prompts B.11.5 and B.11.6.

### B.8. Experimental Setup to Perform Free-Form Inference on Filtered MMLU-Pro

To obtain the per-sample responses of every model on the filtered MMLU-Pro benchmark, we evaluate them using a custom task on the LM Evaluation Harness (Gao et al., 2023). Whereas the MCQ results from the Open LLM Leaderboard were generated using 5-shot evaluation without chain-of-thought (CoT) prompt, we included CoTs when performing free-form inference (Myrzakhan et al., 2024). This was necessary to ensure sufficient instruction following and response quality even

for small base models, because free-form generation is more challenging than MCQ evaluation, where access to reference answers is given.

We modified every 5-shot CoT prompt by removing the answer options from the end of the question and replacing every reference to them in the CoT with the corresponding answer text. An example of this process is shown in Prompts B.11.3 and B.11.4. Our benchmark is implemented as a task for the LM Eval Harness (Gao et al., 2023). Every CoT response is generated until a stop condition is met. The final response that is judged is extracted using regex matching. We use vLLM as the backend for the LM Eval Harness (Kwon et al., 2023). Even for instruction-tuned models, the options `--apply_chat_template` and `--fewshot_as_multiturn` were omitted because for the majority of LMs inspected the quality of responses decreased slightly to severely. However, we did not thoroughly investigate whether this is the case for every single model.

### B.9. Experimental Setup for LLM-as-a-Judge on Filtered MMLU-Pro

This section describes the setup of the experiment for Figure 3. On the x-axis we show the similarity between our LLM-as-a-judge and the LM that is being evaluated, whereas the y-axis shows how the given responses of that model were rated by the same judge. The list of judges is shown in Table 7 and the pool of models evaluated can be seen in Table 8.

For the computation of similarities, we use the logs of the official evaluation runs of Myrzakhan et al. (2024) that are provided on huggingface.co. The set of responses is filtered to include only those questions that were rated as answerable in open-style without access to the reference options, as previously described in Section B.7. Using the logarithmic probabilities of the models for the answer options of this set of questions, we compute CAPA and other similarities. In addition, for each model-judge pair data samples where the ground truth option differed were excluded from the final analysis, for the final sample count per pair see from table **??** to table **??**.

Next, the judgment scores are obtained by prompting each LLM-as-a-judge to decide whether a given response to a question is correct or not. To mimic more common, but ungrounded settings for automatic AI evaluation, such as Arena-hard-auto or AlpacaEval 2.0, we do not provide the judge with access to ground-truth responses or MCQ answer options (Li et al., 2024b; Dubois et al., 2024). Since ground-truth responses for each question are available, it is possible to analyze the affinity bias of different judges and determine if there is any unfair preference. The prompt given to the judge is shown in Section B.11.1. Each final decision was given as token "0" (incorrect) or "1" (correct). Instruction-following is exceptional for the models used as LLM-as-a-judge, so the amount of discarded samples due to invalid responses is negligible. Finally, the *Judge Score* of an evaluated model is computed by averaging the judge decisions across the set of questions.

### B.10. List of Judges and Evaluated Language Models

Our judge preference experiments were performed using nine high-capability, open-weight models from four different model families. The models that represent the current state-of-the-art of open-weight language models from very small up to models with 72 billion parameters. Whereas the judges are all instruction-tuned, the list of evaluated models contains base models as well.

Whenever possible, we evaluated both the base model and the instruction-tuned model for every combination of size and model family. Sometimes this was not possible, because the base model's weights were not available on huggingface, evaluations on the Open LLM Leaderboard v2 were not provided or the LM consistently crashed in vLLM when performing inference (Myrzakhan et al., 2024; Kwon et al., 2023). The list below shows all models that are part of our experiments.

*Table 7.* **LMs used as LLM-as-a-Judge**

| Judge Model Name | |
|---|---|
| `google/gemma-2-9b-it` | (Gemma Team, 2024) |
| `google/gemma-2-27b-it` | (Gemma Team, 2024) |
| `Qwen/Qwen2.5-7B-Instruct` | (Qwen Team, 2025) |
| `Qwen/Qwen2.5-32B-Instruct` | (Qwen Team, 2025) |
| `Qwen/Qwen2.5-72B-Instruct` | (Qwen Team, 2025) |
| `meta-llama/Meta-Llama-3.1-8B-Instruct` | (Llama Team, 2024a) |
| `meta-llama/Meta-Llama-3.1-70B-Instruct` | (Llama Team, 2024a) |
| `meta-llama/Llama-3.3-70B-Instruct` | (Llama Team, 2024c) |
| `mistralai/Ministral-8B-Instruct-2410` | (Mistral AI, 2024) |

*Table 8.* **LMs Evaluated on the Filtered MMLU-Pro Benchmark**

| Model Name | |
|---|---|
| **Base Models** | **Instruction-tuned Models** |
| Gemma-2 Family (Gemma Team, 2024) | |
| `google/gemma-2-2b` | `google/gemma-2-2b-it` |
| `google/gemma-2-9b` | `google/gemma-2-9b-it` |
| `google/gemma-2-27b` | `google/gemma-2-27b-it` |
| SmolLM2 Family (Allal et al., 2024) | |
| | `HuggingFaceTB/SmolLM2-135M-Instruct` |
| | `HuggingFaceTB/SmolLM2-360M-Instruct` |
| `HuggingFaceTB/SmolLM2-1.7B` | `HuggingFaceTB/SmolLM2-1.7B-Instruct` |
| Llama 3.1/3.2/3.3 Model Family (Llama Team, 2024a;b;c) | |
| `meta-llama/Meta-Llama-3.1-8B` | `meta-llama/Meta-Llama-3.1-8B-Instruct` |
| `meta-llama/Meta-Llama-3.1-70B` | `meta-llama/Meta-Llama-3.1-70B-Instruct` |
| `meta-llama/Llama-3.2-1B` | `meta-llama/Llama-3.2-1B-Instruct` |
| `meta-llama/Llama-3.2-3B` | `meta-llama/Llama-3.2-3B-Instruct` |
| | `meta-llama/Llama-3.3-70B-Instruct` |
| Phi-4 Family (Microsoft Research, 2024) | |
| | `microsoft/phi-4` |
| Qwen2.5 Family (Qwen Team, 2025) | |
| `Qwen/Qwen2.5-0.5B` | `Qwen/Qwen2.5-0.5B-Instruct` |
| `Qwen/Qwen2.5-1.5B` | `Qwen/Qwen2.5-1.5B-Instruct` |
| `Qwen/Qwen2.5-3B` | `Qwen/Qwen2.5-3B-Instruct` |
| `Qwen/Qwen2.5-7B` | `Qwen/Qwen2.5-7B-Instruct` |
| `Qwen/Qwen2.5-14B` | `Qwen/Qwen2.5-14B-Instruct` |
| `Qwen/Qwen2.5-32B` | `Qwen/Qwen2.5-32B-Instruct` |
| `Qwen/Qwen2.5-72B` | `Qwen/Qwen2.5-72B-Instruct` |
| Falcon-3 Model Family (Technology Innovation Institute, 2024) | |
| | `tiiuae/Falcon3-1B-Instruct` |
| `tiiuae/Falcon3-7B-Base` | `tiiuae/Falcon3-7B-Instruct` |
| `tiiuae/Falcon3-10B-Base` | `tiiuae/Falcon3-10B-Instruct` |

## B.11. Prompts

### B.11.1. LM-JUDGE PROMPT WITHOUT REFERENCE ANSWER

```
Prompt for Free-Form Evaluation

Your task is to judge whether the given response to a question is correct or
not.  You are only given a question and the response you are judging.

Possible judgments:
"0":  The response is incorrect.
"1":  The response is correct.

Question:  "[Insert the question here]"
Response:  "[Insert the response here]"

To the best of your knowledge:  Does the provided response answer the
question correctly?  This is part of an automated evaluation process,
therefore you must only output a single word:  "0" or "1".  Do not justify
your decision.

Evaluation (0/1):
```

### B.11.2. LM-JUDGE PROMPT WITH MCQ OPTIONS

```
Prompt for Free-Form Evaluation with Access to MCQ Reference Options

Your task is to judge whether the given response to a question is correct or
not.  You are given a question, a ground truth response, incorrect options
and the response you are judging.

Possible judgments:
"0":  The response is incorrect.  It does not match the ground-truth answer
or is more similar to any of the incorrect options than to the ground-truth
answer.
"1":  The response is correct.  It matches the ground-truth.

Question:  "[Insert the question here]"
Ground truth:  "[Insert the ground-truth option here]"
Incorrect option (1):  "[Insert the 1st wrong option here]"
...
Incorrect option (9):  "[Insert the 9th wrong option here]"
Response:  "[Insert the response here]"

To the best of your knowledge:  Does the provided response answer the
question correctly, taking the ground-truth and wrong answer options into
account?  This is part of an automated evaluation process, therefore you
must only output a single word:  "0" or "1".  Do not justify your decision.

Evaluation (0/1):
```

### B.11.3. ORIGINAL MCQ CoT PROMPT

We describe how an original MCQ prompt on MMLU-Pro is transformed into an open-style prompt for free-form inference without access to the reference options. The original chain-of-thought (CoT) prompt consists of general information about

the task, a few-shot list of questions-answer pairs and finally the actual question that is to be solved.

Each question is preceded by the keyword "Question:", followed by the question text and the list of answer options. Every option text is marked with a letter. Next, a reference chain-of-thought is given after the key-phrase "Answer: Let's think step by step" to provide an in-context example on how to solve related questions. This CoT can include references to the answer options. The CoT answer ends with the key-phrase "The answer is (X)" where "X" is the letter of the correct option. The phrase nudges the evaluated LM to answer in the same way, allowing to extract the final response using regex matching. The number of in-context examples depends on the `--num_fewshot` parameter. In our experiment, we use five examples, but for reasons of brevity, only a single one is part of the example prompt below. Finally, the phrase that starts a CoT is repeated right before the model's response.

We automatically transform these MCQ into OSQ CoT prompts. The general information is slightly adjusted to indicate the type of task. All key-phrases remain the same. We completely omit the MCQ options at the end behind the question. Any reference to an option in the chain-of-thought is replaced with the option text itself – e.g. "(G)" is replaced with the corresponding "(The second and third pharyngeal arches)". This includes the final response: "The answer is (XYZ).". Our experiments have shown that even the smallest models evaluated are able to follow these instructions and provide free-form responses that can be automatically extracted in the vast majority of cases.

---

**Few-shot CoT MCQ Prompt**

```
The following are multiple choice questions (with answers) about health.
Think step by step and then finish your answer with ̈the answer is (X) ̈where X
is the correct letter choice.

Question:  What is the embryological origin of the hyoid bone?
Options:
A. The third and fourth pharyngeal arches
B. The fourth pharyngeal arch
C. The third pharyngeal arch
D. The second pharyngeal arch
E. The second, third and fourth pharyngeal arches
F. The first pharyngeal arch
G. The second and third pharyngeal arches
H. The first and third pharyngeal arches
I. The first, second and third pharyngeal arches
J. The first and second pharyngeal arches

Answer:  Let's think step by step.  We refer to Wikipedia articles on
anatomy for help.  Let's solve this problem step by step.  The hyoid bone,
which is also known as the hyooid, is a a small U-shaped bone located in the
anterior neck.  In its resting position, it lies between the base of the
mandible and the third cervical vertebrae.  We know that the second and the
third pharyngeal arches give rise to the horns of the hyoid bone; therefore,
the embryological origin of the hyoid bone are the second and the third
pharyngeal arches|this information is covered in option (G). Therefore, we
conclude that (G) must be the correct answer.  The answer is (G)

Question:  ...

Question:  Which disease do polyomaviruses predominantly cause?
Options:
A. Tumours
B. Brain pathology
C. No disease at all
D. Kidney infections

Answer:  Let's think step by step.
```

B.11.4. Open-style CoT Prompt

---

**Few-shot CoT OSQ Prompt**

```
The following are multiple choice questions (with answers) about health.
Think step by step and then finish your answer with "the answer is (X)" where X
is the correct letter choice.

Question:  What is the embryological origin of the hyoid bone?

Answer:  Let's think step by step.  We refer to Wikipedia articles on
anatomy for help.  Let's solve this problem step by step.  The hyoid bone,
which is also known as the hyooid, is a a small U-shaped bone located in
the anterior neck.  In its resting position, it lies between the base of
the mandible and the third cervical vertebrae.  We know that the second
and the third pharyngeal arches give rise to the horns of the hyoid bone;
therefore, the embryological origin of the hyoid bone are the second and the
third pharyngeal arches|this information is covered in option (The second
and third pharyngeal arches).  Therefore, we conclude that (The second and
third pharyngeal arches) must be the correct answer.  The answer is (The
second and third pharyngeal arches)

Question:  ...

Question:  Which disease do polyomaviruses predominantly cause?

Answer:  Let's think step by step.
```

---

These are the two prompts used for coarse and fine-grained filtering to get the OSQ version of MMLU-Pro. They almost exactly match the original ones provided by Myrzakhan et al. (2024), but we performed minimal adjustments to make them more suitable to MMLU-Pro.

B.11.5. COARSE FILTERING PROMPT

---

**Coarse Prompt**

Your task is to review a series of multiple-choice questions and evaluate
their ability to be answered without the provided answer choices.

For questions that begin with an incomplete sentence (e.g., "During
swallowing, ..."), use your knowledge to attempt to complete the sentence
accurately.  For direct questions that ask for specific information or
identification (e.g., "Which of the following structures is part of the
small intestine?"), assess whether the question is formulated clearly enough
that an informed answer can be given without seeing the multiple-choice
options.  For mathematical or analytical questions (e.g., "Find all cosets
of the subgroup 4Z of 2Z"), determine if the question provides enough
context and information for a solution to be formulated without additional
options.

Please follow this format for your evaluation:

QUESTION: [Insert the question here]

VERDICT: Respond with "YES" if the question is clear and can be directly
answered based on its content alone, or "NO" if it relies on the answer
choices to be understood or answered.  Your response should include only
the verdict without any justification or reasoning.

---

B.11.6. FINE-GRAINED FILTERING PROMPT

---

**Fine-grained Prompt**

You will assign a numerical score from 1 to 10 based on how confidently it can be answered without the choices. The scoring criteria are as follows:

1: The question is entirely dependent on its choices for an answer, making it impossible to answer without them. Example: 'Which of the following statements is correct?'

10: The question can be easily and confidently answered based solely on the question stem, without any need to refer to the provided options. Example: 'What is the first law of thermodynamics in physics?'

Intermediate Scores:

2-4: The question stem gives very little information and is highly reliant on the choices for context. Example: 'Which of these is a prime number?' 'The ________ perspective on sustainability resulted from growth models that analysed the carrying capacity of the planet, overall concluding that the finite capacity of the earth and ______, _______ and ______ by current and past generations could reduce quality of life for future generations.'
5: The question provides some context or information, that gives a moderate possibility to answer the question. Example: 'Which of the following best describes the structure that collects urine in the body?'
6: The question provides a good amount of context or information, that gives a moderate possibility to answer the question. Example: 'Statement 1 | A factor group of a non-Abelian group is non-Abelian. Statement 2 | If K is a normal subgroup of H and H is a normal subgroup of G, then K is a normal subgroup of G.'
7: The question provides a good amount of context or information, that gives a high possibility to answer the question. Example: 'The element (4, 2) of $Z_{12}$ x $Z_8$ has order'
8-9: The question provides a good amount of context or information, that gives a high possibility to answer the question. Example: 'A "dished face" profile is often associated with'

ONLY GIVE THE VALUE BETWEEN 1-10 AS YOUR ANSWER. DO NOT INCLUDE ANY OTHER INFORMATION IN YOUR RESPONSE.

---

# C. Weak-to-Strong Training

## C.1. Setup

Table 9. Datasets, Weak Models and Strong Models Used in the Weak to Strong Experiments.

| Models | Datasets |
|---|---|
| **Weak Models** | `sciq` (Welbl et al., 2017) |
| `google/gemma-2-2b` (Gemma Team, 2024) | `anli-r2` (Nie et al., 2020) |
| `Qwen/Qwen2.5-1.5B` (Qwen Team, 2025) | `boolq` (Clark et al., 2019) |
| `meta-llama/Llama-3.2-1B` (Llama Team, 2024a) | `cola` (Warstadt et al., 2019) |
| `microsoft/phi-2` (Li et al., 2023) | `ethics-utilitarianism` (Hendrycks et al., 2021a) |
| **Strong Models** | `sst2` (Socher et al., 2013) |
| `google/gemma-2-9b` | `twitter-sentiment` (Zhang et al., 2019) |
| `Qwen/Qwen2.5-7B` | `dream` (Sun et al., 2019) |
| `meta-llama/Llama-3.1-8B` | `mc-taco` (Zhou et al., 2019) |
| | `multirc` (Khashabi et al., 2018) |
| | `quail` (Rogers et al., 2020) |
| | `quartz` (Tafjord et al., 2019) |
| | `social-i-qa` (Sap et al., 2019) |
| | `wic` (Pilehvar & Camacho-Collados, 2019) |
| | `cosmos-qa` (Huang et al., 2019) |

We follow the weak to strong generalization setup proposed in Burns et al. (2024), focusing on NLP tasks. The original paper reported results with GPT (Radford et al., 2019) model versions. Instead, we use larger, more capable and recent open-weight models to make observations at the frontier. For this, we used the codebase of Scherlis et al. (2024) that uses open-weight models on Huggingface instead. We now describe the full setup here.

The setup uses a pretrained weak base model $W$, a pretrained strong base model $S$ and a dataset $D$, where $D_{tr}, D_{val}, D_{te}$ are the training (10,000 samples), validation (1,000 samples) and test (5,000 samples) datasplits respectively. $D_{tr}$ is divided into two halves, independently assigning each sample to $D_{tr1}, D_{tr2}$ with 50% probability each. All the datasets studied convert standard NLP MCQ datasets into binary classification, by randomly sampling one of the wrong options. Predictions $\geq 0.5$ are considered as class 1, and $< 0.5$ as class 0. We highlight the models and datasets used in our study in Table 9.

First, the weak base model $W$ is finetuned on ground-truth labels in $D_{tr1}$ to obtain the weak supervisor $W_s$. In the original setup, this is meant to simulate a human that is an expert at the given task. Then, $W_{gt}$ annotates samples in $D_{tr2}$, and the strong student model $S$ is finetuned on these annotations to obtain the Weak to Strong trained model $S_{w2s}$. In the original setup, the strong base model simulates a future model with superhuman intelligence, but not finetuned for specific domain knowledge.

**Finetuning Methodology**: For the above finetuning steps we use Low Rank Adapters (LoRA) (Hu et al., 2022) due to budget constraints, and train a binary classifier the same way as Scherlis et al. (2024). We use the confidence weighted loss proposed by Burns et al. (2024). This loss encourages the strong model's predictions to align with both a weaker model and its own "hardened" predictions. The hardened predictions are derived by thresholding the strong model's output. The loss function is defined as:

$$\mathcal{L}(f) = (1 - \alpha) \cdot \text{CE}(f(x), f_w(x)) + \alpha \cdot \text{CE}(f(x), \hat{f}(x)) \tag{19}$$

where $f(x)$ is the strong model's output, $f_w(x)$ is the weak model's output, $\hat{f}(x) = \mathbb{I}[f(x) > t]$ represents the hardened predictions using an adaptive threshold $t$, and $\alpha$ is a weight that increases over the initial phase of training.

Following Scherlis et al. (2024) we use a cosine learning rate schedule, with 40 warmup steps, the learning rates for the weak, strong model are $5 \times 10^{-4}, 8 \times 10^{-5}$ respectively, and we train for 3 epochs which is sufficient for the train and validation loss to stabilize.

**Weak to Strong Gain Metric**: We wish to study the gain achieved from weak to strong training for the strong student model. To characterize the initial accuracy of the strong student model, we train a binary classifier head to obtain $S_b$. The

weak to strong gain is then quantified as:

$$Acc(S_{w2s}) - Acc(S_b) \tag{20}$$

Note that this is different from the PGR metric reported by Burns et al. (2024). Their goal was to show weak to strong training can make the strong student cross the accuracy of the weak supervisor. Thus, they measured accuracy gained over the weak supervisor $Acc(S_{w2s}) - Acc(W_{gt})$, normalizing it by an "upper-bound" obtained by training the strong student on ground-truth labels on $D_{tr2}$, giving $PGR = \frac{Acc(S_{w2s}) - Acc(W_{gt})}{Acc(S_{gt}) - Acc(W_{gt})}$. In our work, we show that leveraging complementary knowledge effectively might actually allow $S_{w2s} > S_{gt}$, questioning their "upper-bound". Thus we stick to reporting how much the student model improved as described in Equation 20.

**Similarity vs Weak to Strong Gain**: In Figure 16 we reported weak-to-strong gain (Equation 20) on the Y-axis, and similarity ($\kappa_p$) on the X-axis. We group by model pairs, and vary the task within each model pair for the linear fit. Figure 16 shows the same scatter points but colored based on the dataset. This shows that weak-to-strong gain is consistently higher for tasks where models are less similar, and how similar two models are depends mostly on the task, i.e. there is not much variance in similarity across the model pairs for a fixed task.

**Discarded Results**: We had initially run experiments with three more weak models: SmolLM 1.7B, Qwen-2.5-0.5B, Llama-3.2-1B against the same list of strong models reported above. However, we found that on some tasks, the weak-to-strong gain was negative. The weak supervisor ($W_{gt}$) models had lower accuracy compared to the strong student $S_b$, leading to a decrease in accuracy for the strong student after weak to strong training. We thus removed these weak models from our analysis. Similarly, we had also tried the Hellaswag dataset, but found that both weak and strong models had very low accuracies, often below 60% where chance is 50% for binary classification, consistent with Scherlis et al. (2024), and decided to not include it in our analysis.

### C.2. Elicitation vs Complementary Knowledge

*Table 10.* **Models and Sources of Knowledge in Complementary Knowledge vs Elicitation Comparison**.

| Model | Ground-truth labels in $D_{tr1}$ | Latent Knowledge of $W$ | Latent Knowledge of $S$ |
|---|:---:|:---:|:---:|
| $W_{gt}$ | ✓ | ✓ | ✗ |
| $S_{gt}$ | ✓ | ✗ | ✓ |
| $S_{w2s}$ | ✓[4] | ✓ | ✓ |

Figure 4 points to the fact that similarity or difference between the weak supervisor and the initial strong student are strong predictors of weak-to-strong gain. However, the initially proposed explanation of weak-to-strong generalization is "elicitation", i.e. the strong student has latent capabilities that are brought out by finetuning on weak annotations (Burns et al., 2024). To quantify the contribution of these two sources for weak-to-strong gain, elicitation and complementary knowledge, we establish the following setup.

First, our functional similarity metric cannot capture latent knowledge in the strong student's representations. For this, we follow Burns et al. (2024) and finetune the strong student $S$ on ground-truth labels of $D_{tr1}$ to obtain the elicited strong student model $S_{gt}$. Note that we use $D_{tr1}$ instead of $D_{tr2}$ here so that the training set of $S_{w2s}$, $D_{tr2}$, remains held-out, and we can analyze the relative effect of eliciation and complementary knowledge on both the train and test set.

Table 10 summarizes sources of knowledge for the weak supervisor $W_{gt}$, strong elicited $S_{gt}$, and weak-to-strong trained student $S_{w2s}$ in our setup. $S_{w2s}$ benefits from the latent knowledge of $S$, complementary knowledge transfer of latent knowledge of $W$, and distillation of knowledge in $D_{tr1}$ from $W_{gt}$. It learns imperfectly from all three sources of knowledge. Given this, we now discuss how Figure 5 compares elicitation and complementary knowledge transfer:

- **Bottom-Left = Elicitation**: $W_{gt}$ does not benefit from latent knowledge of $S$, so $S_{w2s}$ accuracy on samples where it is wrong but $S_{gt}$ is correct signify knowledge that could only be from elicitation.

- **Top-Right = Complementary Knowledge Transfer**: $S_{gt}$ does not benefit from the latent knowledge of $W$, so $S_{w2s}$ accuracy on samples where it is wrong but $W_{gt}$ is correct signify knowledge that could only be from complementary knowledge transfer.

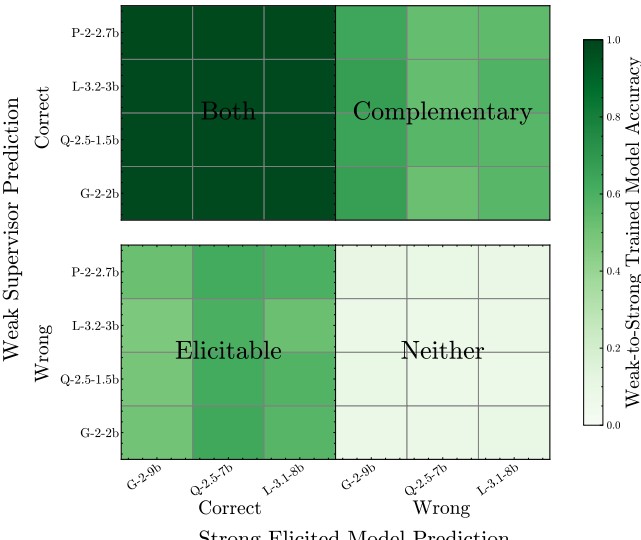

*Figure 15.* We decompose the accuracy of the weak to strong trained model on four parts of the train data distribution based on whether the weak supervisor and an oracle strong elicited model (using ground-truth annotations) are correct or wrong. All results are averaged over 15 datasets. Sub-rectangles represent weak, strong model pairs. On the train dataset, complementary knowledge transfer (mean accuracy 0.59) plays an equal role as elicitation (mean accuracy 0.56).

- **Top-Left = Could be Both**: Accuracy of $S_{w2s}$ on samples where both $W_{gt}, S_{gt}$ are correct could come from both their latent knowledge, and the ground-truth annotations in $D_{tr1}$. Thus, these could be both elicitation and complementary knowledge transfer, or also learning from the finetuning data.

- **Bottom-Right = Random flips**: We find that $10\%$ predictions can flip even when finetuning $W, S$ on ground-truth labels from $D_{tr2}$ instead of $D_{tr1}$, which were split into two halves at random from $D_{tr}$. Thus, the roughly $10\%$ accuracy on samples that both $W_{gt}, S_{gt}$ got wrong could just be random flips to the correct prediction (since its a binary classification setting).

**Behavior on the Train Set**: Figure 15 reports the same comparison of elicitation and complementary knowledge transfer but on $D_{tr2}$ on which the weak-to-strong training occurs. This set is unseen for both the weak supervisor $W_{gt}$ and the strong elicited model $S_{gt}$. We find that in fitting the training data complementary knowledge transfer plays an equal or bigger role than elicitation. This is to be expected as $S_{w2s}$ is trained by fitting on $W_{gt}$'s annotations of $D_{tr2}$. The weak-to-strong trained student however still generalizes more similarly to the strong elicited model than the weak supervisor, though complementary knowledge transfer is also visible on test set predictions as seen in Figure 5.

### C.3. Effect of Different similarity metrics

We now report similarity vs weak-to-strong gain for various alternate similarity metrics. Here, we color the scatter points by dataset instead of model pair, and fit a single line, for ease of interpretation. We report the following similarity metrics:

- Error Consistency - In this setting of binary classification, this is equivalent to the non-probabilistic version of CAPA, as there is only one incorrect option so models cannot disagree when both are incorrect on a sample.

- CAPA ($\kappa_p$) - Our metric which incorporates probabilistic information into error consistency.

- $1 - JSD$ - Since Jensen-Shannon Distance measures difference between distributions and is normalized between 0 and 1, by subtracting it from 1 we can obtain a similarity metric for ease of comparison with the previous metrics.

In Figure 16 we see that all metrics can show the same trend, that is, tasks where models differ more have larger gain from weak-to-strong training. The highest correlation is seen for CAPA, though in the binary classification setup of weak-to-strong generalization the probabilistic information does not add much value compared to error consistency. $1 - JSD$ gives a more noisy scatter plot, with lower correlation ($r$).

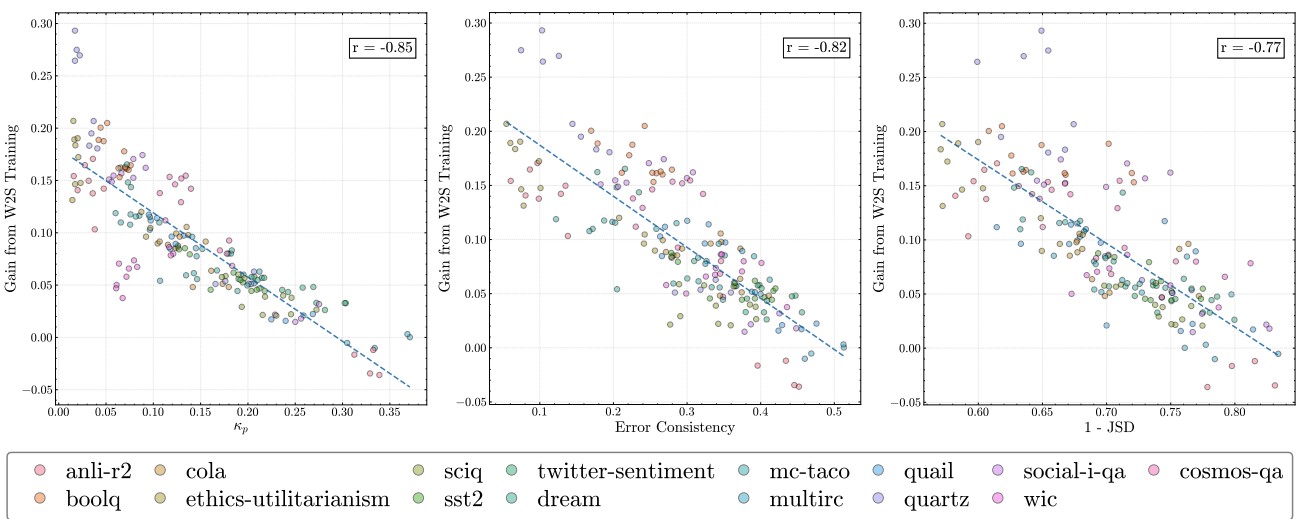

*Figure 16.* **Various Similarity Metrics vs Weak-to-Strong gain**. The highest correlation is seen for CAPA $\kappa_p$, though in the binary classification setup of weak-to-strong generalization the probabilistic information does not add much value compared to error consistency. $1 - JSD$ gives a more noisy scatter plot, with lower correlation ($r$).

### C.4. Accuracies in Weak-to-Strong training

In Figure 17 we report average across the 12 model pairs for all 15 datasets. Consistently, the ordering is as follows: the initial strong student has lower accuracy than the weak supervisor, but surpasses it after weak-to-strong training. However, it is not able to match the performance ceiling of ground-truth elicitation. Finally, if the take a union over the correct predictions of the weak supervisor and strong elicited model, the performance ceiling can be even higher.

### C.5. Weak-to-strong Accuracy Value Details in Elicitation vs Complementary Knowledge Analysis

In Table 11 and Table 12 we report the underlying numbers for Figure 5 and Figure 15 respectively. The astute observer may be confused about the around 12% accuracy on the test set when when both the weak supervisor and strong elicited model are wrong (bottom-right quadrant). We find that merely finetuning on a different random subset of training data leads to around 11% predictions being flipped. Thus, much of this accuracy could just be due to random chance because of the binary classification setup. This also indicates that complementary knowledge transfer explains much of the beyond-chance accuracy not accounted for by elicitation.

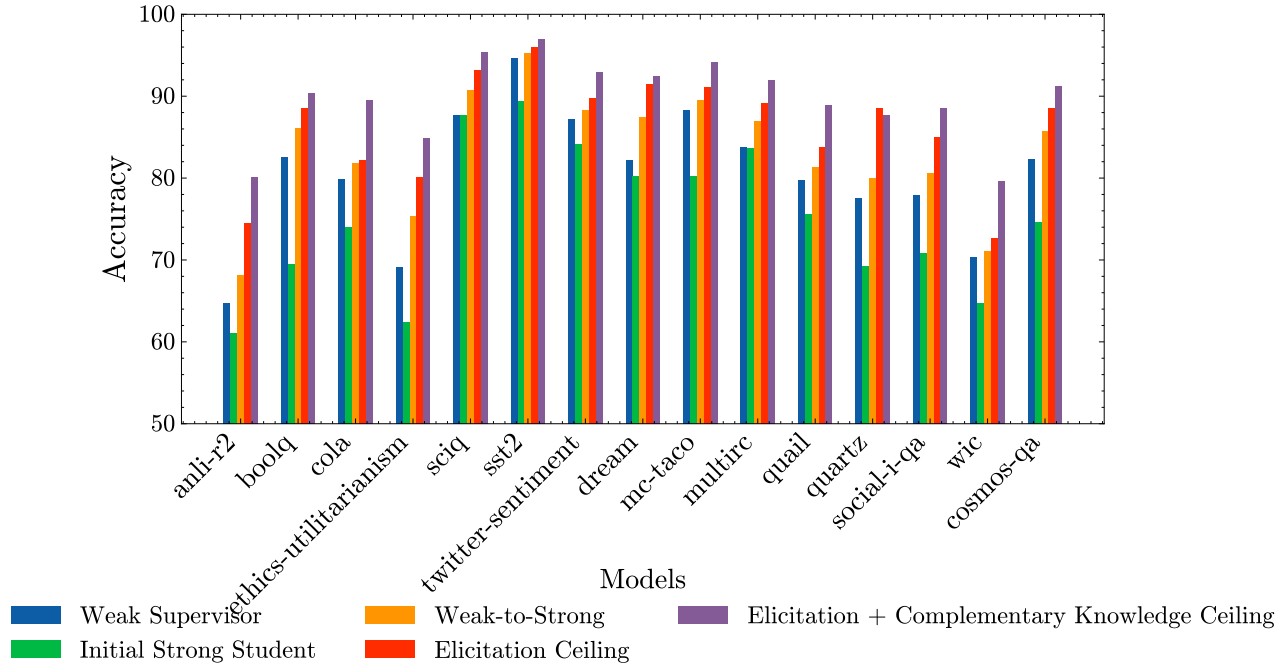

*Figure 17.* **Test Accuracies for various models and ceiling estimates in Weak-to-Strong training**. The accuracies are averaged over 12 model pairs. The initial strong student model has consistently lower accuracy than the weak supervisor consistent with Burns et al. (2024); Scherlis et al. (2024). The weak-to-strong trained student surpasses the weak-supervisor across datasets. However it has lower accuracy than the elicitation ceiling which trains the strong student on ground-truth annotations. Finally, our new estimated ceiling which incorporates the complementary knowledge of the weak supervisor has even higher accuracies, showing even more scope for improvements.

*Table 11.* **Weak-to-strong trained model's accuracies on four parts of the test data distribution, based on relative mistakes of weak-supervisor, strong elicited model**. This table reports the underlying numbers for Figure 5, with accuracy averaged across the 15 datasets studied for each model pair. We see that the weak-to-strong model is almost always correct when both the weak-supervisor, strong elicited model are correct. It is more correct when the strong elicited model is correct and the weak-supervisor is wrong than vice-versa. This indicates weak-to-strong training currently exploits more of the possible gains from elicitation, but less of the possible gains from complementary knowledge transfer.

| Common Knowledge | | Complementary Knowledge Transfer | |
|---|---|---|---|
| **Pair** | **Acc (%)** | **Pair** | **Acc (%)** |
| (gemma-2-2b, gemma-2-9b) | 97.4 | (gemma-2-2b, gemma-2-9b) | 45.2 |
| (gemma-2-2b, Qwen2.5-7B) | 97.1 | (gemma-2-2b, Qwen2.5-7B) | 34.9 |
| (gemma-2-2b, Llama-3.1-8B) | 97.0 | (gemma-2-2b, Llama-3.1-8B) | 40.1 |
| (Qwen2.5-1.5B, gemma-2-9b) | 97.1 | (Qwen2.5-1.5B, gemma-2-9b) | 47.1 |
| (Qwen2.5-1.5B, Qwen2.5-7B) | 97.4 | (Qwen2.5-1.5B, Qwen2.5-7B) | 36.9 |
| (Qwen2.5-1.5B, Llama-3.1-8B) | 96.5 | (Qwen2.5-1.5B, Llama-3.1-8B) | 39.6 |
| (Llama-3.2-3B, gemma-2-9b) | 97.6 | (Llama-3.2-3B, gemma-2-9b) | 46.2 |
| (Llama-3.2-3B, Qwen2.5-7B) | 97.5 | (Llama-3.2-3B, Qwen2.5-7B) | 36.2 |
| (Llama-3.2-3B, Llama-3.1-8B) | 97.3 | (Llama-3.2-3B, Llama-3.1-8B) | 41.7 |
| (phi-2, gemma-2-9b) | 97.3 | (phi-2, gemma-2-9b) | 50.2 |
| (phi-2, Qwen2.5-7B) | 97.3 | (phi-2, Qwen2.5-7B) | 44.2 |
| (phi-2, Llama-3.1-8B) | 97.4 | (phi-2, Llama-3.1-8B) | 44.2 |
| Elicitation | | Both Wrong | |
| **Pair** | **Acc (%)** | **Pair** | **Acc (%)** |
| (gemma-2-2b, gemma-2-9b) | 71.0 | (gemma-2-2b, gemma-2-9b) | 12.9 |
| (gemma-2-2b, Qwen2.5-7B) | 75.0 | (gemma-2-2b, Qwen2.5-7B) | 10.7 |
| (gemma-2-2b, Llama-3.1-8B) | 72.3 | (gemma-2-2b, Llama-3.1-8B) | 13.0 |
| (Qwen2.5-1.5B, gemma-2-9b) | 69.4 | (Qwen2.5-1.5B, gemma-2-9b) | 11.4 |
| (Qwen2.5-1.5B, Qwen2.5-7B) | 73.3 | (Qwen2.5-1.5B, Qwen2.5-7B) | 11.6 |
| (Qwen2.5-1.5B, Llama-3.1-8B) | 72.1 | (Qwen2.5-1.5B, Llama-3.1-8B) | 13.5 |
| (Llama-3.2-3B, gemma-2-9b) | 71.0 | (Llama-3.2-3B, gemma-2-9b) | 12.5 |
| (Llama-3.2-3B, Qwen2.5-7B) | 77.2 | (Llama-3.2-3B, Qwen2.5-7B) | 11.2 |
| (Llama-3.2-3B, Llama-3.1-8B) | 73.4 | (Llama-3.2-3B, Llama-3.1-8B) | 13.8 |
| (phi-2, gemma-2-9b) | 67.9 | (phi-2, gemma-2-9b) | 12.3 |
| (phi-2, Qwen2.5-7B) | 71.1 | (phi-2, Qwen2.5-7B) | 11.6 |
| (phi-2, Llama-3.1-8B) | 69.0 | (phi-2, Llama-3.1-8B) | 11.5 |

*Table 12.* **Weak-to-strong trained model's accuracies on four parts of the train data distribution, based on relative mistakes of weak-supervisor, strong elicited model**. This table reports the underlying numbers for Figure 15, with accuracy averaged across the 15 datasets studied for each model pair. On the train distribution, the weak-to-strong model is almost equally correct on the only-elicitable and only learnable from complementary knowledge samples, with a slight lean towards the latter. Yet, Table 11 showed it generalizes more similarly to the strong elicited model.

| Common Knowledge | | Complementary Knowledge Transfer | |
|---|---|---|---|
| **Pair** | **Acc (%)** | **Pair** | **Acc (%)** |
| (gemma-2-2b, gemma-2-9b) | 98.6 | (gemma-2-2b, gemma-2-9b) | 66.8 |
| (gemma-2-2b, Qwen2.5-7B) | 98.6 | (gemma-2-2b, Qwen2.5-7B) | 52.5 |
| (gemma-2-2b, Llama-3.1-8B) | 98.5 | (gemma-2-2b, Llama-3.1-8B) | 56.8 |
| (Qwen2.5-1.5B, gemma-2-9b) | 98.5 | (Qwen2.5-1.5B, gemma-2-9b) | 65.2 |
| (Qwen2.5-1.5B, Qwen2.5-7B) | 98.6 | (Qwen2.5-1.5B, Qwen2.5-7B) | 56.9 |
| (Qwen2.5-1.5B, Llama-3.1-8B) | 98.5 | (Qwen2.5-1.5B, Llama-3.1-8B) | 57.0 |
| (Llama-3.2-3B, gemma-2-9b) | 98.7 | (Llama-3.2-3B, gemma-2-9b) | 67.2 |
| (Llama-3.2-3B, Qwen2.5-7B) | 98.7 | (Llama-3.2-3B, Qwen2.5-7B) | 53.8 |
| (Llama-3.2-3B, Llama-3.1-8B) | 98.5 | (Llama-3.2-3B, Llama-3.1-8B) | 58.8 |
| (phi-2, gemma-2-9b) | 98.0 | (phi-2, gemma-2-9b) | 64.4 |
| (phi-2, Qwen2.5-7B) | 98.4 | (phi-2, Qwen2.5-7B) | 53.7 |
| (phi-2, Llama-3.1-8B) | 98.2 | (phi-2, Llama-3.1-8B) | 54.8 |

| Elicitation | | Both Wrong | |
|---|---|---|---|
| **Pair** | **Acc (%)** | **Pair** | **Acc (%)** |
| (gemma-2-2b, gemma-2-9b) | 50.7 | (gemma-2-2b, gemma-2-9b) | 8.6 |
| (gemma-2-2b, Qwen2.5-7B) | 63.5 | (gemma-2-2b, Qwen2.5-7B) | 8.3 |
| (gemma-2-2b, Llama-3.1-8B) | 57.3 | (gemma-2-2b, Llama-3.1-8B) | 9.5 |
| (Qwen2.5-1.5B, gemma-2-9b) | 49.2 | (Qwen2.5-1.5B, gemma-2-9b) | 9.6 |
| (Qwen2.5-1.5B, Qwen2.5-7B) | 62.2 | (Qwen2.5-1.5B, Qwen2.5-7B) | 7.4 |
| (Qwen2.5-1.5B, Llama-3.1-8B) | 58.2 | (Qwen2.5-1.5B, Llama-3.1-8B) | 7.7 |
| (Llama-3.2-3B, gemma-2-9b) | 47.9 | (Llama-3.2-3B, gemma-2-9b) | 8.8 |
| (Llama-3.2-3B, Qwen2.5-7B) | 60.3 | (Llama-3.2-3B, Qwen2.5-7B) | 7.9 |
| (Llama-3.2-3B, Llama-3.1-8B) | 52.4 | (Llama-3.2-3B, Llama-3.1-8B) | 8.5 |
| (phi-2, gemma-2-9b) | 52.6 | (phi-2, gemma-2-9b) | 10.8 |
| (phi-2, Qwen2.5-7B) | 62.3 | (phi-2, Qwen2.5-7B) | 9.4 |
| (phi-2, Llama-3.1-8B) | 60.1 | (phi-2, Llama-3.1-8B) | 9.1 |

# D. Similarity Trends with Increasing Capabilities

## D.1. Setup Details

We utilize two prominent benchmark datasets from the OpenLLM leaderboard to explore the relationship between model similarity and capability: MMLU Pro and BBH. For the BBH dataset, we aggregate 23 distinct tasks that can be studied as multiple-choice questions from the Big-Bench Hard benchmark to ensure that each model is evaluated on sufficient questions, thereby ensuring statistically significant results. The MMLU Pro dataset consists of MCQs across 14 different subjects, with varying numbers of options per question. Notably, some questions are repeated with shuffled option orders. To maintain consistency, we filter the dataset by retaining only those questions for which both the question text and the correct option index remain consistent across all models. This filtering process yields a refined dataset of 11,828 questions.

To analyze trends across model capabilities, we divide 130 models (Table 14) into five bins based on their individual accuracy percentiles. This binning strategy is followed for all experimental setups and ensures an approximately equal distribution of models per bin, maintaining a consistent sample size across bins. We select model pairs within each bin and compute their similarity and average accuracy. This approach ensures that the average accuracy of the pairs remains representative of the individual model accuracies within the bin. We do not consider model pairs from the same family to avoid confounding effects of model similarity being attributed to model family rather than the capability.

## D.2. Why are model mistakes becoming more similar? A preliminary analysis

### D.2.1. INSTRUCTION-TUNING EXACERBATES THE TREND

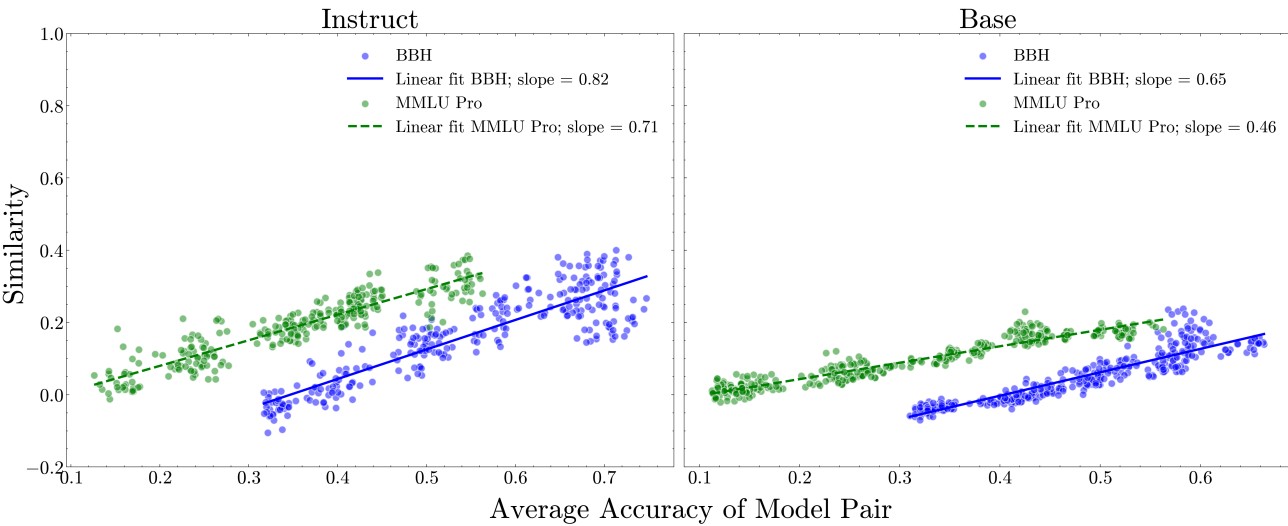

*Figure 18.* **LM Similarity ($\kappa_p$) vs Capabilities in Instruct-tuned and Base models on MMLU pro and BBH**. After applying the same model binning stratergy and pairwise similarity, a steeper trend is observed in the instruct-tuned models compared to base models for both datasets.

Instruction-tuned models are base models that have been fine-tuned on instruction datasets and their corresponding outputs, enhancing their ability to follow user instructions accurately. Among the models analyzed for the capability-similarity trend, we categorize them into instruction-tuned and base models. Using the same binning strategy as discussed in the previous section, we first assign all models to bins based on their accuracy percentiles. When computing pairwise similarity and accuracy, we restrict to pairs of the same model type— base-base and instruct-instruct models. As illustrated in Fig. 18, instruction-tuned model pairs exhibit a stronger similarity trend with a steeper slope compared to base models. This can likely be attributed to the fact that instruction-tuned models may have been fine-tuned on similar instruction datasets, leading to a higher similarity trend among them.

### D.2.2. IS THE TREND CONFOUNDED BY QUESTION DIFFICULTIES?

To address the potential confounder that models might exhibit higher similarity as their capability increases simply due to their inherent ability to solve harder problems, we analyze the relationship between question hardness and model similarity

on MMLU Pro and BBH. Question hardness is determined by the percentage of models that answer a question correctly, with harder questions being those that fewer models answer correctly. We split the data samples into percentile bins based on question hardness and compute the average similarity across all capability bins of the initial setting, as illustrated in Fig. 19(a).

Fig. 19(a) demonstrates that the overall average similarity remains consistent across different levels of question hardness, with only a slight increase observed for the hardest questions (100th percentile). This consistency indicates that the hardness of the questions does not significantly confound the observed trend of increasing similarity with model capability. These findings reinforce the hypothesis that the growing similarity among models is not merely a byproduct of their ability to solve harder problems but reflects a deeper trend in model behavior as their capabilities improve.

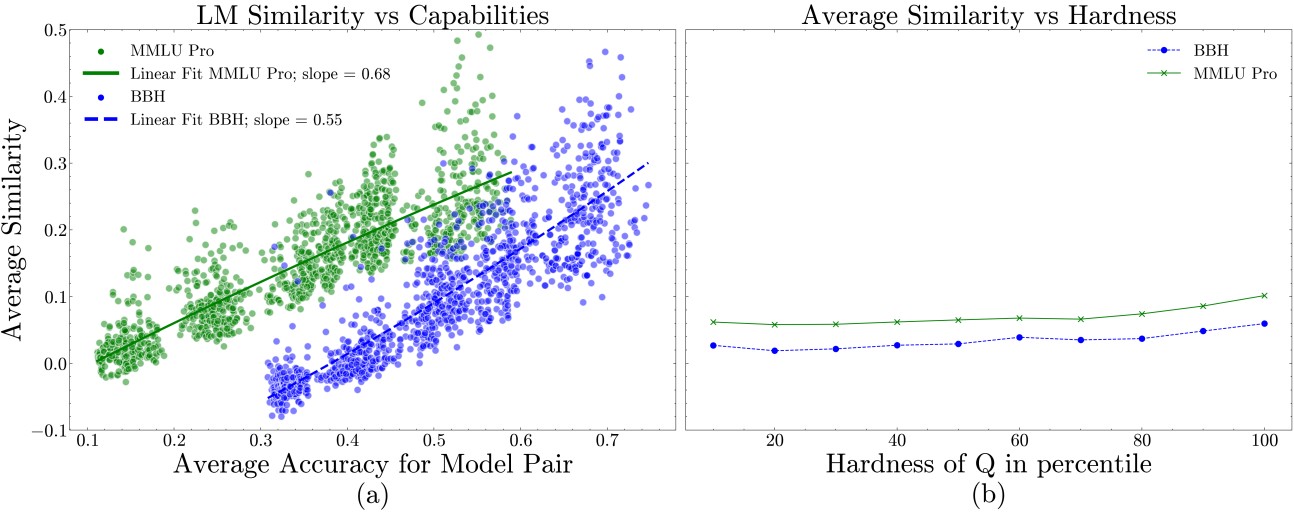

*Figure 19.* **Role of question difficulty in similarity-capability trend.** We plot in parallel (a) Scatter plot with model pairs, illustrating the increasing trend of similarity ($\kappa_p$) with model capability. (b) Average similarity ($\kappa_p$) across all capability bins for different levels of question hardness. CAPA is mostly consistent across question hardness, with a slight increase on the hardest questions. This shows that question difficulty is not a significant confounder for increasing similarity in mistakes.

### D.2.3. CAN CHANGING ARCHITECTURE REDUCE MODEL SIMILARITY?

To study the effect of model similarities across different architectures, we analyze Falcon3 Mamba 7B base and instruct models by computing their CAPA values with Falcon3 7B transformer, Mistral 7Bv0.3, and Llama 3.1 8B base and instruct models. We ensure that the accuracies of the non-Falcon3 transformers are within ±5% of the Mamba model to ensure comparable capabilities.

In Table 13, $Similarity_1$ presents the CAPA between the Falcon3 Mamba and Falcon3 Transformer, $Similarity_2$ the CAPA between Falcon3 Mamba and Llama/Mistral Transformer, and $Similarity_3$ between Falcon3 Transformer and Llama/Mistral Transformers. The results reveal that base models exhibit lower overall similarity compared to instruction-tuned models, with pairwise similarity between Falcon3 Mamba and non-Falcon Mistral/Llama Transformers showing the least similarity. Within the base model category, Falcon3 Mamba and Falcon Transformers demonstrate the highest similarity. For instruction-tuned models, Falcon3 Transformer and Mistral/Llama Transformer pairs exhibit the highest similarity, followed closely by Falcon3 Mamba and Falcon3 Transformer.

The Falcon Mamba-Falcon Transformers maintain higher similarity overall, potentially due to their shared model family, despite differences in their underlying architectures. This observation highlights that architectural differences may play a less significant role in model similarity compared to factors such as training data and fine-tuning procedures. From the earlier section, instruction-tuned models exhibit a stronger similarity trend, similar to as observed in this setting.

### D.3. Alternative Similarity Metrics

As discussed in earlier sections, multiple metrics can be employed to quantify the similarity between models, each with its own strengths and limitations. In this analysis, we evaluate several alternative metrics under the same experimental setting

*Table 13.* **Analyzing the effect of difference in architecture on CAPA** $\kappa_p$. Using base and instruct variants of Falcon3 Mamba and Falcon3 Transformer of size 7B, we compare it with transformers with similar size and accuracy from a different model family- LLama and Mistral. $Similarity_1$ consistently has an overall higher similarity due to models belonging to the same family. In instruct-tuned models, $Similarity_3$ is the highest, possibly due to the instruct-tuning.

| Falcon Mamba | Falcon Transformer | Transformer Model | $Similarity_1$ | $Similarity_2$ | $Similarity_3$ |
|---|---|---|---|---|---|
| 7B Base | 7B Base | Llama 3.1 8B | 0.0619 | 0.0167 | 0.0422 |
| | | Mistral v0.3 7B | | 0.0105 | 0.0235 |
| 7B Instruct | 7B Instruct | Llama 3.1 8B Instruct | 0.1111 | 0.0665 | 0.173 |
| | | Mistral v0.3 7B Instruct | | 0.0582 | 0.1584 |

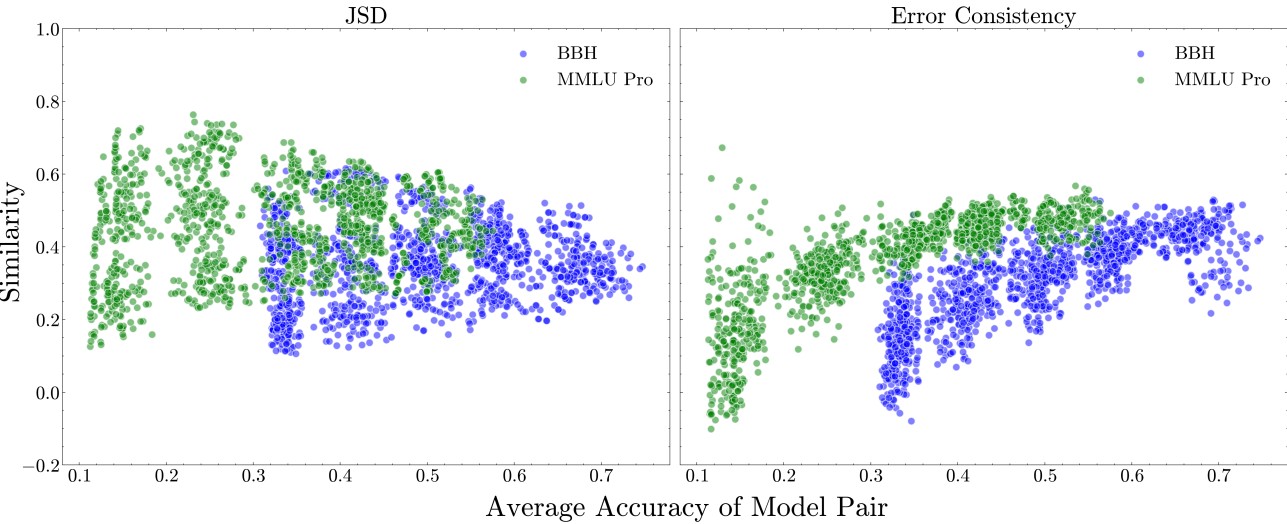

*Figure 20.* **Error consistency and JSD for model similarity on BBH and MMLU Pro.** The y-axis represents the similarity computed using JSD and Error consistency. JSD exhibits high variance and a flat trend, whereas Error Consistency shows an increasing trend with model capability, similar to the trend observed in $\kappa_p$.

used for CAPA, including the binning and averaging strategies, for the two benchmark datasets. Fig. 21a presents the results for discrete $\kappa_p$, which does not utilize logit information, while Fig. 21b demonstrates a similar trend using $\kappa_p$ for $M > 2$ ($\kappa_p$ extended for more than 2 models). Additionally, Fig. 20 includes results for Jensen-Shannon Divergence (JSD) and Error Consistency (Geirhos et al., 2020).

Discrete $\kappa_p$ exhibits an increasing trend with model capability, closely mirroring the trend observed with $\kappa_p$. Similarly, $\kappa_p$ for $M > 2$, which leverages probabilistic information, unlike Discrete $\kappa_p$, shows a strong upward trend. Unlike other metrics, $\kappa_p$ for $M > 2$ provides a direct measure of similarity that quantifies agreement among all models within a bin, eliminating the need for averaging pairwise similarities. Models of same family within the same bin are retained when computing the metric. In contrast, JSD does not exhibit a clear trend and remains flat with high variance across the capability axis. Error Consistency, however, aligns with the upward trend observed in other metrics, further supporting the hypothesis that model similarity increases with capability.

### D.4. Model capability vs similarity across domains

The scatter plot in Fig 6 shows the increasing similarity trend after aggregating across the subjects (MMLU pro) and tasks (BBH). Fig 22 and Fig 23 show the observed trend within each individual subject and task for MMLU Pro and BBH respectively.

In the MMLU Pro dataset, the trend of increasing average similarity within each bin as model capability improves is consistently observed across all individual subjects. For the BBH tasks, while the trend is not as pronounced in some tasks, it remains significant in the majority of them. This weaker trend in certain BBH tasks can be attributed to the limited number

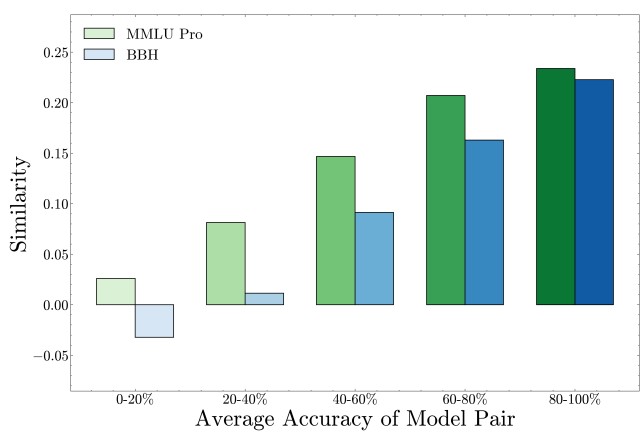 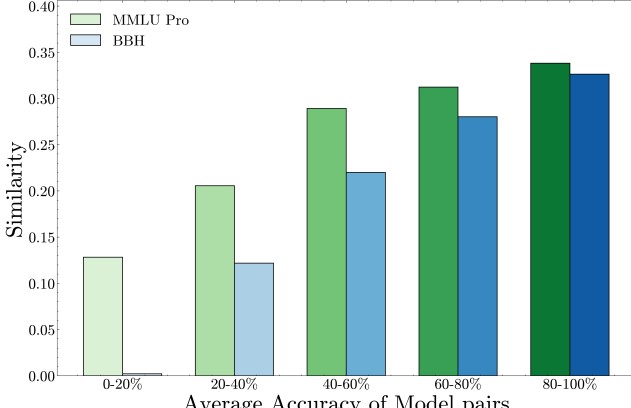

(a) **LM Similarity ($\kappa_p$ for $M > 2$) vs Average Accuracy of Model Pairs in each Capability bin**

(b) **LM Similarity (Discrete $\kappa_p$) vs Average Accuracy of Model Pairs in each Capability bin**

*Figure 21.* Discrete $\kappa_p$ and $\kappa_p$ for $M > 2$ values computed on the MMLU Pro and BBH dataset. An increasing trend in similarity is observed across both datasets in accordance with the hypothesis. Discrete $\kappa_p$ uses similar averaging idea as used in $\kappa_p$ while in $\kappa_p$ for $M > 2$, the similarity is computed using all models in a capability bin at once.

of questions per task for each model, with a maximum of 250 questions per task, which reduces the reliability of the results compared to the more robust MMLU Pro dataset. This is also visible through the high confidence interval in the BBH tasks, unlike MMLU pro subjects.

### D.5. List of models

*Table 14.* **Models from OpenLLM leaderboard used to study the capability-similarity trend**. Models across different families, architectures, size and versions were used to ensure robustness in the experimental results. The models included are both base and fine-tuned versions where available.

| Dev | Model Family | Size | Type |
|---|---|---|---|
| 01-ai | Yi-1.5 | 9B, 34B | Base, Instruct |
| | Yi | 34B | Base, Instruct |
| CohereForAI | c4ai-command-r-plus | – | Base, Instruct |
| | aya-expanse | 32b | Base |
| EleutherAI | Pythia | 160m, 410m, 2.8b, 6.9b, 12b | Base |
| Google | Gemma | 2b, 7b | Base, Instruct |
| | Gemma-1.1 | 2b, 7b | Instruct |
| | Gemma-2 | 2b, 9b, 27b | Base, Instruct |
| | Flan-T5 | Small, Base, Large, XL, XXL | Base |
| Meta | Llama-2 | 7b, 13b, 70b | Base, Instruct |
| | Llama-3.2 | 1B, 3B | Base, Instruct |
| | Llama-3 | 8B, 70B | Base, Instruct |
| | Llama-3.1 | 8B, 70B | Instruct |
| | Llama-3.3 | 70B | Instruct |
| Mistral AI | Mistral-7B | v0.1, v0.2, v0.3 | Base, Instruct |
| | Mixtral-8x7B | v0.1 | Base, Instruct |
| | Mistral-Large | – | Instruct |
| Nvidia | Mistral-NeMo-Minitron | 8B | Base, Instruct |
| Qwen | Qwen2 | 0.5B, 1.5B, 7B, 72B | Base, Instruct |
| | Qwen2.5 | 0.5B, 1.5B, 3B, 7B, 14B, 32B, 72B | Base, Instruct |
| | Qwen2-Math | 7B, 72B | Base, Instruct |
| | Qwen2-VL | 7B, 72B | Instruct |
| | Qwen2.5-Coder | 7B | Base, Instruct |
| | Qwen1.5 | 32B, 110B | Base, Chat |
| Tiiuae | Falcon | 7b, 11B, 40b | Base, Instruct |
| | Falcon3 | 7B, 10B | Base, Instruct |
| | Falcon-mamba | 7b | Base |
| Upstage | solar-pro-preview | – | Instruct |

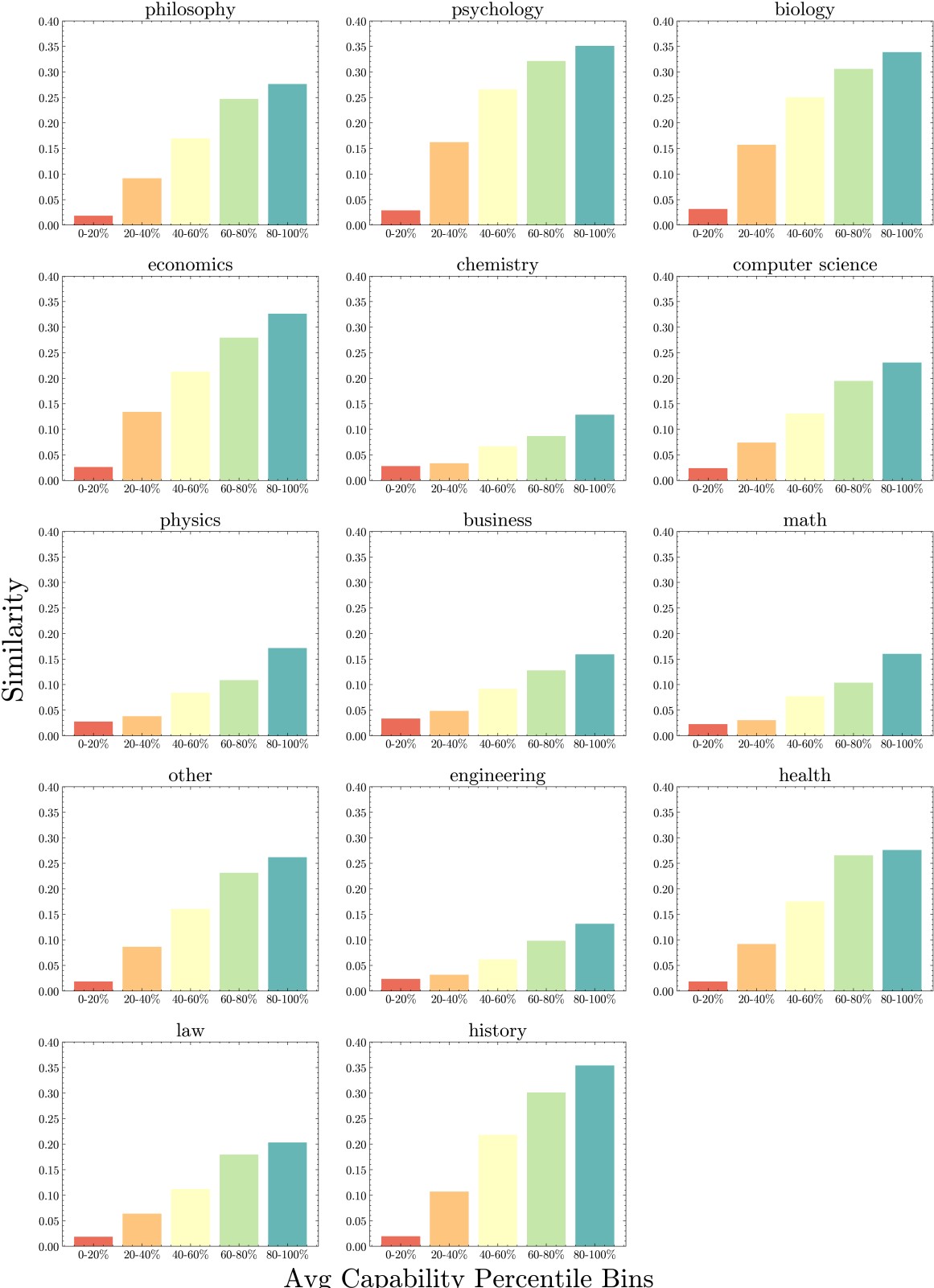

*Figure 22.* **LM Similarity ($\kappa_p$) vs Capability on MMLU pro for each subject.** The increasing trend holds for all 14 subjects in MMLU pro. The similarity trend is therefore not a consequence of a particular domain or subject in MMLU Pro.

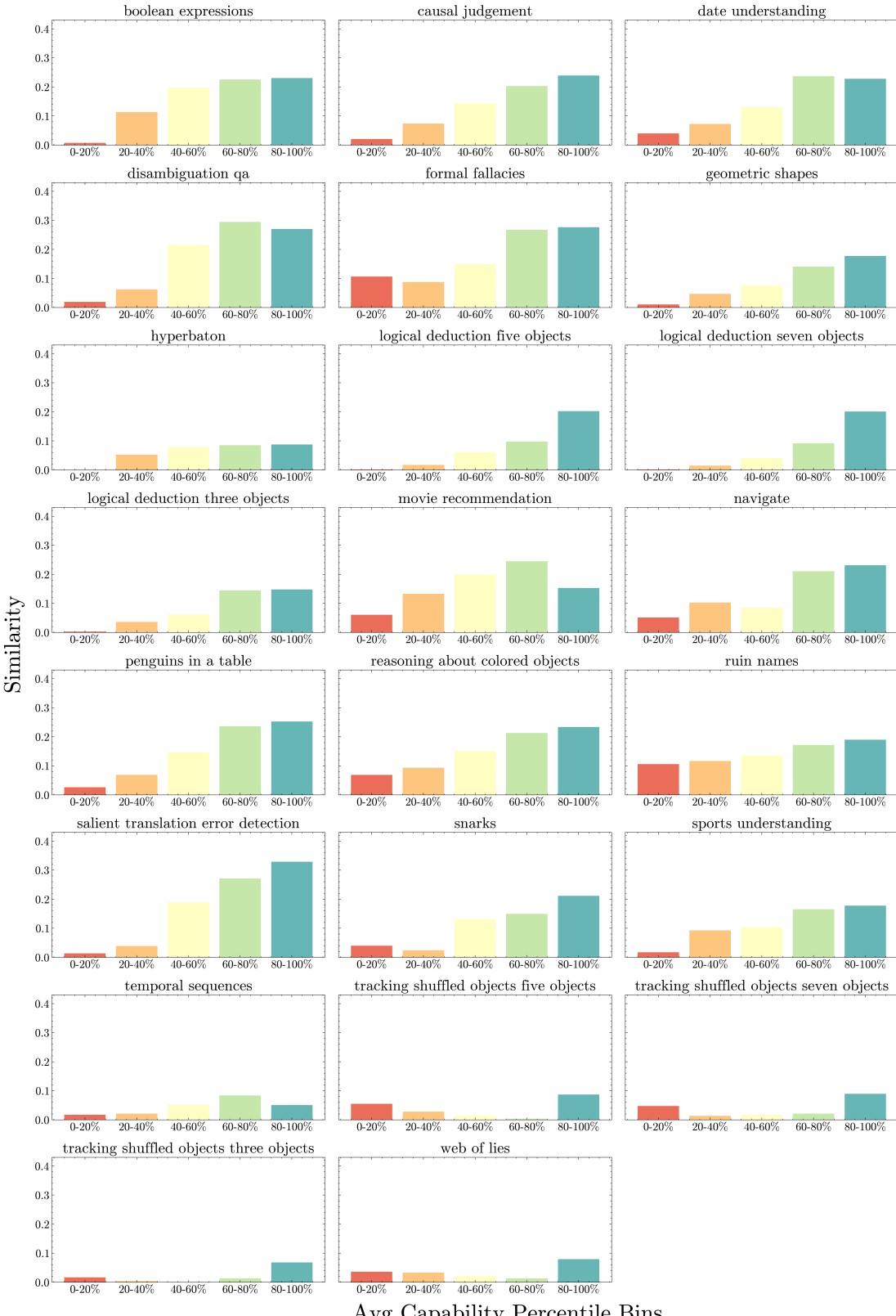

*Figure 23.* **LM Similarity ($\kappa_p$) vs Capability on each Big-Bench Hard task.** The increasing trend holds for most BBH tasks. Each task has atmost 250 questions, resulting in minimal data to compute similarity for the individual tasks.

