# OpenReview forum: "Great Models Think Alike and this Undermines AI Oversight"
_ICML.cc/2025/Conference — ICML 2025 spotlightposter_

### Official Review · Reviewer_nzM3 · 2025-03-09

**Overall Recommendation:** 2

**Summary:**

The authors design a metric for measuring model similarity, using a combination of model performance, the type of mistakes as well as probabilistic decisions made by the model. The authors demonstrate that for various pairs of models, there is a high correlation between the measured similarity between the pair as well as the evaluation score when using one model to judge another, as well as a bias for models to judge a similar model better. They then explore the setting of weak-to-strong supervised training with a teacher model. Finally, the authors make a final observation that as models become more capable, mistakes simultaneously become more similar across a number of different model families.

**Claims And Evidence:**

### Claim

Models favour similar models when used as a judge.

### Evidence

I believe that there is sufficient evidence to provide preliminary support for this claim; the LLM-as-a-Judge setting appears to demonstrate that models tend to favour similar models appears to be consistent, supporting this claim.

-----

### Claim

The model similarity metric, $\kappa_p$, is an accurate predictor of model similarity.

### Evidence

The authors use the model judgement score to show that the model similarity metric correlated well with judgement scores.

**Essential References Not Discussed:**

N/A.

**Experimental Designs Or Analyses:**

The design is appropriate. I believe that evaluating judgement scores and comparing them against the similarity metric is a sensible experiment. In Section 4, the experimental setup is appropriate (first evaluate for similarity, then train in a student-teacher setup) and there are no claims that appear problematic. In section 5, the authors again compare accuracy on two tasks (MMLU and BBH) and compare against accuracy, demonstrating a strong correlation between accuracy and similarity.

**Methods And Evaluation Criteria:**

The methods for evaluation are appropriate and the benchmarks/datasets are reasonably chosen based on the construction of the metric.

**Other Comments Or Suggestions:**

See above.

**Other Strengths And Weaknesses:**

### Strengths

- The work is clearly motivated and the empirical experiments are well designed.
- There is a clear existing use-cases for the metric the authors propose.

---
### Weaknesses

- The tasks that are used for measuring similarity appears to only be based off of accuracy. However, various tasks do not directly use hard ground-truth labels, such as those for summarization or instruction following. I believe a diversification in the number of tasks that are being evaluated for can better demonstrate the benefits of the method, or reveal some potential areas that need to be taken into consideration (ex. measuring similarity within reasoning chains, etc.).
- In Table 2, the correlation scores can vary quite a bit based on the model size. This may be the direct result of model capability, as the authors explore in Section 5, but I believe there may be a need to either account for model size within the metric computation or potentially better understand the relationship between the metric and model size.
- While I appreciate the comment that having similar models undermines oversight, I feel that claims about model similarity have been extant for an extended period of time. I do not particularly believe that this work offers a significantly novel perspective on this front, especially as many of the claims remain speculative in some part and as a result make it difficult to draw broader claims from. Furthermore, while the metric the authors introduce can be useful, it doesn't provide a direct solution towards mitigating the problem of AI oversight. Hence in my opinion, it may be better to reframe this perspective of the work in some respect.

**Questions For Authors:**

See strengths and weaknesses.

**Relation To Broader Scientific Literature:**

The results are complementary to work in evaluating the trustworthiness or robustness of LLMs, which have also shown a degree of bias in LLM-as-a-judge settings as well as overlap in LLM responses. The authors provide a new way to explicitly measure similarity, providing an interesting alternative interpretation to this setting.

**Theoretical Claims:**

N/A. Appendix A.6. computes some bounds but these are generally elementary mathematics therefore I do not see any issues with their correctness.

---

> ### Author Rebuttal · Authors · 2025-03-31
>
> We are glad you found our **metric useful, experiments well-designed, evidence sufficient, and motivation clear**. Given these positive points, we were a bit surprised by the recommendation. We hope the new analyses and clarifications below address the concerns.
>
> **(W1)**
> > The tasks that are used for measuring similarity appears to only be based off of accuracy. However, various tasks do not directly use hard ground-truth labels
>
> Our metric uses ground truth labels to prevent the inflation of similarity scores as we illustrate with [new plots](https://imgur.com/a/XL7OBwH). **In applications where ground-truth labels are not available, one can still use the observed agreement $c_{obs}^p$ as computed by our metric**. Evaluating open-ended tasks is an exciting direction of future work [L392-404].
>
> **(W1)**
> > I believe a diversification in the number of tasks that are being evaluated for can better demonstrate the benefits of the method
>
> We appreciate the reviewer's suggestion regarding task diversification. To further confirm the broad relevance of our method, **we extend our analysis to AlpacaEval, open-ended chat responses which do not have objective ground truth**. We find that MMLU-Pro benchmark similarity correlates strongly ($r=0.75$ to $r=0.87$) with judge preferences on AlpacaEval in [new plots](https://imgur.com/a/CIKQOmf). This reinforces conclusions in Sec 3, so we will include it in App. B of the revision. **We believe our experiments across MMLU-Pro (14 categories), BBH (23 categories, Sec 5), and 15 NLP tasks (Sec 4) are comprehensive, as noted by Reviewers RC4b and qUdH**.
>
> **(W2)**
> > In Table 2, the correlation scores can vary quite a bit based on the model size… better understand the relationship between the metric and model size.
>
> To check whether model size is a possible confounder, **we extend our multiple regression results by including model size as a predictor -- [new plots](https://imgur.com/a/y06LkCT)**. We find that similarity remains a significant predictor of judgement score when controlling for both model size and accuracy. Furthermore, the effect size of model size when controlling for similarity and accuracy is close to 0, confirming that our focus on model capability rather than characteristics is a sound approach, consistent with prior work [1].
>
> **(W3)**
> > I feel that claims about model similarity have been extant for an extended period of time. I do not particularly believe that this work offers a significantly novel perspective on this front”
>
> While there might have been a common feeling in the community about models being similar, **our work is the first to quantify it comprehensively (as highlighted by reviewer qUdH) at scale** — showing a concerning increase in similarity with improving capability by analyzing 130 LLMs from various developers. Our analysis of LLM-as-a-judge extends existing self-preference results showing mitigations like using separate judges [4] are not enough. In Sec 6, we discussed previous work that explored model differences, and we will also cite theoretical results like [3] in our revision, which study the effects of algorithmic monoculture on fairness.
>
> **(W3)**
> > many of the claims remain speculative in some part
>
> We are happy to clarify our claims and provide additional support in our revision. Could you please indicate which specific claims you found speculative?
>
> **(W3)**
> > while the metric the authors introduce can be useful, it doesn't provide a direct solution towards mitigating the problem of AI oversight
>
> We acknowledge your observation, our metric does not directly mitigate the impact of model similarity on AI oversight. Our paper's **primary goal was to establish a robust framework for quantifying this effect, which is a crucial first step** in addressing the issue. Specifically, our findings (1) shed light on the limitations of LLM-as-a-judge systems, popular in many leaderboards [L195-204], (2) offer insights into open problems in weak-to-strong generalization, as highlighted by reviewer qUdH, and (3) reveal an inverse scaling phenomenon [2] where increasing capabilities exacerbates these issues. By quantifying the problem, we set the stage for future research on potential mitigation strategies, as discussed in Sec 7.
>
> We thank you for the valuable feedback, which has made our paper stronger! Please let us know if there are any further questions or concerns we can resolve to increase your support for our work.
>
> **References**
>
> [1] Huh, Minyoung, et al. "Position: The platonic representation hypothesis." ICML 2024.
>
> [2] McKenzie, Ian R., et al. "Inverse scaling: When bigger isn't better." TMLR (2023).
>
> [3] Bommasani, Rishi, et al. "Picking on the same person: Does algorithmic monoculture lead to outcome homogenization?." *NeurIPS* (2022)
>
> [4] Decentralized Arena via Collective LLM Intelligence, [https://de-arena.maitrix.org](https://de-arena.maitrix.org/)

---

> > ### Comment · Reviewer_nzM3 · 2025-04-04
> >
> > I appreciate the authors’ time and effort spent in providing a response. After reading the response as well as considering the other reviews and the authors’ responses, I am willing to slightly raise my score.

---

### Official Review · Reviewer_qUdH · 2025-03-13

**Overall Recommendation:** 5

**Summary:**

The paper proposes a similarity metric between LLMs based on their logits, which measures the similarity of mistakes two models make on a task. The authors use this similarity metric to perform a variety of analyses. In particular they find similarity can predict scores from LLM judges, weak-to-strong generalization results, but also model capability predicts similarity, i.e., more capable models tend to be more similar to each other.

**Claims And Evidence:**

The authors make nuanced claims and I think all claims are well supported by evidence.

**Essential References Not Discussed:**

None

**Experimental Designs Or Analyses:**

I looked at all experiments and results and did not find any major issues with the analysis. Overall I find the experiments comprehensive, convincing and clearly presented. The authors are very clear about the claims they make and how the experiments test these claims. The experimental rigour is significantly greater than for a typical ICML paper.


Minor issue: I found the experimental setup for the weak-to-strong generalization experiments difficult to follow. Most relevant details are described in the appendix but I think the clarity in the main paper could be improved.

A specific detail I wasn't able to understand from the paper is: "as shown in Table 3, the actual ceiling is significantly higher if complementary knowledge of the weak supervisor is fully leverage"

Do you finetune the model first on ground truth labels and then on the outputs of the weak supervisor?

**Methods And Evaluation Criteria:**

Yes all experiments and analyses make sense.

The LLM-as-a-judge experiments would benefit from looking at a wider range of settings / datasets.

**Other Comments Or Suggestions:**

none

**Other Strengths And Weaknesses:**

Some results around the weak-to-strong generalization experiments are difficult to understand from the main paper.

**Questions For Authors:**

* Could you explain the training of the complementary knowledge ceiling model in the weak-to-strong generalization experiments?

**Relation To Broader Scientific Literature:**

The paper significantly improves upon previous work measuring model similarity, both in terms of the actual similarity metric proposed (which is a "method" contribution) as well as in the breadth and depth of the experimental analysis based on this metric. For example the weak-to-strong generalization experiments provide insides about some of the key open questions discussed in Burns et al. 2024.

Overall, I think this paper is a strong contribution and will be of great interest to the ICML community.

**Theoretical Claims:**

No theoretical claims.

---

> ### Author Rebuttal · Authors · 2025-03-31
>
> We are grateful for your strong support of our work. We are glad you that you found our claims nuanced and well-supported by comprehensive, rigorous experiments.
>
> On your question about how we compute the “elicitation $\cup$ complementary” ceiling estimate in Table 3, we should indeed have made this clearer in the main paper itself. We take the union of samples either the strong model fine-tuned on ground-truth labels (elicited) or the weak supervisor (complementary) get right. This is likely an overestimate of what can be realised from weak-to-strong training, but we think that is a lesser harm for a ceiling estimate. The previously proposed ceiling of ground-truth elicitation on the strong model ignores potential gain from complementary knowledge, which can be significant as indicated in Figure 4 and 12. We think such underestimation is worse for a ceiling estimate, as it might explain concurrent work [1] reporting >100% PGR, beating the “ceiling” of ground-truth elicitation on the strong model. Thus, we avoid estimating the ceiling by methods that can introduce any suboptimalities. For example, if we sequentially finetuned on ground-truth labels and then the weak supervisor's predictions or vice-versa, this could lead to catastrophic forgetting of the first part of training instead of fully combining their knowledge as desired.
>
> We agree with you that many important details about the weak-to-strong setup are in the Appendix. This is due to the limited space for the initial submission. In the camera ready, we will utilise the extra page available to shift setup details for the weak-to-strong experiments currently in Appendix C.1 and C.2 back to Section 4.
>
> We are happy to discuss any further questions about the weak-to-strong and other experiments!
>
> [1] Shi, Junhao, et al. "How to Mitigate Overfitting in Weak-to-strong Generalization?." *arXiv* (2025).

---

### Official Review · Reviewer_WJR8 · 2025-03-13

**Overall Recommendation:** 3

**Summary:**

This paper investigates the challenges of AI oversight when using Language Models (LMs) to evaluate and supervise other LMs. It highlights how model similarity can undermine oversight, as similar models tend to make correlated mistakes and exhibit affinity bias—where they rate outputs from similar models more favorably.

To address this, the authors introduce a novel metric, κp, that accurately measures the functional similarity of LMs based on overlapping mistakes while accounting for chance agreement and output probabilities. The study also demonstrates that greater diversity between supervising and student models enhances generalization and learning.

The paper emphasizes the risk that, as LMs become more capable, their mistakes become more similar, which may amplify oversight blind spots. It calls for better diversity in oversight models, transparent reporting of model similarities, and more robust evaluation frameworks to ensure reliable AI oversight.

**Claims And Evidence:**

•	Claim 1: “LLM judges exhibit affinity bias, favoring models similar to themselves.”
It is supported with relatively convincing analysis. However, While Figure 2 suggests family-level bias, it lacks a clear statistical comparison between “same family” and “different family” pairs.

•	Claim 2: “Training gains from weak-to-strong generalization are higher when the supervisor and student models are less similar.”
The authors provide analysis to support this claim. However, the paper’s conclusions about complementary knowledge are entirely dependent on the κp metric. Moreover, the selection criteria for “weak” vs. “strong” models are not transparently discussed.

•	Claim3 : “Model mistakes are becoming more similar with increasing capabilities, which is concerning for AI safety.”
The study only shows a correlation between model capability and similarity of mistakes. There is no causal evidence to prove that increasing capabilities cause more similar mistakes. Additionally, it fails to account for key confounders that could explain the increasing similarity trend. For example, training data overlap: many LLMs use similar datasets, which naturally biases them towards similar mistakes.

**Essential References Not Discussed:**

None

**Experimental Designs Or Analyses:**

•	In section 3, the authors control for accuracy using partial correlations and regression but neglect other possible confounding factors such as: training data overlap. Models from the same family may have been trained on overlapping datasets, leading to similar outputs and thus higher judgment scores.
•	In section 5, there is no real-world experiment demonstrating how these similarities lead to oversight failures, so increasing similarity is a “safety concern” is questionable.

**Methods And Evaluation Criteria:**

•	While the authors provide evaluations in two different usage scenarios for LMs, the benchmark datasets are limited. In section 3, there is only dataset MMLU-Pro being used.
•	Additionally, there is no evaluation of the qualitative nature of mistakes such as factual errors, reasoning errors etc. Thus, it’s lack of insights to connect to/improve the real LMs applications.

**Other Comments Or Suggestions:**

•	Figure 2 & 3 are really informative, but also overwhelming. It could be hard to comprehend all the details and interpret the analysis based on it.
•	For 3.2 Q1, it will be better to provide the table where the data is from. The single example is difficult to validate the analysis.

**Other Strengths And Weaknesses:**

Strengths:
•	Novel Contribution: κp Similarity Metric
•	Utilizes partial correlations and multiple regression to control for accuracy when analyzing judgment bias
Weakness:
•	No rigorous comparison with alternative similarity metrics (e.g., KL Divergence, JSD, RSA).
•	Some confounders are not sufficiently controlled in the experimental designs, including: training data, architectural similarities.
•	Limited casual evidence in similarity trends
•	Lack of qualitative error analysis

Overall, while the paper shows the strong motivation and good effort in experiment design, there are many critical information and analysis missing to convince the audience and provide better insights

**Questions For Authors:**

As listed in previous parts and weakness section.

**Relation To Broader Scientific Literature:**

The κp metric extends traditional agreement metrics like: Cohen’s κ (Cohen, 1960): Measures inter-annotator agreement while adjusting for chance; Krippendorff’s α (Krippendorff, 1970): A more flexible metric for measuring agreement with chance correction; Error Consistency Metrics (Burns et al., 2022): Focused on measuring consistency in model mistakes.

**Theoretical Claims:**

•	The paper lacks theoretical proofs explaining why κp captures probabilistic differences better than divergence metrics.
•	The derivation of chance agreement (cpexp) simplifies the assumption by uniformly distributing probabilities across incorrect options. However, it doesn’t consider tasks where wrong answers aren’t uniformly distributed or where some options are systematically more likely than others.

---

> ### Author Rebuttal · Authors · 2025-03-31
>
> We are glad you liked the motivation and experimental design. We hope our new analyses address your concerns.
>
> **(W1)**
> > Comparison with alternative similarity metrics (e.g., KL Divergence, JSD, RSA)
>
> Divergence metrics have desirable information theoretic properties, but **we did not find any way to adjust the effect of higher accuracy inflating the similarity reported by these metrics** [L88-99 (R), Table 1]. We re-plot Figs. 6-8 with JSD, a symmetric and bounded variant of KL, to demonstrate this -- [new plots](https://imgur.com/a/XL7OBwH), e.g., the second plot studies 90% accurate models with independent errors, where $JSD > 0.75$, but as desired, $k_p \sim 0$.
>
> App. A4 discusses design choices for $\kappa_p$ by comparing with many alternate similarity metrics, including why we use probabilistic overlap instead of JSD [L1029-1031]. Here, we will also include that **alternative priors, if known, can be used to compute chance agreement $c^p_{exp}$** on *``tasks where wrong answers aren’t uniformly distributed''*, thanks for this suggestion. Finally, we focus on functional similarity metrics and not model-dependent representational similarity analysis (RSA) [L60-80 (R)].
>
> **(W2, W3)**
> > Some confounders are not sufficiently controlled …, including: training data, architectural similarities; Limited casual evidence in similarity trends
>
> We agree deeper analysis of the causal factors is an important next step [L386-390 (R)]. **We tried our best not to make causal claims in the paper, and will rephrase any we overlooked**.
>
> We will acknowledge training data overlap as a potential reason for our observations. Unfortunately, training data mixtures are not known even for open-weight frontier models to control for this. As requested, we now exclude *``same-family pairs''* for Fig. 2 in [new plots](https://imgur.com/a/qKr2yi3). **Judge scores remain highly correlated** with similarity $r \in \[0.89, 0.95\]$. Further, [new table](https://imgur.com/a/y06LkCT) controls for *model size* as requested by reviewer nzM3. **Similarity still has high partial correlation**.
>
> For similarity trends with increasing capability, we already excluded same-family pairs in Fig. 5. In Sec 5.2 we mention how switching to **state space models (Mamba) does not reduce similarity**. App D.2 has more details and observations, showing **sample-hardness has a small effect on this trend, while instruction tuning exacerbates it**. We are happy to demonstrate further controls if you have any suggestions.
>
> > the benchmark datasets are limited. In section 3, there is only dataset MMLU-Pro
>
> In Sec 3, we were constrained to tasks which can be evaluated as both free-response and MCQ [L199-203]. This is why we cannot use benchmarks like BBH in Sec 5 and the 15 NLP tasks in Sec 4. **MMLU-Pro itself is quite diverse, and [new plots](https://imgur.com/a/grG06X1) show consistent results across its 14 domains ranging from law to engineering**. In fact, could this diversity lead to similarities on MMLU-Pro predicting judgements on more open-ended tasks? **[New plot](https://imgur.com/a/CIKQOmf) shows MMLU-Pro similarity has high correlation with LM judge based elos on the chat response dataset AlpacaEval 2.0** [1].
>
> **(Evidence for W2S)**
> > conclusions about complementary knowledge are entirely dependent on the κp metric
>
> We forgot to add a pointer in our main paper to Fig 13 which shows that the **complementary knowledge conclusion consistently holds with 1 - JSD, though $\kappa_p$ explains more variance ($r=0.77$ vs $0.85$)** and has better normative grounding for analyzing model similarity trends as discussed in Sec 2. We will mention the *``selection criteria for weak vs strong models''* is model size, consistent with Burns et al. (2024) and Scherlis et al. (2024) in the revision.
>
> **(W4)**
> > no evaluation of the qualitative nature of mistakes.
>
> Qualitative analysis of mistakes is an interesting related, yet complementary direction [L360-369 (R)]. We respectfully disagree that this implies a *``lack of insights to connect to/improve the real LMs applications''*. Our quantitative measurement of similarity provides a **necessary foundation for applications like evaluating qualitative descriptions of model differences [L431-437 (R)], debiasing LLM-as-a-judge based leaderboards [L195-205]**, and many exciting directions for future work discussed in Sec 7.
>
> > For 3.2 Q1, provide the table where the data is from.
>
> [New tables](https://imgur.com/a/7XSrXt3) have the data of all 351 model pairs studied. To make this data easier to interpret, we reported it as scatter points in Fig 2. In the final version, we will link our **PyPi package and interactive tool** to aid readers in exploring similarities of chosen model pairs.
>
> Thanks! Your detailed feedback has helped us greatly improve the paper. We hope this increases your support for our work.
>
> [1] Dubois, Yann, et al. "Length-Controlled AlpacaEval: A Simple Debiasing of Automatic Evaluators." *COLM* (2024)

---

### Official Review · Reviewer_RC4b · 2025-03-14

**Overall Recommendation:** 4

**Summary:**

This paper introduces a probabilistic metric for model similarity that adjusts for chance agreement due to accuracy, distinguishes different types of mistakes, and incorporates confidences. Using this metric, the authors reveal three key insights:
1. LM judges demonstrate affinity bias, favoring models similar to themselves.
2. Greater gains in weak-to-strong generalization occur when training strong student models with annotations generated by weak supervisors that are more different from the student.
3. Model errors get more correlated as model capabilities increase, raising concerns about correlated failures in AI oversight.

**Claims And Evidence:**

Yes. While I couldn't verify all the materials in the appendix, the main results in the paper seem reasonable.

**Essential References Not Discussed:**

To my knowledge, the paper discussed the essential references in the field.

**Experimental Designs Or Analyses:**

The experimental designs and analyses that I reviewed are comprehensive and demonstrative of the claims made in the paper.

**Methods And Evaluation Criteria:**

Yes, the methods and evaluation criteria proposed in the paper are reasonable. The proposed similarity metric is well-motivated, with the relation and difference compared to the existing metrics clearly explained.

**Other Comments Or Suggestions:**

Regarding the definition of "observed error overlap" on the right of line 87, it may be worth remarking why "the fraction of samples on which both models are correct or both models are wrong" is a more reasonable metric than things like "the fraction of samples on which the two models agree (in the multi-class setting)". Is the binary nature of the definition crucial here?

**Other Strengths And Weaknesses:**

Strengths:
- The proposed similarity metric $\kappa_p$ is well-motivated and addresses the limitations of existing metrics.
- The observation on negative correlation between weak-to-strong generalization and model similarity is quite insightful.
- The observation on the potential risk of correlated failures in AI oversight is important and thought-provoking.

(Minor) weaknesses:
- The proposed similarity metric $\kappa_p$ seems to be tailored for MCQ tasks. It would be insightful if some discussions on possible extensions to other tasks (e.g., regression) could be provided, even just as future directions.
- The negative correlation between weak-to-strong generalization and model similarity is arguably surprising. It would be helpful to provide more insights or possible explanations for this phenomenon.

**Questions For Authors:**

Some questions are raised in the "Other Strengths And Weaknesses" section. I do not have further questions.

**Relation To Broader Scientific Literature:**

This work introduced a probabilistic metric for model similarity that is built upon the existing metrics while addressing their limitations. Some insightful observations on AI oversight are provided based on the proposed metric, which further solidifies the broader impact of this work.

**Theoretical Claims:**

The proofs in Appendix A are reasonable.

---

> ### Author Rebuttal · Authors · 2025-03-31
>
> We appreciate that you found our metric well-motivated (with comparisons to alternatives), experiments comprehensive, and observations insightful.
>
> > The proposed similarity metric seems to be tailored for MCQ tasks. It would be insightful if some discussions on possible extensions to other tasks (e.g., regression) could be provided, even just as future directions.
>
> Indeed, our metric as described in the main paper is designed to address a key challenge in MCQ tasks—specifically, the lack of coherent classes needed to compute the marginal distributions for inter-annotator agreement metrics like Cohen’s κ [L976-979]. You might find Appendix A.3 interesting, where we extend the metric to classification, along with a brief discussion of challenges for exact match settings, for which we only provide a discrete version.
>
> On the other hand, for free-response tasks like creative writing, one could use both perplexity and model-based metrics, which have their own challenges as discussed in L392-403. That said, we ran experiments where we compute similarity on MMLU-Pro MCQs and plot Elo ratings assigned to evaluated models by making an LLM-judge pick between open-ended chat responses on AlpacaEval -- [New Plot](https://imgur.com/a/CIKQOmf). The strong correlation shows initial evidence that MCQ similarity might transfer across tasks as a predictor.
>
> For tasks like regression, for the observed agreement, $c_{obs}$, one could measure a distance metric over the two models' predictions, aggregating across samples. Once again, models with lower error would have lower distance in prediction, so the challenge lies in defining chance agreement, $c_{exp}$, for a model with a given error. We would have to make appropriate assumptions about the distribution of errors, such as gaussian errors, based on which $c_{exp}$ can then be computed. Thanks for this interesting question; We will add this discussion to Appendix A.3. We are excited about adapting the metric for other tasks in future work!
>
> > The negative correlation between weak-to-strong generalization and model similarity is arguably surprising. It would be helpful to provide more insights or possible explanations for this phenomenon.
>
> The result can seem surprising if we view weak and strong models purely through the lens of accuracy. This is where we think our framing of model similarity (or difference) at a sample level is insightful! Lower accuracy does not imply that the knowledge of weak models is a strict subset of stronger ones. Rather, weak models can have complementary knowledge, and we hypothesise the transfer of this also contributes to weak-to-strong generalization. Model similarity provides a way to measure complementary knowledge in terms of the difference in samples they get right. Lesser complementary knowledge to transfer might be the explanation for the seemingly surprising trend of lower weak-to-strong generalization when model similarity is higher (negative correlation). We tried to motivate this in L248-260. We will be sure to utilise the extra page allowed in the revision to expand this section, by including content currently in Appendix C.1 and C.2.
>
> > “Regarding the definition of "observed error overlap" on the right of line 87, it may be worth remarking why "the fraction of samples on which both models are correct or both models are wrong" is a more reasonable metric than things like "the fraction of samples on which the two models agree (in the multi-class setting)". Is the binary nature of the definition crucial here?”
>
> In L87, we are stating the definition used for error consistency defined in Geirhos et al. (2020). Our own metric modifies this to measure “the fraction of samples on which two models agree”, just as you proposed. We agree it’s a better definition, as it distinguishes differing mistakes [L129-133]. As you correctly noticed, our metric $\kappa_p$ is equivalent to error consistency (Geirhos et al.) for binary classification [L894-896, Appendix A.1]. Great minds do think alike ;)
>
> Thanks for your question and suggestions. We hope our response increases your support for our work and we are happy to discuss further!

---

> > ### Comment · Reviewer_RC4b · 2025-04-02
> >
> > I appreciate the authors’ detailed responses to my questions. After considering the other reviews and the authors’ responses, I remain convinced that this work offers strong empirical evidence and valuable insights. I will maintain my current evaluation.

---

### Decision · Program_Chairs · 2025-05-01

**Decision:**

Accept (spotlight poster)

**Comment:**

This paper offers two contributions. First, it introduces a novel metric called Chance Adjusted Probabilistic Agreement (CAPA), which improves upon the previously used error consistency metric. CAPA is effective because it can distinguish between different types of errors and captures probabilistic aspects of model predictions. Second, utilizing this new metric, the paper provides analyses demonstrating affinity bias in LLM-as-a-judge scenarios, evidence that complementary knowledge offers advantages beyond elicitation in the weak-to-strong generalization setup, and an observation that models tend to make more similar mistakes as their capabilities increase.

Following the rebuttal phase, three reviewers expressed positive opinions (WA, A, SA), and the authors adequately addressed earlier concerns such as lack of benchmarking, evaluation and experimental clarity, and the novelty of their contributions compared to existing literature and metrics. One reviewer remained skeptical with a WR score due to the speculative claims and lack of direct solution towards mitigating the problem of AI oversight. In my view, the contributions presented in this paper are sufficiently significant, even considering any remaining concerns such as lack of a direct solution. Therefore, I recommend acceptance.